# Cellular compartmentalisation and receptor promiscuity as a strategy for accurate and robust inference of position during morphogenesis

Krishnan S Iyer[1], Chaitra Prabhakara[2], Satyajit Mayor[2]*, Madan Rao[1]*

[1]Simons Center for the Study of Living Machines, National Center for Biological Sciences - TIFR, Bangalore, India; [2]National Center for Biological Sciences - TIFR, Bangalore, India

**Abstract** Precise spatial patterning of cell fate during morphogenesis requires accurate inference of cellular position. In making such inferences from morphogen profiles, cells must contend with inherent stochasticity in morphogen production, transport, sensing and signalling. Motivated by the multitude of signalling mechanisms in various developmental contexts, we show how cells may utilise multiple tiers of processing (compartmentalisation) and parallel branches (multiple receptor types), together with feedback control, to bring about fidelity in morphogenetic decoding of their positions within a developing tissue. By simultaneously deploying specific and nonspecific receptors, cells achieve a more accurate and robust inference. We explore these ideas in the patterning of *Drosophila melanogaster* wing imaginal disc by Wingless morphogen signalling, where multiple endocytic pathways participate in decoding the morphogen gradient. The geometry of the inference landscape in the high dimensional space of parameters provides a measure for robustness and delineates *stiff* and *sloppy* directions. This distributed information processing at the scale of the cell highlights how local cell autonomous control facilitates global tissue scale design.

*For correspondence:
mayor@ncbs.res.in (SM);
madan@ncbs.res.in (MR)

## Editor's evaluation

The manuscript introduces a compelling theoretical framework to investigate architectures of signal processing. The predictions of the computational model have been convincingly validated with data from fly wing precursor tissues. The work is important and will be highly valuable to biological physicists and developmental biologists interested in morphogenesis and pattern formation.

## Introduction

Precise positioning of cell fates and cell fate boundaries in a developing tissue is of vital importance in ensuring a correct developmental path (reviewed in *Tkačik and Gregor, 2021*; *Wolpert, 2016*). The required positional information is often conveyed by concentration gradients of secreted signalling molecules, or morphogens (reviewed in *Tabata and Takei, 2004*; *Briscoe and Small, 2015*). Typically, a spatially varying input morphogen profile is translated into developmentally meaningful transcriptional outputs. Morphogen profile measurements, across several signalling contexts, show that the gradients are inherently noisy *Houchmandzadeh et al., 2002*; *Gregor et al., 2007a*; *Kicheva et al., 2007*; *Bollenbach et al., 2008*; *Zagorski et al., 2017*. However, precision of the signalling output should be robust to inherent genetic or environmental fluctuations in the concentrations of the ligands and receptors engaged in translating the positional information. For example, the noisy profile of the

morphogen Bicoid (Bcd) that activates hunchback (hb) in the early *Drosophila* embryo *Gregor et al., 2007a*; *Gregor et al., 2007b* , and the expression of gap genes that activate pair-rule genes *Dubuis et al., 2013*; *Petkova et al., 2019* result in cell fate boundaries that are positioned to a remarkable accuracy of about one cell's width. This points to a local, cell autonomous morphogenetic decoding that is precise and robust to various sources of noise *Kerszberg and Wolpert, 2007*; *Kerszberg, 2004*; *Jaeger et al., 2004*.

Cell autonomous decoding of noisy morphogen profiles includes reading of morphogen concentration, followed by cellular processing, finally leading to inference in the form of transcriptional readout. Several strategies have been proposed to ensure precision in output (reviewed in *Barkai and Shilo, 2009*; *Lander et al., 2009*): feedbacks such as self-enhanced morphogen degradation *Eldar et al., 2002*; *Eldar et al., 2003*, spatial and temporal averaging *Gregor et al., 2007a*, use of two opposing gradients *McHale et al., 2006*, pre-steady state patterning *Bergmann et al., 2007* and serial transcytosis *Bollenbach et al., 2007*.

Most cell signalling systems have regulatory mechanisms that fine-tune signalling by controlling ligand-specific receptor interactions *Rogers and Schier, 2011*. Ligands such as TGF $\beta$/BMP *Mueller and Nickel, 2012*, *Jiang and Cong, 2016*, *D'Souza et al., 2008*, show promiscuous interactions with different receptors. *Chen and Schier, 2002*; *Sick et al., 2006* or sequestering components within the extracellular matrix *Marjoram and Wright, 2011* or interactions with binding receptors such as heparin sulphate proteoglycans (HSPGs) *Baeg et al., 2001*; *Baeg et al., 2004*; *Yan and Lin, 2009* can control availability of the ligand. Additionally, the multiple endocytic pathways that operate at the plasma membrane can control the extent of signalling *Bökel and Brand, 2014*; *Di Fiore and von Zastrow, 2014*. These examples argue for *distributed information processing* within the cell.

In this paper, we show how cellular compartmentalisation, a defining feature of multicellularity, provides a compelling realisation of such distributed cellular inference. We show that compartmentalisation together with multiple receptors, receptor promiscuity and feedback control, ensure precision and robustness in positional inference from noisy morphogen profiles during development. Compartments associated with specific chemical (e.g. lipids, proteins/enzymes) and physical (e.g. pH) environments, have been invoked as regulators of biochemical reactions during cellular signalling and development *Ellisdon and Halls, 2016*; *Omerovic et al., 2007*; *Omerovic and Prior, 2009*; *Shilo and Schejter, 2011*; *Bökel and Brand, 2014*. Deploying promiscuous receptors against a morphogen, in addition to its specific receptor, is a strategy to buffer variations in morphogen levels. These observations provide the motivation for a general conceptual framework for *morphogenetic decoding* based on a multi-tiered, multi-branched information channel. While our framework has broader applicability, we will, for clarity, use the terminology of Wingless signalling in *Drosophila* wing imaginal disc *Hemalatha et al., 2016*.

## Conceptual framework and quantitative models

We pose the task of morphogenetic decoding as a problem in local, cell autonomous inference of position from a morphogen input (*Figure 1*), where each cell acts as an information/inference channel with the following information flow:

1. 'reading' of the morphogen input by receptors on the cell surface,
2. 'processing' by various cellular mechanisms such as receptor trafficking, secondary messengers, feedback control, and
3. 'inference' of the cell's position in the form of a transcriptional readout.

At a phenomenological level, *reading* of the morphogen input is associated with the binding of the morphogen ligand to various receptors with varying degree of specificity, leading to the notion that the information channel describing positional inference must possess *multiple branches*. Furthermore, the multiple *processing* steps associated with compartmentalisation of cellular biochemistry and/or signal transduction modules, for example phosphorylation states, provide the motivation for invoking *multiple tiers* in the channel architecture. At an abstract level, one may think of the branch-tier architecture of the cellular processing as a bipartite Markovian network/graph *Hartich et al., 2014*, with a *fast* direction (involving multiple branches) consisting of ligand-bound and unbound states along with chemical state changes, and a *slower* direction (involving multiple tiers) consisting of intracellular transport, fission and fusion, characterised by energy-utilising processes or a flux imbalance. A general developmental context with multiple morphogens may involve several such bipartite Markov

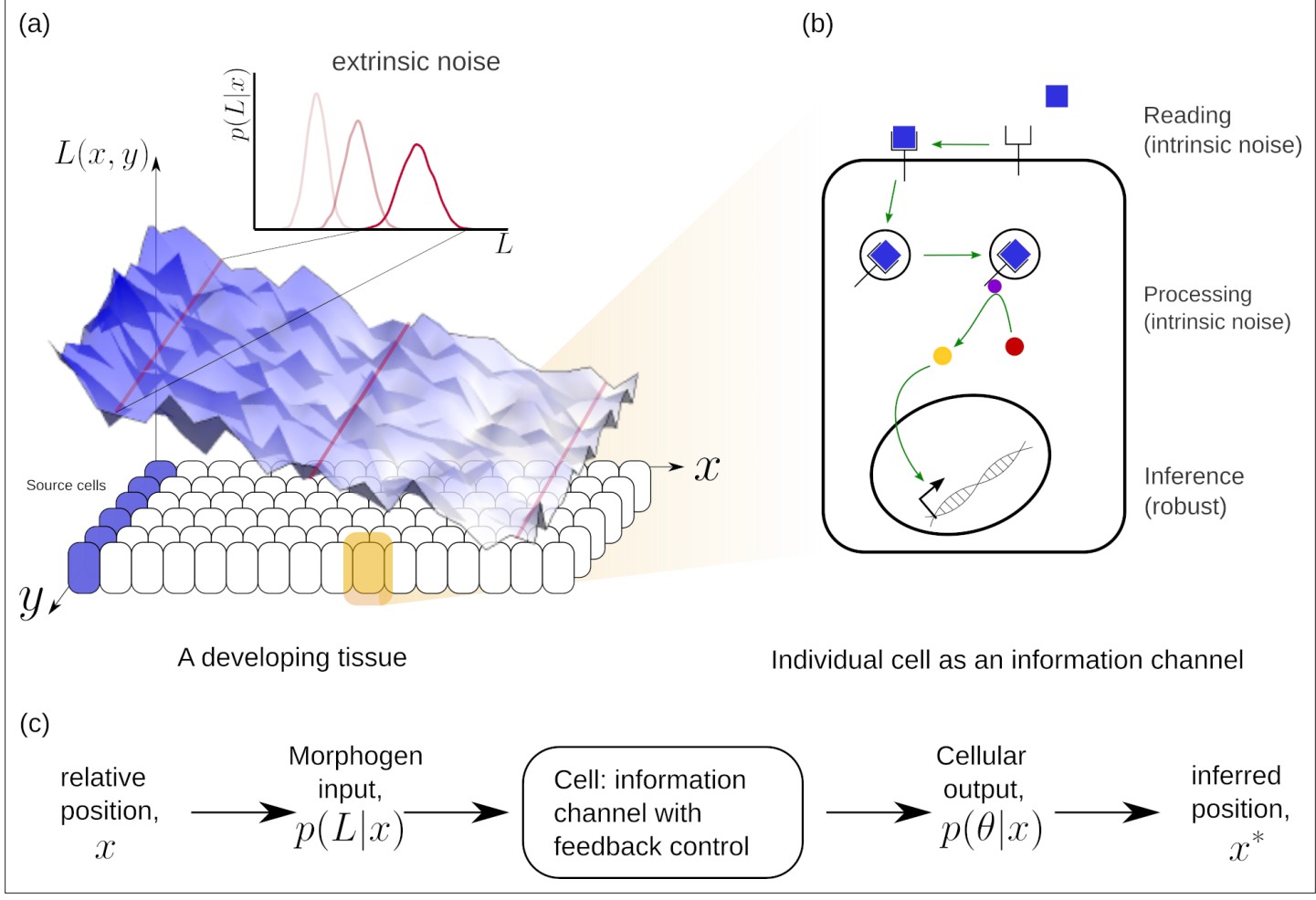

**Figure 1.** Schematic of information processing in the developing tissue. (**a**) A morphogen is produced by a specific set of cells (blue), and secreted into the lumen surrounding the tissue. Due to stochasticity of the production and transport processes, the morphogen concentration received by the rest of the cells is contaminated by *extrinsic noise*, which defines a distribution of morphogen concentration along the $y$-direction at any position $x$. (**b**) The route from morphogens to a developmental outcome requires each cell to read, process and infer its position. This task is further complicated by the stochasticity of the reading and processing steps themselves, that lead to *intrinsic noise*. (**c**) The problem of robust inference of position can be considered in a channel framework. The positional information is noisily encoded in the local morphogen (ligand) concentrations, $p(L|x)$. The cells receive this as input and process it into a less noisy output to ensure robustness in inferred positions.

networks/graphs with different receptors (or branches) in parallel. Some of these receptors could be shared between different morphogens. We refer to *signalling* receptors as those which transduce a signal upon binding to their specific morphogen ligand and *non-signalling* receptors as those that participate in the signalling pathway without directly eliciting a signalling response. At the end of processing, each individual cell may pool information from the various branches for the final inference of position, i.e. a transcriptional readout (***Figure 2***).

The task of achieving a precise inference is complicated by the noise in morphogen input arising from both production and transport processes, and by the stochasticity of the reading and processing steps; thus the *inference* must be robust to the *extrinsic* and *intrinsic* sources of noise. The use of feedback control mechanisms is a common strategy to bring about robustness in the context of morphogen gradient formation and sensing ***Averbukh et al., 2017***. Motivated by this, in Section 'Quantitative models for cellular reading and processing' we consider different feedback controls in conjunction with the tiers and branches. With these three elements to the channel architecture, the task of morphogenetic decoding can be summarised in the following objective.

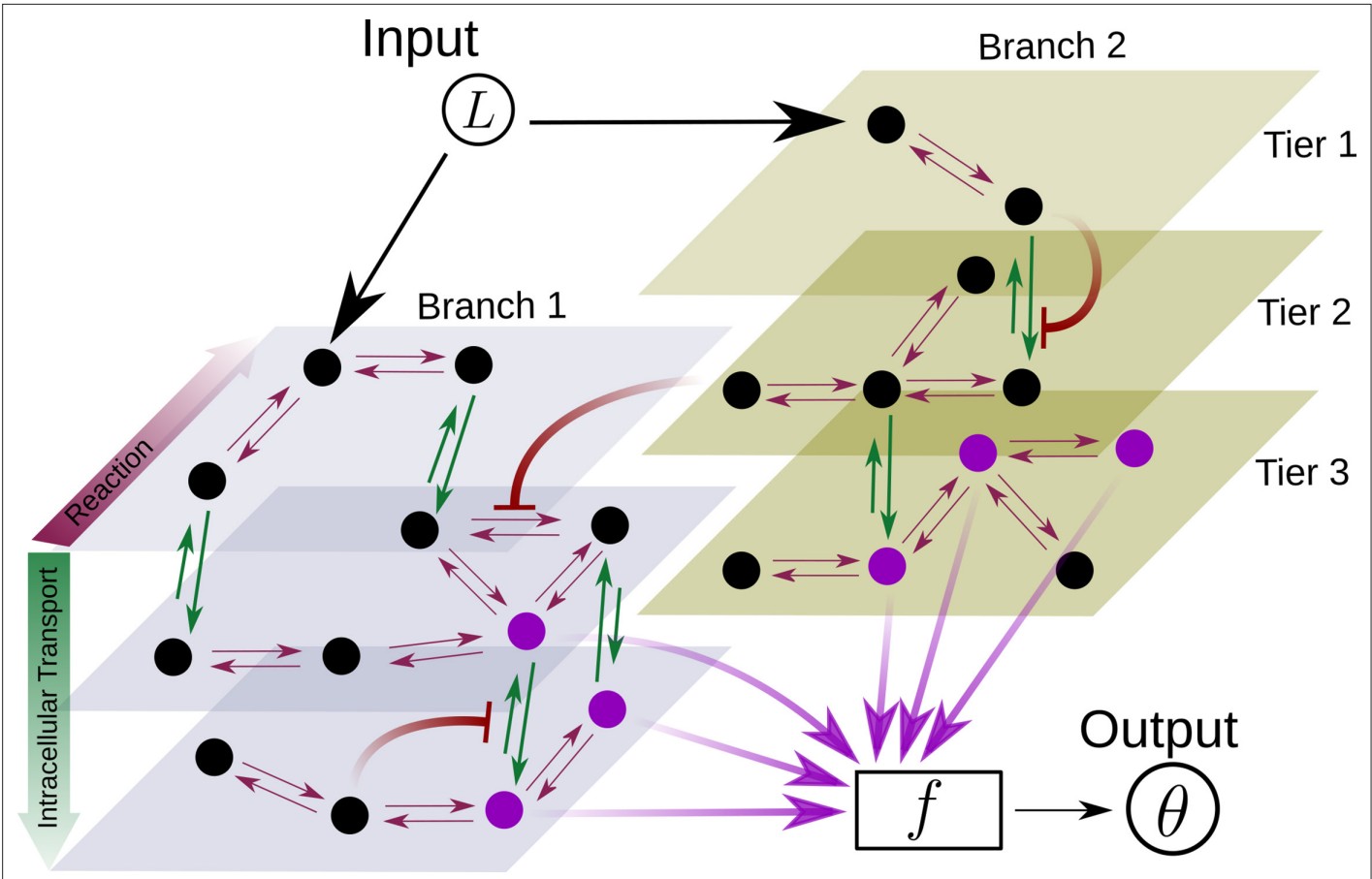

**Figure 2.** Schematic for the branch-tier channel architecture. *Branches* correspond to different receptor types and *tiers* denote the layers of compartmentalisation used in cellular processing. Cellular processing associated with each receptor type (here, branches 1 and 2) is depicted by a generic Markov network. The gray and brown planes depict the tiers in the two branches respectively (here, tiers 1, 2, and 3 in each branch). The bi-directional in-plane purple arrows correspond to faster transitions between receptor states, e.g. bound/unbound, and the green bi-directional arrows depict slower transitions involving intracellular transport driven by flux-imbalanced processes. There may exist several feedback control loops (red ⊣ arrows) in the network. Ligand concentration $L$ drives one or several reaction rates in such Markov networks as in *Harvey et al., 2020*. The output $\theta$ is a collection $f$ of several signalling states (purple nodes) from one or many branches. The statistics of the output $\theta$ then enables inference of position.

---

Objective:

---

Given a noisy ligand input distribution at position $x$, i.e. $p(L|x)$, what are the requirements on the reading (number of receptor types and receptor concentrations) and processing steps (number of tiers and feedback type) such that the positional inference is precise and robust to extrinsic and intrinsic noise?

---

## Mathematical framework

*Figure 1* describes information processing during development across a two dimensional tissue of $n_x$, $n_y$ cells in $x$ and $y$ directions, respectively. The direction of morphogen gradient is taken to be along $x$, with the morphogen source to the left of $x = 0$. Each cell is endowed with a chemical reaction network (CRN) with the same multi-tiered, multi-branched architecture with feedbacks described previously, that reads a noisy input $L(x, y)$ (morphogen concentration) and produces an 'output' (biochemical 'signal') $\theta(x, y)$ that is also noisy. Here, we choose to construct the noisy morphogen profile in the following manner: for a given position $x \in [0, 1]$, cells along the $y$-direction see different amounts of ligand coming from the same *input distribution* $p(L|x)$,

$$p(L|x) = \frac{2}{\sqrt{2\pi\sigma_L^2(x)}} \mathrm{Exp}\left[-\frac{(L-\mu_L(x))^2}{2\sigma_L^2(x)}\right]\left(1 + \mathrm{Erf}\left[\frac{\mu_L(x)}{\sqrt{2}\sigma_L(x)}\right]\right)^{-1}. \tag{1}$$

characterised parametrically by a mean $\mu_L(x)$ and standard deviation $\sigma_L(x)$. Experimental data can be fit to this distribution *Equation 1* (or another distribution suitable for the specific experimental system) to obtain the parameters. Here, we consider an exponentially decaying mean $\mu_L$ and standard deviation $\sigma_L$.

$$\mu_L(x) = Ae^{-x/\lambda} \tag{2}$$

$$\sigma_L(x) = \sqrt{\mu_L(x)} \tag{3}$$

Alternatively, one could choose a different parametrisation consistent with experimental observations for a morphogen profile with a monotonically decaying mean. The values of $A, \lambda$ chosen for our analysis are listed in *Table 1*. The corresponding output distribution $p(\theta|x)$ can be used to infer the cell's position. Since we do not know the precise functional relationship between the output and inferred position, we invoke Bayes rule *MacKay and Mac Kay, 2003*, as in previous work *Tkačik et al., 2015*, to infer the cell's position,

$$p(x|\theta) = \frac{p(\theta|x)p(x)}{p(\theta)} \tag{4}$$

where $p(\theta) = \int_0^1 dx\, p(\theta|x)\,p(x)$ and $p(x)$ is the prior distribution which we take to be uniform over a tissue of unit length, $p(x) = 1$. We quantify precision in the inference by the local inference error, $\sigma_X(x)$. For each position $x$, the inferred position $x^*$ of cells along the $y$-direction is taken to be the *maximum a posteriori* estimate,

$$x^*(x, y) = \underset{\tilde{x}}{\mathrm{argmax}}\, p(\tilde{x}|\theta(x, y)) \tag{5}$$

where we use $\tilde{x}$ to differentiate from the true position $x$. From this, the local and average inference error can be computed.

$$\sigma_X^2(x) = \langle(x^* - x)^2\rangle_y \tag{6}$$

$$\bar{\sigma}_X = \int_0^1 \sigma_X(x)\,p(x)\,dx \tag{7}$$

where the average in *Equation 6* is over cells in the $y$-direction. The logic behind this definition of the inference error is that development of the tissue relies on the precision in the inference of cells' positions *throughout the tissue*. However, there may be tissue developmental contexts, where only the positions of certain *regions* or *cell fate boundaries* need to be specified with any precision (as in the case of short-range morphogen gradients like Nodal *Liu et al., 2022*). The definition of inference error may be readily extended to incorporate such specifications (see Section 'Choice of objective function').

We have been motivated to use the maximum a posteriori (MAP) estimate in *Equation 5* by its successful use in previous studies in *Drosophila* embryo *Dubuis et al., 2013*; *Petkova et al., 2019*; *Tkačik et al., 2015* and, more importantly, that it is a local estimate not requiring the computation of $p(\theta)$ (which is independent of $x$). We have checked that a different definition of the inference error, which does not use the MAP estimate and takes into account the entire distribution $p(x^*|x)$,

$$\sigma_x^2(x) = \int_0^1 dx(x^* - x)^2 p(x^*|x)$$

leads to the same qualitative results.

**Table 1.** Parameters associated with rates, feedback and receptor profiles along with their range of values.

The chemical rate values used in numerical analysis are scaled by the unbinding rate $r_u, \kappa_u$ taken to be 1. The corresponding experimental values have been taken from *Lauffenburger and Linderman, 1996* where available.

| Parameter | Symbol | Numerical values | Experimental values |
| --- | --- | --- | --- |
| Chemical rates | | | |
| *Signalling branch* | | | |
| Unbinding rate | $r_u$ | 1 | $0.34\,\mathrm{min}^{-1}$ |
| Binding rate | $r_b$ | $0.1–1\,\mathrm{nM}^{-1}$ | $0.072\,\mathrm{nM}^{-1}\mathrm{min}^{-1}$ |
| Degradation rate | $r_d$ | 0.001–0.01 | $0.0022\,\mathrm{min}^{-1}$ |
| Internalisation rate | $r_I$ | 0.1–1 | $0.03–0.3\,\mathrm{min}^{-1}$ |
| Recycling rate | $r_R$ | 0.1–1 | $0.058\,\mathrm{min}^{-1}$ |
| *Non-signalling branch* | | | |
| Unbinding rate | $\kappa_u$ | 1 | - |
| Binding rate | $\kappa_b$ | $0.1–1\,\mathrm{nM}^{-1}$ | - |
| Degradation rate | $\kappa_d$ | 0.001–0.01 | - |
| Internalisation rate | $\kappa_I$ | 0.1–1 | - |
| Recycling rate | $\kappa_R$ | 0.1–1 | - |
| Conjugation rate | $\kappa_C$ | $0.1–1\,\mathrm{nM}^{-1}$ | - |
| Splitting rate | $\kappa_S$ | 0.1–1 | - |
| Feedback control | | | |
| Amplification | $\alpha$ | $0.1-10$ | - |
| Feedback Sensitivity | $\gamma$ | $0-1$ $\mathrm{nM}^{-1}$ | - |
| Feedback strength | n | $0-5$ | - |
| Receptor control | | | |
| *Signalling receptors* | | | |
| Hill coefficient | $a$ | $0-5$ | |
| Minimum concentration | $A_0$ | $50-250\,\mathrm{nM}$ | - |
| Maximum concentration | $A_0 + A_1$ | $50-500\,\mathrm{nM}$ | - |
| Position of half-maximum | $A_2$ | $0.01-1$ | - |
| *Non-signalling receptors* | | | |
| Hill coefficient | $b$ | $0-5$ | |
| Minimum concentration | $B_0$ | $50-250\,\mathrm{nM}$ | - |
| Maximum concentration | $B_0 + B_1$ | $50-500\,\mathrm{nM}$ | - |
| Position of half-maximum | $B_2$ | $0.01-1$ | - |
| Ligand input | | | |
| Maximum concentration | $A$ | $30\,\mathrm{nM}$ | - |
| Decay length | $\lambda$ | $0.2-0.5$ | - |
| Number of cells along $x$-direction | $n_x$ | 101 | - |

*Table 1 continued on next page*

*Table 1 continued*

| Parameter | Symbol | Numerical values | Experimental values |
|---|---|---|---|
| Number of cells along $y$-direction | $n_y$ | 101 | - |

## Quantitative models for cellular reading and processing

In order to calculate the probability of the inferred position given the output $p(x^*|\theta)$ and hence the inference error $\bar{\sigma}_X$, one needs to know the prior $p(x)$ and the input-output relation giving rise to the output distribution $p(\theta|x)$ in *Equation 4*. While a uniform prior may be justified by a homogeneous distribution of cells in the developing tissue at the stage considered, the input-output relation needs to be developed using a specific model based on the general channel design principles described previously. Thus, we will take each cell to be equipped with a chemical reaction network (CRN) that has up to two receptor types both of which bind the ligand on the cell surface but only one is signalling competent *Hemalatha et al., 2016*; *Tabata and Takei, 2004*. This latter aspect breaks the symmetry between the receptor types and hence the branches, a point that we will revisit in Section 'Asymmetry in branched architecture: promiscuity of non-signalling receptors'. In multi-tier architectures, the bound states of both the receptors are internalised and shuttled through several compartments. The last compartment allows for a conjugation reaction between the two receptors (as in the case of Wingless and Dpp *Hemalatha et al., 2016*; *Zhu et al., 2020*). The signalling states, defined by all the bound states of the signalling receptor, contribute to the output. Within this schema, we consider control mechanisms on the surface receptor concentrations and in the chemical reactions downstream to binding on the surface (i.e. on internalisation, shuttling, conjugation, etc). We formulate the control on processing steps as a feedback/feedforward regulation from one of the signalling species in the CRN. On the other hand, the control of surface receptors is considered in the form of an open-loop control by allowing receptor profiles to vary within certain bounds, as described below. The key parameters are *chemical rate parameters* describing the rates of various reactions in the CRN, *receptor parameters* describing the receptor concentration profiles, *feedback topology* in the CRN that is a combination of actuator and rate under regulation, *control parameters* describing the strength and sensitivity of the feedback/feedforward. With these parameters specified, an input-output relation, calculated as a tier-wise weighted sum of all signalling states, can then be used to infer the cell's position by *Equation 4*.

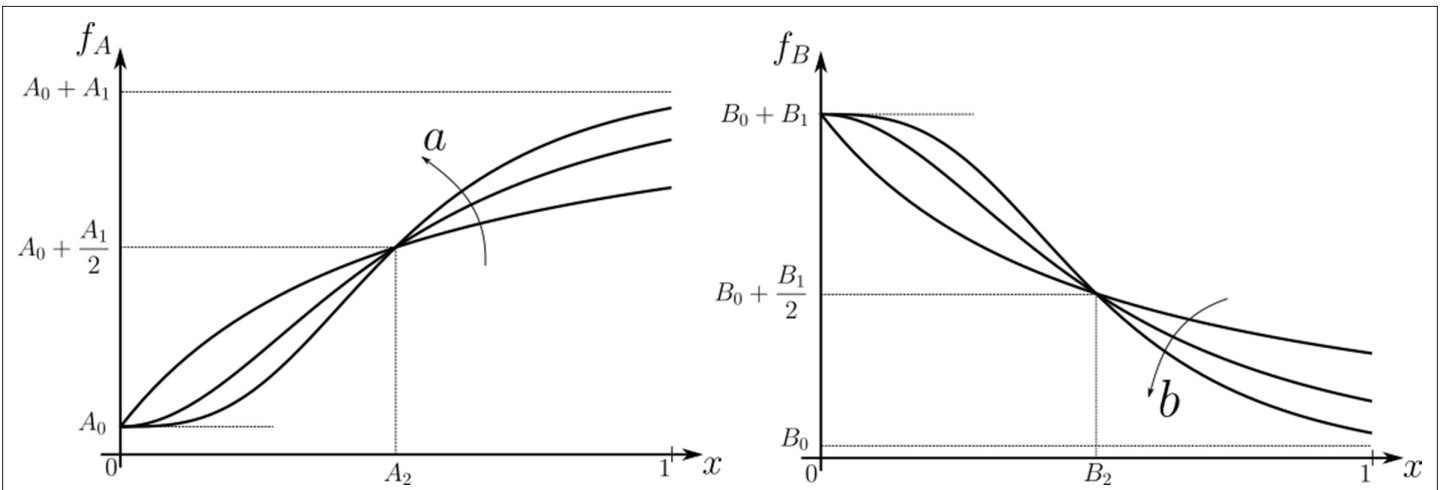

**Figure 3.** Family of receptor profiles $f_A$ (monotonically increasing in $x$) and $f_B$ (monotonically decreasing in $x$) with an interpretation of function parameters (*Equations 8; 9*). The total surface concentrations of both signalling and non-signalling receptors are taken from these families of receptor profiles.

## Cellular Reading via surface receptors

In the framework described previously, we consider the morphogen ligand as an *external input* to the receiving cells, outside the cellular information processing channel. The signal and noise of this external input are captured by the distribution **Equation 1**. This implicitly assumes that there is no feedback control from the output to the ligand input, that is no 'sculpting' of the morphogen ligand profile. We revisit this point in the Discussion. Given a distribution of the morphogen input, we address the *local, cell autonomous* morphogenetic decoding that allows the cells to tune their reading dynamically.

We subject the *local, cellular* reading to an open-loop control on total (ligand bound plus unbound) surface availability of the signalling $\psi$ and non-signalling $\phi$ receptors. This implies that for each evaluation of inference error within the optimisation routine (see Section 'Performance of the Channel Architectures'), the local surface receptor levels are held constant in time through a chemostat (see Appendix 1). In our analysis, we consider a family of monotonic (increasing or decreasing in $x$ and independent of $y$) receptor profiles, which for convenience we take to be of the Hill form (**Figure 3**), that is either

$$\text{Monotonically increasing in } x : f_A(x) = A_0 + \frac{A_1 x^a}{A_2^a + x^a} \quad \text{or} \tag{8}$$

$$\text{Monotonically decreasing in } x : f_B(x) = B_0 + \frac{B_1}{1 + (x/B_2)^b} \tag{9}$$

The range of values for these parameters considered in the numerical analysis are listed in **Table 1**. Therefore, when considering $\psi(x)$ to be monotonically increasing in $x$, we parametrise it with $f_A$. It follows that in a one-branch channel, there are two possibilities: $\psi \in \{f_A, f_B\}$ while in a two-branch channel, there are a total of four possibilities: $(\psi, \phi) \in \{f_A, f_B\} \times \{f_A, f_B\}$. This allows us to simulate the 'reading' step performed by the cells (see **Figure 1b**).

Note that we are not fixing a receptor profile but taking it from a class of monotonic profiles (including a uniform profile), over which we vary to determine the optimal inference (see Section

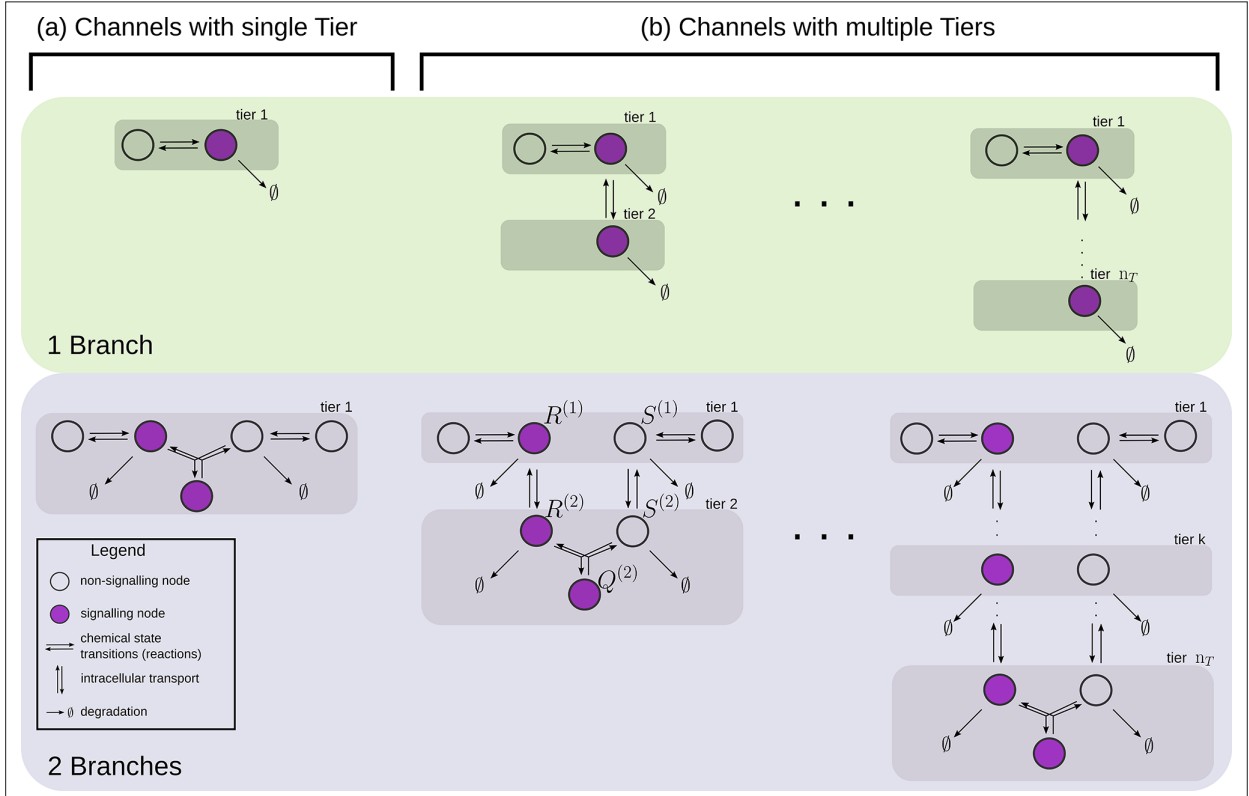

**Figure 4.** Examples of channel architectures with single and multiple tiers, and upto two branches. Signalling receptors in the bound state (colour purple) from each of the tiers contribute to the cellular output. The interpretation of the arrows is shown in the legend.

'Performance of the Channel Architectures' below). Further, in the optimisation scheme (Section 'Performance of the Channel Architectures'), we allow the receptor concentrations to vary over the space of all monotonically increasing, decreasing or flat profiles, and do not encode the positional information in the receptor profiles. Monotonicity implicitly assumes a spatial correlation in the receptor concentrations across cells – we return to this point in the Discussion.

## Dynamics of processing in a single-tier channel

In a single tier channel, all processing is restricted to the cell surface. We represent the bound state of the signalling receptor as $R^{(1)}$ and that of the non-signalling receptor as $S^{(1)}$. The conjugated state is represented by $Q^{(1)}$. The CRN for such a system with one and two branches is shown in *Figure 4a*. Rates associated with these reactions are listed in *Table 1*. The differential equations that describe the binding, unbinding, conjugation, splitting and degradation reactions of the receptors are given by

$$\partial_t R^{(1)} = r_b L(\psi(x) - R^{(1)}) - (r_u + r_d)R^{(1)} - \kappa_C R^{(1)}S^{(1)} + \kappa_S Q^{(1)} \tag{10}$$

$$\partial_t S^{(1)} = \kappa_b L(\phi(x) - S^{(1)}) - (\kappa_u + \kappa_d)S^{(1)} - \kappa_C R^{(1)}S^{(1)} + \kappa_S Q^{(1)} \tag{11}$$

$$\partial_t Q^{(1)} = \kappa_C R^{(1)}S^{(1)} - \kappa_S Q^{(1)} \tag{12}$$

The steady-state output $\theta$, defined as the sum of all the ligand-bound signalling states, is given by $\theta = R^{(1)} + Q^{(1)}$. Note that to describe the 1-branch system, we simply set all rates $\kappa$ to zero.

## Dynamics of processing in a multi-tier channel

In a multi-tiered channel, the receptors go through additional steps of processing before generating an output. We represent the bound state of a receptor in $k$-th tier of the first branch as $R^{(k)}$, that of the second branch as $S^{(k)}$, and the conjugate species that forms in the last $n_T$-th tier as $Q^{(n_T)}$. The CRN for such a system with $n_T$ tiers is shown in *Figure 4b*. Rates associated with these reactions are listed in *Table 1*. The differential equations that describe the binding, unbinding, trafficking, recycling, conjugation, splitting and degradation reactions of the receptors are given by

$$\partial_t R^{(1)} = r_b L(\psi(x) - R^{(1)}) - (r_u + r_d + r_I)R^{(1)} + r_R R^{(2)} \tag{13}$$

$$\partial_t S^{(1)} = \kappa_b L(\phi(x) - S^{(1)}) - (\kappa_u + \kappa_d + \kappa_I)S^{(1)} + \kappa_R S^{(2)} \tag{14}$$

$$\partial_t R^{(n_T)} = r_I R^{(n_T-1)} - (r_R + r_d)R^{(n_T)} - \kappa_C R^{(n_T)}S^{(n_T)} + \kappa_S Q^{(n_T)} \tag{15}$$

$$\partial_t S^{(n_T)} = \kappa_I S^{(n_T-1)} - (\kappa_R + \kappa_d)S^{(n_T)} - \kappa_C R^{(n_T)}S^{(n_T)} + \kappa_S Q^{(n_T)} \tag{16}$$

$$\partial_t Q^{(n_T)} = \kappa_C R^{(n_T)}S^{(n_T)} - \kappa_S Q^{(n_T)} \tag{17}$$

The output, realised from all the ligand-bound signalling states, now becomes $\theta = w_{n_T}Q^{(n_T)} + \sum_{k=1}^{n_T} w_k R^{(k)}$ at steady state with $w_k$, such that $\sum_k w_k = 1$, representing the weight allotted to the tier (according to the mean residence time in the tier, for instance). For details regarding the setup of *Equations 10–17* refer to Appendix 1. These differential equations for single-tiered and multi-tiered systems are to be augmented by stochastic contributions from both extrinsic and intrinsic sources. Extrinsic noise is a consequence of stochasticity of the ligand concentration presented to the cell, $L \sim p(L|x)$, and enters the equations as a source term. On the other hand, intrinsic noise is a consequence of copy-number fluctuations in the CRNs that characterise the channel, and are treated using chemical master equations (CMEs) *Sengupta, 2008*.

## Feedback Control

We consider all rates in the CRN, except the ligand binding and unbinding rates, as potentially under feedback regulation. Any chemical rate $r \in \{r_I, \kappa_I, \kappa_C, ....\}$ that is under feedback control actuated by the node $R \in \{R^{(1)}, S^{(1)}, ....\}$ is modelled as.

$$r_+ = r_0 \left(1 + \frac{\alpha R^n}{\gamma^{-n} + R^n}\right) \quad \text{if under positive feedback} \tag{18}$$

$$r_- = \frac{r_0}{1 + (\gamma R)^n} \quad \text{if under negative feedback} \tag{19}$$

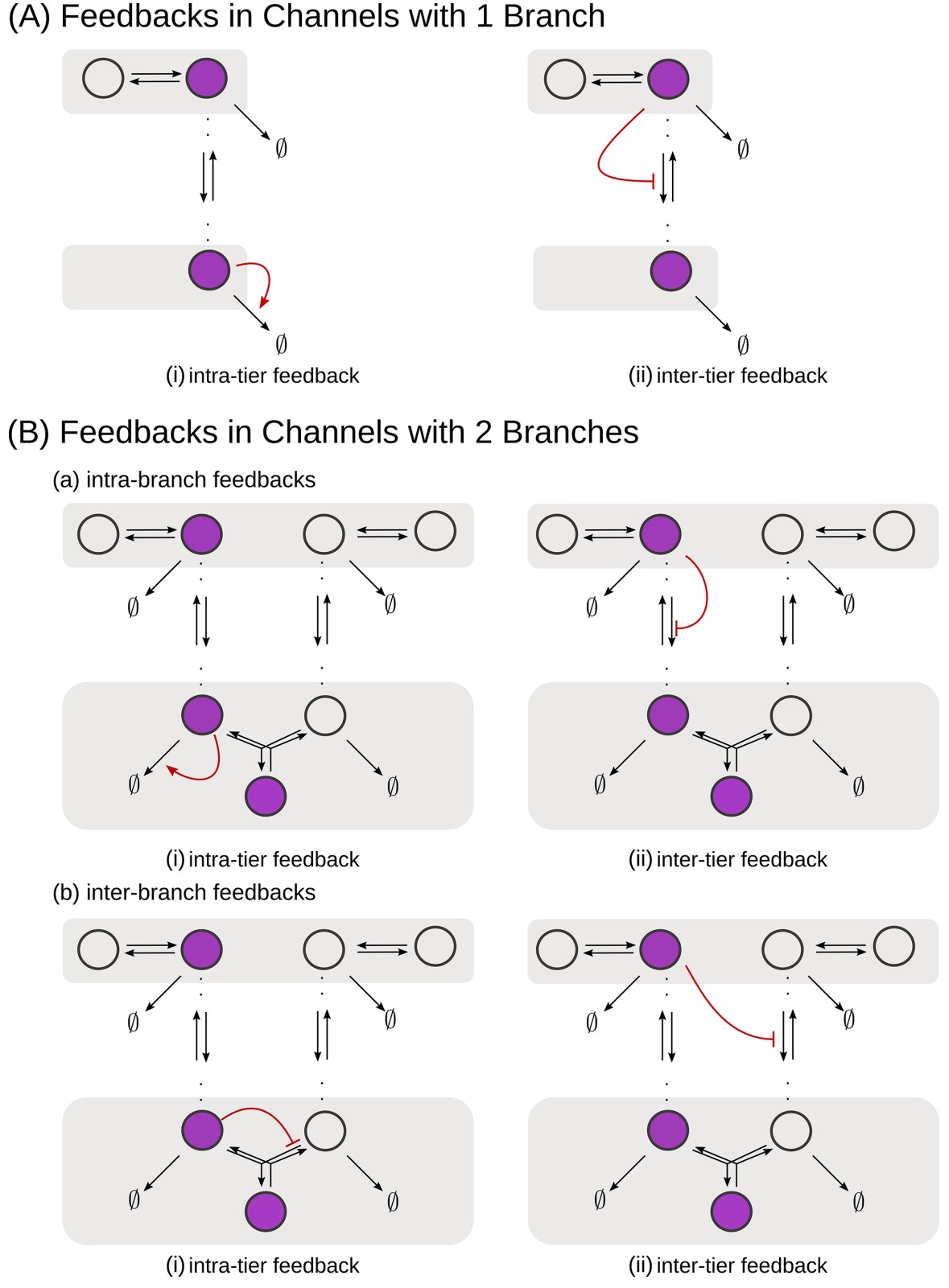

**Figure 5.** Schematic of feedback types. (**A**) In a one-branch channel, feedbacks are considered on internalisation rates or degradation rates. (**B**) A second branch in the channel opens up the possibilities of (**a**) intra-branch and (**b**) inter-branch, (**i**) intra-tier and (**ii**) inter-tier feedbacks.

with $r_0$ as the reference value of the chemical rate in the absence of feedback. The range of values for amplification $\alpha$, feedback sensitivity $\gamma$ and feedback strength $n$ are listed in **Table 1**. **Figure 5** shows the different categories of possible feedback controls. We discuss the heuristics underlying the feedback controls in Appendix 2.

## Performance of the channel architectures

With the model in place, we address the Objective discussed previously, by studying the performance of different channel architectures, i.e. number of tiers and branches, and feedback topology. We define a vector $\vec{v}$ belonging to a parameter space $\mathcal{V}$ of the channel parameters related to chemical rates, receptor profiles and feedback (see **Table 1**). While the chemical rates and feedback parameters are the same in all cells, the receptor profile parameters help define the receptor concentrations at each cell position $x, y$. For a given morphogen input distribution $p(L|x)$ and a channel architecture under consideration, the optimisation can be stated as

$$\bar{\sigma}_X^{\text{opt}} = \min_{\vec{v} \in \mathcal{V}} \bar{\sigma}_X(\vec{v}; p(L|x)) \tag{20}$$

and implemented by the following algorithm, the details of which are presented in Appendix 3.

---

Optimisation scheme

---

1. Fix a morphogen input distribution for each position, $p(L|x)$ using **Equation 1**.
2. Define the channel architecture hierarchically, i.e. first declare the number of tiers and branches in the channel, and then choose a feedback topology (as in **Figure 5**).
3. Optimise the average inference error **Equation 20** w.r.t. to the channel parameters $\vec{v} \in \mathcal{V}$ within the bounds provided in **Table 1**. We use a gradient independent method viz. Pattern Search algorithm for this step (implemented in MATLAB). For every poll (iteration) of the Pattern Search, we evaluate the average inference error $\bar{\sigma}_X$ using the steady-state outputs of the equations corresponding to the CRN under optimisation that is **Equations 10–17**. The steady state solution is obtained analytically when possible or solved using ODE15s (MATLAB) algorithm.
4. Repeat Step 3 until all feedback topologies under consideration are exhausted.
5. Repeat Steps 2 and 3 until all channel architectures are scanned.

---

## Results

As discussed previously, cells of a developing tissue face both extrinsic as well as intrinsic sources of noise. We first look at the issue of extrinsic noise in the morphogen input (described by **Equation 1**). The output then is a deterministic function of the morphogen input and parameters of the channel i.e. receptor concentrations, feedback topology, chemical rates and feedback parameters. The range of values considered for these parameters is listed in **Table 1**, consistent with the timescale separation between the rates of chemical reactions and transport as discussed in Section 'Conceptual framework and quantitative models'. We apply the numerical analysis and the optimisation algorithm outlined in Section 'Performance of the Channel Architectures' to determine the design characteristics of 'reading' (receptor profiles) and 'processing' (tiers and feedback control) steps. Later, we check how channels, optimised in the reading and processing steps to deal with extrinsic noise, respond to intrinsic noise and what roles the elements of channel architecture play there. All the essential results are presented in this section and the reader may look up the appendices for further details.

## Branched architecture with multiple receptors provides accuracy and robustness to extrinsic noise

We begin with architectures comprising single-tiered channels with one and two branches. Such architectures are similar in design to the classic picture of ligand-receptor kinetics *Lauffenburger and Linderman, 1996*; *Alberts et al., 2017*, but also to the self-enhanced degradation models for robustness of morphogen gradients *Eldar et al., 2003*. Before we proceed, it helps to recall a simple heuristic regarding signal discrimination. *Appendix 4—figure 1* illustrates that precision in positional inference requires both that the output variance at a given position be small and that the mean output at two neighbouring positions be sufficiently different.

Let us first consider a *minimal* architecture of a one-tier one-branch channel without feedback control on any of the reaction rates. The output of this channel, here $R^{(1)}$, is a monotonic, saturating function of the input, with the surface receptor concentration setting the asymptote. As in *Appendix 5—figure 1a*, if the receptor concentrations decrease with mean ligand input, i.e. increases with distance from source ($f_A$ in *Figure 3a*; ), the outputs for different input ranges overlap significantly. On the other hand, if the receptor concentrations increase with mean input ($f_B$ in *Figure 3b*), the outputs overlap to a lesser degree (see *Appendix 5—figure 1b*). Thus within this minimal architecture, the inference error is optimised when the receptor concentrations increase with the mean input.

Introducing a feedback in this one-tier one-branch architecture, either on receptor levels or degradation rate, only partially reduces the inference errors (*Figure 6a, c*). As seen in *Figure 6d*, this is because the surface receptor concentration $\psi$ sets both the asymptote and the steepness of the input-output functions, resulting in significant overlaps between outputs at neighbouring positions. The receptor control introduces a competition between *robustness* of the output to input noise and sensitivity to systematic changes in the mean input (see Appendix 4).

Including a non-signalling receptor $\phi$ via an additional *branch* in the channel architecture opens up several new possibilities of feedback controls, in addition to providing an extra tuning variable. Now, as opposed to the one-tier one-branch case, an inter-branch feedback control (*Figure 7a*) results in an input-output relation with a sharp rise followed by a saturation (*Figure 7d*). By appropriately placing the receptors at spatial locations that receive different input, as shown by black arrow in *Figure 7d*, one can cleanly separate out the cellular outputs in neighbouring positions. For a detailed description see Appendix 6. This mitigates the above-mentioned tension between *robustness* to input noise and *sensitivity* to systematic changes in the mean input to a considerable extent (see *Appendix 4—figure 2*).

As seen in *Figure 7c*, the two-branch architecture with inter-branch feedback leads to a dramatic reduction in the inference errors, to reach one cell's width precision at most spatial locations in the tissue.

We would like to highlight two unexpected features of the optimised two-branch architecture. (i) The signalling and non-signalling receptors present opposing optimal profiles – a consequence of the negative inter-branch feedback. (ii) The optimal non-signalling receptor decreases away from the source, indicating that the non-signalling receptor 'reads' the ligand input, while the signalling receptor increases away from the source, buffering the noise in the output (*Figure 7*). A heuristic understanding of the opposing optimal receptor profiles is provided in Appendix 7. In contrast, in the one-branch architectures, it is the signalling receptor that does the reading and buffering.

## Tiered architecture with compartmentalisation adds robustness to intrinsic noise

We next investigate the effects of addition of tiers (compartments) on the inference errors. Our optimisation shows there are two distinct optimised two-tier two-branch architectures, one with inter-branch feedback on the internalisation rate of the non-signalling receptors $\kappa_I$ and the other on the conjugation rate $\kappa_C$, that have comparable inference errors (*Figure 8b, c*). Both the receptor profiles and the input-output relations of these two optimised two-tier two-branch channels are qualitatively similar (*Appendix 8—figure 1*).

It would seem that addition of further tiers, that is more than two, would lead to further improvement in the inference. However, in both these optimised architectures, addition of tiers leads only to a marginal reduction of inference errors (*Figure 8a*) while invoking a cellular cost. Of course, extensions of our model that involve modification of the desired output could favour the addition of more tiers. For instance, additional tiers could facilitate signal amplification or improvement in *robustness* to input noise through an increase in signal-to-noise ratio (SNR) *Stoeger et al., 2016*. Further, by making the output $\theta$ a multi-variate function of the tier index (compartment identity) one can multitask the various cellular outcomes (as in Ras/MAPK signalling *Fehrenbacher et al., 2009* or with GPCR compartmentalisation *Ellisdon and Halls, 2016*).

So far, we have only considered noise due to fluctuations in the morphogen profile, that is extrinsic noise. Given that we are considering a distributed channel, intrinsic noise due to low copy numbers

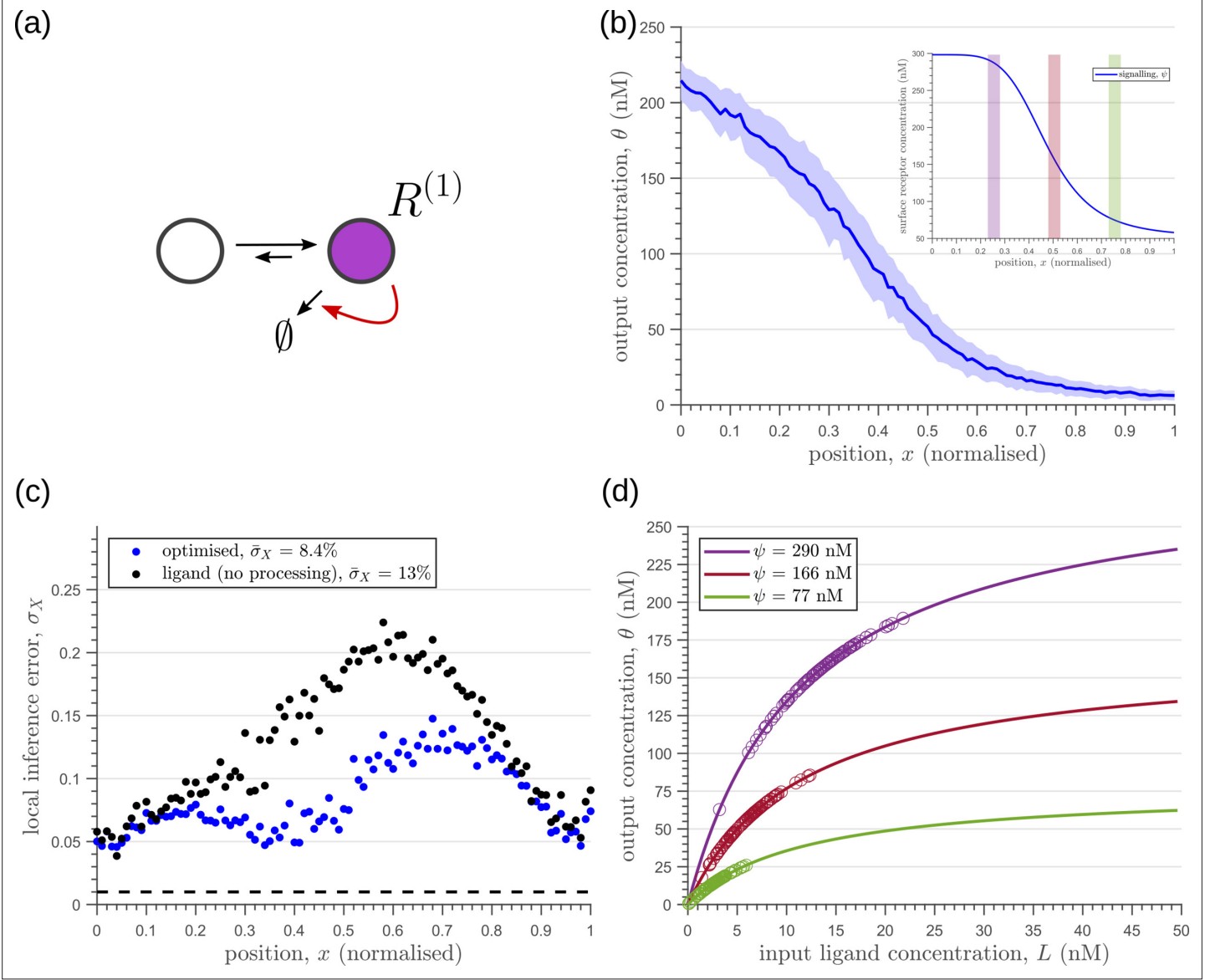

**Figure 6.** Characteristics of an optimised (**a**) one-tier one-branch channel with only the signalling receptor and feedback. The optimised channel shows a moderately strong positive feedback on the degradation rate. (**b**) The optimal output is obtained when (b, inset) the total (bound plus unbound) signalling receptor concentration profile decreases away from the source. (**c**) Local inference errors in this optimised channel show a reduction compared to the expected inference errors from ligand with no cellular processing (i.e. reading directly from the free ligand). The minimum average inference error in this channel is $\bar{\sigma}_X \sim 8\%$, which corresponds to 8 cells' width. The dashed line denotes a local inference error of one cell's width $\sim 1/n_x$. (**d**) The input-output relations in this channel are monotonically increasing sigmoid functions saturating at only large values of input. The solid lines correspond to the input-output relations at selected positions $x = 0.25, 0.5, 0.75$, shaded with the same colour as the position-markers in (b inset, coloured rectangles). The signalling $\psi(x)$ receptor concentration is mentioned in the legend. For a fixed distribution of ligand input (**Equation 1**), the range of input values recorded by the receptors at the selected positions gives rise to a range of outputs (circles). It is clear that neighbouring positions have significant overlaps in their outputs. The optimised parameter values for the plots in (**b–d**) can be found in **Table 2** under the column corresponding to $n_T = 1, n_B = 1, r_+ = r_d^{(1)}$.

of the reacting species in the CRN will have a significant influence on the inference. As discussed in Section 'Conceptual framework and quantitative models' and Appendix 3, we solve the stochastic chemical master equations (CMEs) to compute the output distributions and the positional inference. It is here that we find that the addition of tiers contribute significantly to reducing inference errors. A comparison of the one-tier two-branch and two-tier two-branch channel architectures (**Figure 9a and b**) optimised for extrinsic noise, shows that in the presence of intrinsic noise, additional tiers

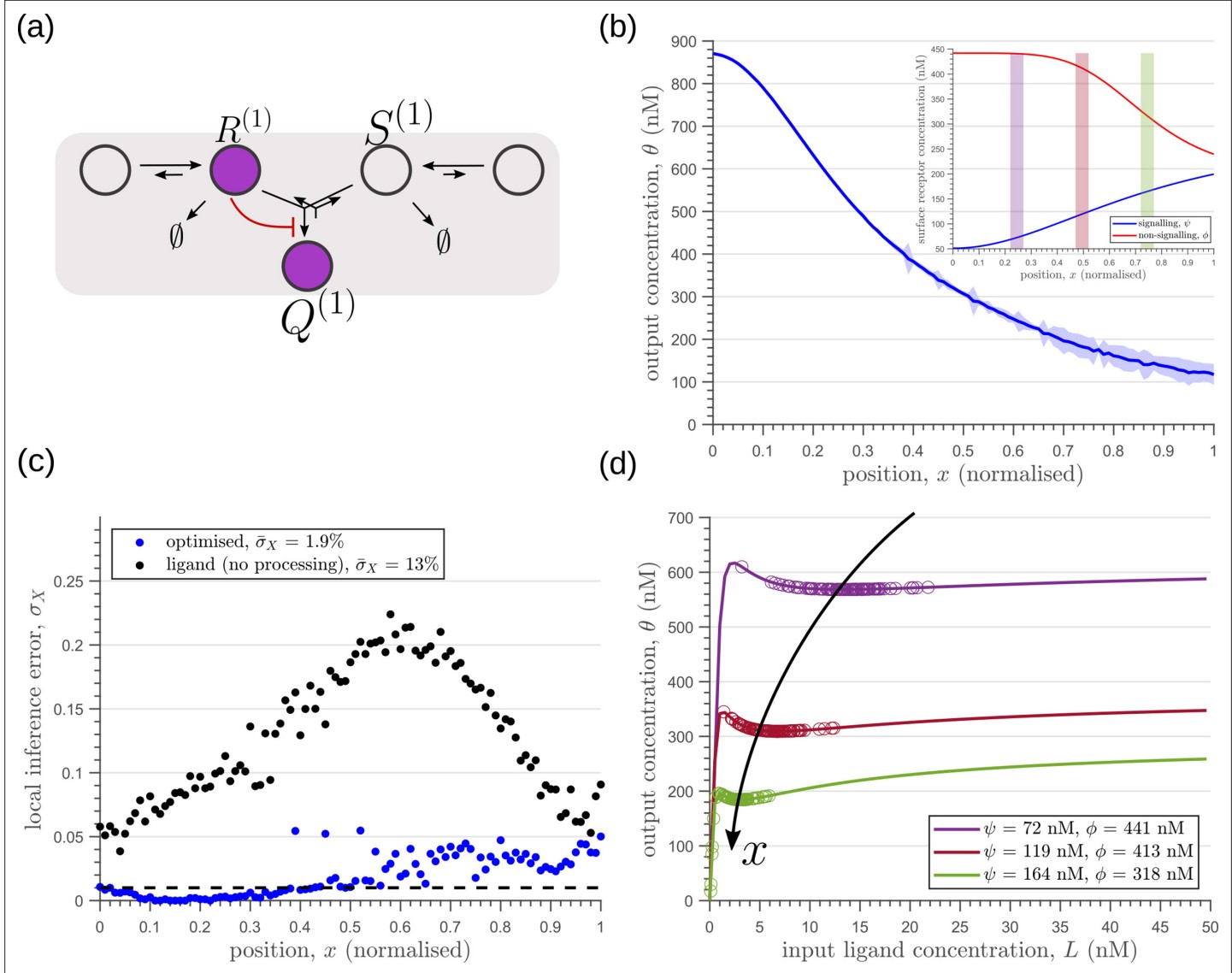

**Figure 7.** Results of optimisation of (**a**) one-tier two-branch channel. (**b**) The output profile (with standard error in shaded region) corresponding to the (inset) optimised signalling (blue) and non-signalling (red) receptor profiles. The optimal signalling receptor now increases away from the source as opposed to the situation in the optimal one-tier one-branch channel (*Figure 6*). On the other hand, the optimal non-signalling receptor decreases away from the source. (**c**) The local inference error $\sigma_X(x)$ is reduced throughout the tissue, when compared to the expected inference errors from ligand with no processing. (**d**) The input-output relations at selected positions $x = 0.25, 0.5, 0.75$ (in the direction of the black arrow) are shown as solid lines, shaded with the same colour as the position-markers in (b inset, coloured rectangles). The signalling $\psi(x)$ and non-signalling $\phi$ receptor concentrations are mentioned in the legend. For a fixed distribution of ligand input (*Equation 1*), the range of input values recorded by the receptors at the selected positions gives rise to a range of outputs (circles). Tuning of input-output relations through receptor concentrations reduces output variance and minimises overlaps in the outputs of neighbouring cell cohorts. The optimised parameter values for the plots in (**b–d**) can be found in *Table 2* under the column corresponding to $n_T = 1, n_B = 2, r_- = \kappa_C$.

lead to significantly lower inference errors (*Figure 9c*). The large inference errors seen in the one-tier one-branch channel in the presence of intrinsic noise, can be traced to the instabilities of steady-state trajectories of the two signalling species $R^{(1)}$ and $Q^{(1)}$ driven by the non-linear feedback (*Figure 9d–f*). This effect is more prominent for larger values of ligand concentrations, that is closer to the source at $x = 0$. On the other hand, we find that in the two-tier two-branch architecture (*Figure 9g–i*), the fluctuations in the signalling species are more tempered, the inter-branch feedback leads to a mutual damping of the fluctuations of the signalling species from the two branches. Details of this heuristic argument appear in Appendix 9.

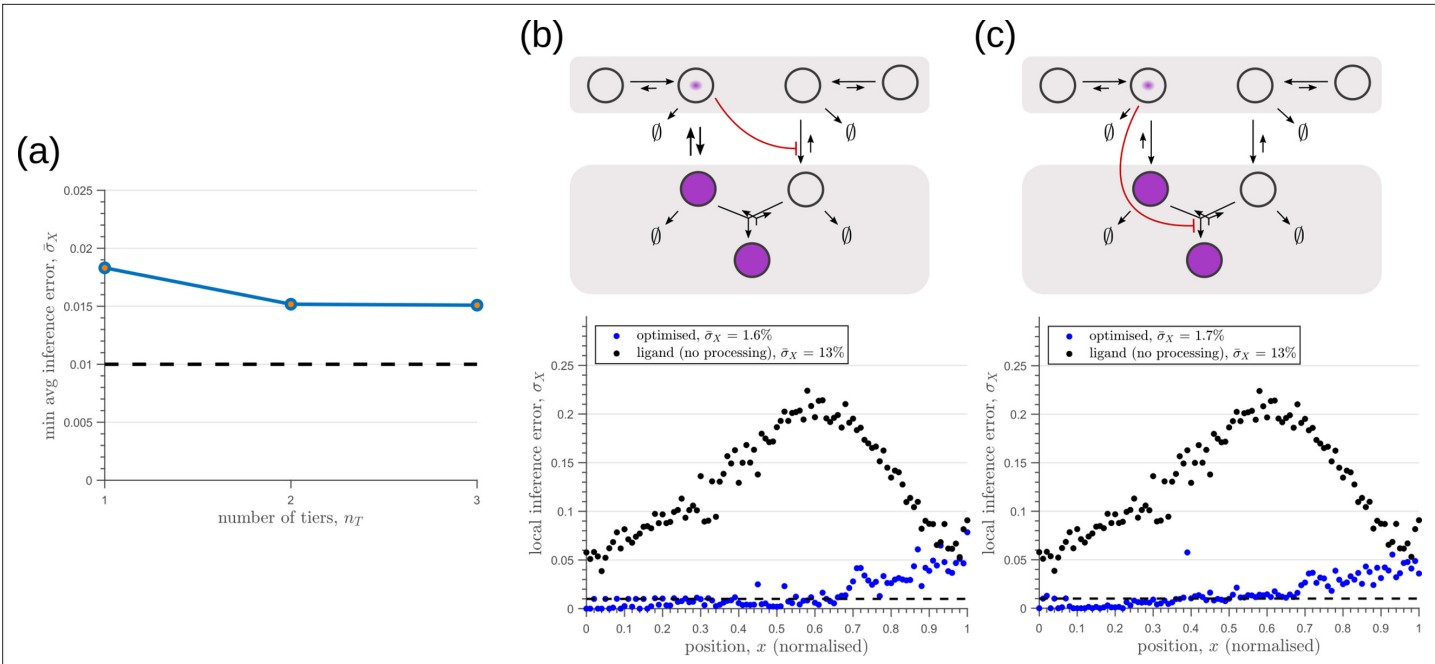

**Figure 8.** Performance of the optimised two-branch channels with increasing numbers of tiers. (**a**) Minimum average inference error $\bar{\sigma}_X$ in two-branch architectures with increasing number of tiers $n_T$. The dashed line corresponds to a local inference error of one cell's width $\sim 1/n_x$. (**b,c**) Results of optimisation of two-tier two-branch channels with inter-branch feedback. These two architectures perform equally well: local inference errors in both the channels (blue dots) are low throughout the tissue (with average inference errors $\sim 1.6\%$ and $\sim 1.7\%$) as compared to a case with no processing of ligand prior to inference (black dots). Note that the local inference errors in the optimised channels increase towards the end of the tissue due to lower ligand concentrations. The dashed line corresponds to a local inference error of one cell's width $\sim 1/n_x$. The optimised parameter values for the plots in (**b–c**) can be found in *Table 2* under the column corresponding to $n_T = 2, n_B = 2, r_- = \kappa_I$ and $n_T = 2, n_B = 2, r_- = \kappa_C$, respectively.

In summary, we find that the nature of the channel architectures play a significant role in robustness of morphogenetic decoding to both extrinsic and intrinsic sources of noise. Of the three elements to the channel architecture - branches, tiers, and feedback control, we find that a branched architecture can significantly reduce inference errors by employing an inter-branch feedback and a control on its local receptor concentrations. For this, the receptor concentration profiles required to minimise inference errors are such that the concentration of signalling (non-signalling) receptor should decrease (increase) with mean morphogen input. Crucially, in the absence of feedback, performance of the channel diminishes and the optimised receptor profiles *both* decrease away from the source (*Appendix 10—figure 1*). Further, we show in *Appendix 11—figure 1* that having uniform profiles for the signalling and non-signalling receptors, with or without uncorrelated noise, fares poorly in terms of inference capability. This provides a posteriori justification for the monotonicity in receptor profiles. Addition of tiers can help in further bringing down inference errors due to extrinsic noise, but with diminishing returns. An additional tier, however, does provide a buffering role for feedback when dealing with intrinsic noise. We note that these qualitative conclusions remain unaltered for different morphogen input characteristics, that is input noise and morphogen decay lengths (see Appendix 12).

## Asymmetry in branched architecture: promiscuity of non-signalling receptors

Before comparing the theoretical results with experiments, we comment on the implications for the cellular control of the signalling $\psi$ and non-signalling $\phi$ receptors. In the two-branch architecture, the symmetry between the signalling and non-signalling receptors is broken by the inter-branch feedback and the definition of output $\theta$, the latter taken to be a function only of the signalling states $R^{(k)}$ and $Q^{(k)}$ (Section 'Conceptual framework and quantitative models', purple nodes in *Figure 7a* and *Figure 8b and c*). What are the phenotypic implications of this asymmetry? In *Appendix 13— figure 1*, we plot the contours of average inference errors $\bar{\sigma}_X$ in the $\psi - \phi$ plane around the optimal point. We compute the eigenvalues of the local curvature of $\bar{\sigma}_X(\Delta\psi, \Delta\phi)$ around the optimal point

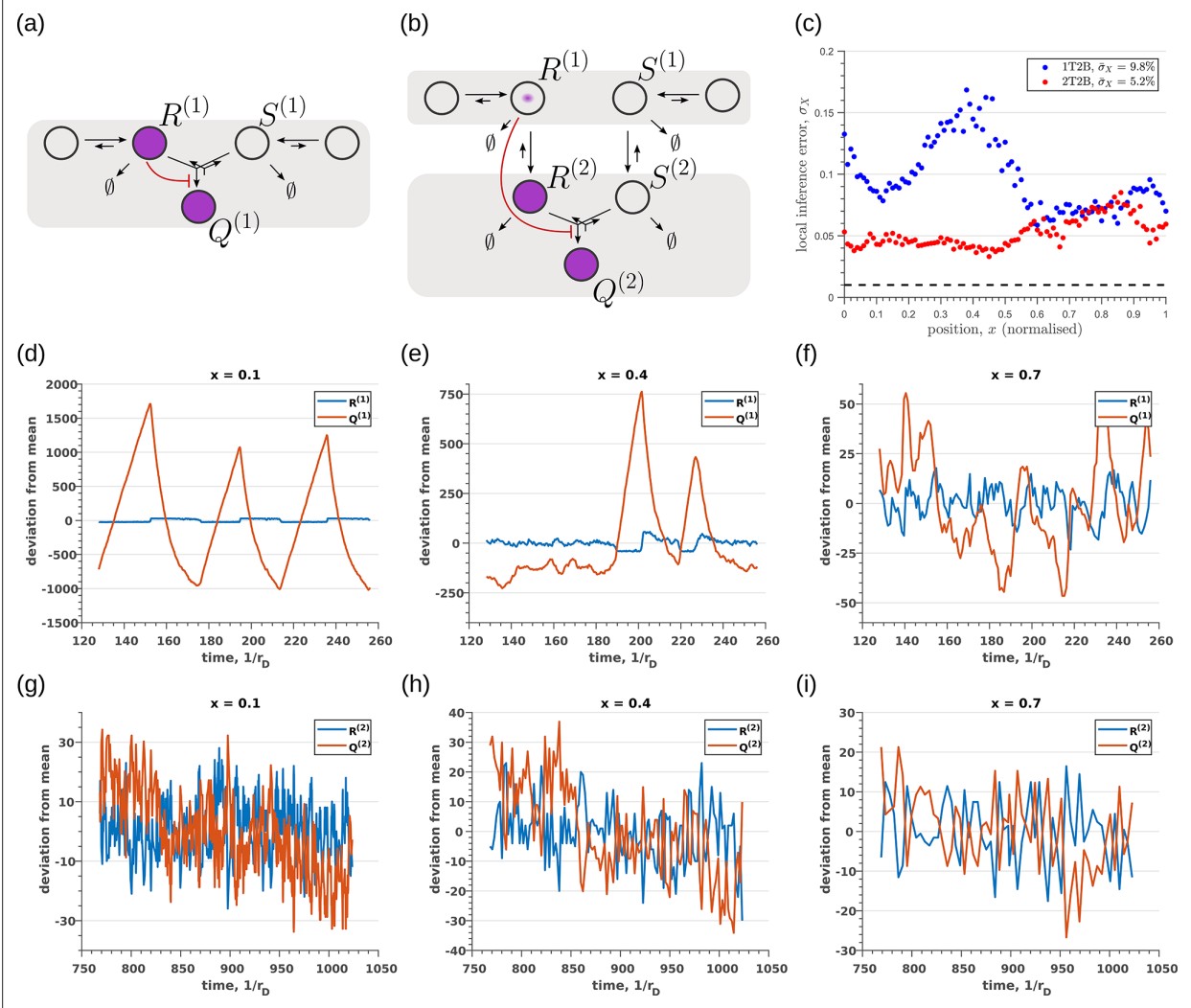

**Figure 9.** Robustness to intrinsic noise in (**a**) one-tier two-branch (1T2B) channel and (**b**) two-tier two-branch (2T2B) channel architectures, previously optimised for extrinsic noise alone. (**c**) A comparison of local inference errors due to intrinsic noise shows consistently better performance in the case of a two-tier two-branch channel (red dots). (**d-f**) Sample steady-state trajectories of the signalling species $R^{(1)}$ (blue) and $Q^{(1)}$ (red) of a one-tier two-branch channel (purple nodes in (**a**)) at positions $x = 0.1, 0.4, 0.7$, respectively. (**g–i**) Sample steady-state trajectories of the signalling species $R^{(2)}$ (blue) and $Q^{(2)}$ (red) of a two-tier two-branch channel (purple nodes in (**b**)) at positions $x = 0.1, 0.4, 0.7$, respectively. The optimised parameter values for the plots in (**c,d–f,g–i**) can be found in *Table 2* under the column corresponding to $n_T = 1, n_B = 2, r_- = \kappa_C$ and $n_T = 2, n_B = 2, r_- = \kappa_C$, respectively.

($\Delta\psi = \Delta\phi = 0$). The difference in the magnitudes of these eigenvalues, as discussed in Appendix 13, immediately describes stiff and sloppy directions *Transtrum et al., 2015* along the $\psi$ and $\phi$ axes, respectively. This implies that while the signalling receptor is under tight cellular control, the control on the non-signalling receptor is allowed to be sloppy. A similar feature is observed in the contour plots for the robustness measure $\chi$ (defined as the ratio of coefficients of variation in the output to that in the input). *Appendix 4—figure 3* shows that for any given input distribution, reduction in output variance requires a stricter control on $\psi$, while the control on $\phi$ can be lax.

This sloppiness in the levels of non-signalling receptor would manifest at a phenotypic level in the context of multiple morphogen inputs as in the case of *Drosophila* imaginal disc *Tabata and Takei, 2004*. Participation of the same non-signalling receptor in the different signalling networks would imply its promiscuous interactions with all ligands. The signalling receptors, therefore, are *specific* for the various ligands while the non-signalling receptor, being promiscuous, is *non-specific*. This, as we see below, is the case with the Heparan sulfate proteoglycans (HSPGs) such as Dally and Dally-like

protein (Dlp) that participate in the Wingless (Wg) and Decapentaplegic (Dpp) signalling networks *Lin and Perrimon, 2000*; *Romanova-Michaelides et al., 2021*.

## Geometry of fidelity landscape

The above section and Appendix 13 motivate us to study the changes in the inference error upon perturbations of all the channel parameters. We therefore discuss the nature of optima in terms of the local geometry of the *fidelity* landscape around the optimum, and the geometry of the low inference error states. We work with the case of the optimised one-tier two-branch channel (shown in *Figure 7a* with optimum channel parameters listed in *Table 2*, *Table 3*) in presence of extrinsic noise.

To address the geometry of the local fidelity landscape around the optimum, we compute (i) percent changes in inference error $\bar{\sigma}_X$ due to perturbations in channel parameters (*Figure 10a*), and (ii) the eigenspectrum of the Fisher information metric (FIM, *Figure 10b*). The FIM $g_{\mu\nu}$ is evaluated in the log-parameter space as *Transtrum et al., 2015*.

$$g_{\mu\nu} = \sum_{x_i} \sum_{y_j} \frac{\partial x^*(M(x_i, y_j), \vec{v})}{\partial \ln v^\mu} \frac{\partial x^*(M(x_i, y_j), \vec{v})}{\partial \ln v^\nu} \tag{21}$$

where, $\vec{v} \in \mathcal{V}$ is the channel parameter vector, and $x_i, y_j$ are the indices of cells that run along the $x$- and $y$-directions. As shown in *Figure 10a*, we see that the inference error does not change significantly (up to 20% change with most parameters), that is it remains within $\bar{\sigma}_X \leq 2.2\%$. Varying the feedback strength $n$, however, drives a much stronger deviation from the minimum. Similarly, as seen from the heat map (*Figure 10b*), eigenvectors with the larger eigenvalues (index 1–6) have an appreciable component of the feedback parameters $\gamma, n$. This implies that variation of the feedback parameters from the optimum would result in significant changes in the inferred positions. Perturbing conjugation $\kappa_C$ and splitting $\kappa_S$ rates simultaneously (see eigenvector 16) does not produce any notable change to the inferred positions (eigenvalue $\sim 10^{-13}$). Further, perturbations to channel parameters other than the feedback parameters (eigenvectors 7–16) produce marginal changes in inferred positions.

Moving now from a local to global analysis of the fidelity landscape, we run the optimisation algorithm (Section 'Performance of the Channel Architectures') on the one-tier two-branch channel architecture with $2^{16}$ space-filling initial points in the 16-dimensional parameter space of this architecture. We then define the low inference error states as those channel parameters $\vec{v}^{\text{opt}}$ that yield $\bar{\sigma}_X \leq 2\%$. This cutoff, which equals $\lceil \frac{\bar{\sigma}_X}{0.01} \rceil$, corresponds to declaring as equivalent all the inference errors $\bar{\sigma}_X$ that lie between one and two cells' widths. Consistent with the local analyses, we find that the frequency distribution of optimal feedback parameters $\gamma, n$ is narrowly distributed about the global optimum (*Figure 11a*). As shown in *Figure 11a*, the parameters corresponding to forward and backward rates are skewed towards the upper and lower bounds of the allowed parameter range, respectively. We see that the optimal binding rates in the non-signalling branch (*Figure 11a*) are more broadly distributed across the permissible range than the optimal binding rates in the signalling branch, which are concentrated towards the upper bound of the permissible range. This again reflects the promiscuity of the non-signalling receptors as described in Section 'Asymmetry in branched architecture: promiscuity of non-signalling receptors'. All other optimal parameters corresponding to degradation rates, minimum and maximum receptor values and steepness of the receptor profiles, show a very broad spread over this range (*Appendix 14—figure 1*). To explore the topography of the low inference error landscape, we evaluate the components of the 'position vectors' of these minima $\vec{v}^{\text{opt}}$ in the parameter space $\mathcal{V}$ along the eigenvectors of the Hessian of $\bar{\sigma}_X$, defined as

$$h_{\mu\nu} = \frac{\partial^2 \bar{\sigma}_X(M, \vec{v})}{\partial v^\mu \partial v^\nu} \tag{22}$$

where $M$ stands for the entire morphogen profile and we have assumed a Euclidean metric. As shown in *Figure 11b and c*, components of the 'position vector' of the minima $\vec{v}^{\text{opt}} \in \mathcal{V}$ lie predominantly along the *sloppy* directions of the Hessian that is along the eigenvectors with small eigenvalues. This suggests that geometry of the low inference error landscape resembles a deep valley, which is shallow along the several *sloppy* directions and steep along the few *stiff* directions.

**Table 2.** Values of rates, feedback and receptor control parameters obtained after optimising the different channel architectures with $n_T$ tiers and $n_B$ branches.

The optimised values of the chemical rates quoted below are scaled by the unbinding rate $r_u$, $\kappa_u$ taken to be 1. The symbols $r_-$ and $r_+$ denote positive and negative feedbacks, respectively, on the rates following the equals sign; { } implies absence of feedback.

| Parameter (Symbol) | Value obtained in the optimised channel ($n_T, n_B$) | | | | |
|---|---|---|---|---|---|
| | (1,1) | (1,2) | (2,2) | (2,2) | (2,2) |
| | $r_+ = r_d^{(1)}$ | $r_- = \kappa_C$ | $r_- = \kappa_I$ | $r_- = \kappa_C$ | $r_- = \{\}$ |
| **Chemical rates** | | | | | |
| *Signalling branch* | | | | | |
| Binding rate ($r_b$, nM$^{-1}$) | 0.0898 | 0.0949 | 0.0932 | 0.0893 | 0.0787 |
| Degradation rate in tier 1 ($r_d^{(1)}$) | 0.0013 | 0.0081 | 0.0086 | 0.0098 | 0.0038 |
| Degradation rate in tier 2 ($r_d^{(2)}$) | - | - | 0.0066 | 0.0087 | 0.0016 |
| Internalisation rate ($r_I$) | - | - | 0.0531 | 0.0784 | 0.0363 |
| Recycling rate ($r_R$) | - | - | 0.0681 | 0.0359 | 0.0758 |
| *Non-signalling branch* | | | | | |
| Binding rate ($\kappa_b$, nM$^{-1}$) | - | 0.0590 | 0.0954 | 0.0835 | 0.0288 |
| Degradation rate in tier 1 ($\kappa_d^{(1)}$) | - | 0.0086 | 0.001 | 0.0043 | 0.0068 |
| Degradation rate in tier 2 ($\kappa_d^{(2)}$) | - | - | 0.0037 | 0.0031 | 0.0033 |
| Internalisation rate ($\kappa_I$) | - | - | 0.0741 | 0.0846 | 0.0559 |
| Recycling rate ($\kappa_R$) | - | - | 0.0123 | 0.0134 | 0.0998 |
| Conjugation rate ($\kappa_C$, nM$^{-1}$) | - | 0.9926 | 0.9823 | 0.9722 | 0.6019 |
| Splitting rate ($\kappa_S$) | - | 0.1285 | 0.1545 | 0.1350 | 0.7512 |
| **Feedback control** | | | | | |
| Amplification ($\alpha$) | 3.2085 | - | - | - | - |
| Feedback Sensitivity ($\gamma$) | 0.2491 | 0.1831 | 0.5535 | 0.8259 | - |
| Feedback strength ($n$) | 2.6825 | 2.3683 | 2.0953 | 2.1880 | - |
| **Tier-wise weights** | | | | | |
| weight of tier 1 ($w_1$) | 1 | 1 | 0.0018 | 0.1232 | 0.9259 |
| weight of tier 2 ($w_2$) | - | - | 0.9982 | 0.8768 | 0.0741 |
| **Receptor control** | | | | | |
| *Signalling receptors* | | | | | |
| Hill coefficient ($a$) | 4.9231 | 1.9974 | 3.8363 | 3.5251 | 3.3835 |
| Minimum concentration ($A_0$, nM) | 51.8130 | 51.0960 | 69.6940 | 51.9770 | 51.2 |
| Maximum concentration ($A_0 + A_1$, nM) | 298.283 | 290.356 | 304.114 | 134 | 301 |
| Position of half-maximum ($A_2$) | 0.4752 | 0.7818 | 0.9405 | 0.8344 | 0.4091 |
| *Non-signalling receptors* | | | | | |
| Hill coefficient ($b$) | - | 4.8951 | 1.0802 | 1.7472 | 3.1821 |
| Minimum concentration ($B_0$, nM) | - | 192.32 | 248.69 | 192.4 | 94.1850 |

*Table 2 continued on next page*

*Table 2 continued*

| Parameter (Symbol) | Value obtained in the optimised channel $(n_T, n_B)$ | | | | |
|---|---|---|---|---|---|
| | (1,1) | (1,2) | (2,2) | (2,2) | (2,2) |
| | $r_+ = r_d^{(1)}$ | $r_- = \kappa_C$ | $r_- = \kappa_I$ | $r_- = \kappa_C$ | $r_- = \{\}$ |
| Maximum concentration $(B_0 + B_1, \mathbf{nM})$ | - | 442 | 489.77 | 441.67 | 305 |
| Position of half-maximum ($B_2$) | - | 0.7428 | 0.5177 | 0.3196 | 0.0902 |

**Table 3.** Values of chemical rates and feedback parameters obtained after optimising the two-tier two-branch channel with inter-branch feedback on the internalisation rate $\kappa_I$ of the non-signalling branch, keeping the receptor profiles spatially uniform, with and without uncorrelated noise. The optimised values of the chemical rates quoted below are scaled by the unbinding rate $r_u$, $\kappa_u$ taken to be 1.

| Parameter (Symbol) | Optimised value | |
|---|---|---|
| | uniform receptor profiles | uniform receptor profiles with uncorrelated noise |
| Chemical rates | | |
| *Signalling branch* | | |
| Binding rate ($r_b, \mathbf{nM}^{-1}$) | 0.0922 | 0.0782 |
| Degradation rate in tier 1 ($r_d^{(1)}$) | 0.0089 | 0.0041 |
| Degradation rate in tier 2 ($r_d^{(2)}$) | 0.0092 | 0.0095 |
| Internalisation rate ($r_I$) | 0.0225 | 0.0611 |
| Recycling rate ($r_R$) | 0.0403 | 0.0971 |
| *Non-signalling branch* | | |
| Binding rate ($\kappa_b, \mathbf{nM}^{-1}$) | 0.0464 | 0.0265 |
| Degradation rate in tier 1 ($\kappa_d^{(1)}$) | 0.0035 | 0.0045 |
| Degradation rate in tier 2 ($\kappa_d^{(2)}$) | 0.0071 | 0.0068 |
| Internalisation rate ($\kappa_I$) | 0.02 | 0.0513 |
| Recycling rate ($\kappa_R$) | 0.0989 | 0.0770 |
| Conjugation rate ($\kappa_C, \mathbf{nM}^{-1}$) | 0.7605 | 0.7579 |
| Splitting rate ($\kappa_S$) | 0.7038 | 0.3036 |
| Feedback control | | |
| Amplification ($\alpha$) | - | - |
| Feedback Sensitivity ($\gamma$) | 0.0939 | 0.1946 |
| Feedback strength ($n$) | 4.6310 | 0.6202 |
| Tier-wise weights | | |
| weight of tier 1 ($w_1$) | 0.0046 | 0.2875 |
| weight of tier 2 ($w_2$) | 0.9954 | 0.7125 |

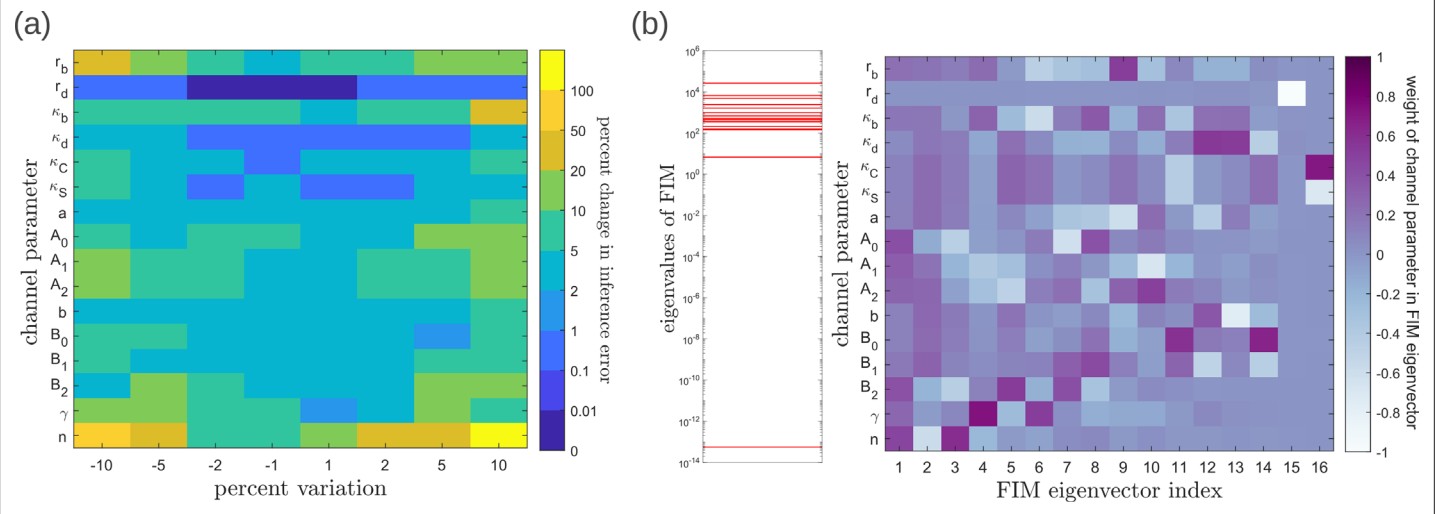

**Figure 10.** Geometry of the fidelity landscape around the optimum. (**a**) Percent changes in the inference error upon perturbations in the channel parameters (as described in **Table 1**) around the optimum for one-tier two-branch channel (optimised $\bar{\sigma}_X = 1.9\%$). For most perturbations, the inference error deviates by up to 20% of the optimum i.e. the inference error $\bar{\sigma}_X$ remains below 2.2%. (**b**) *Left:* eigen spectrum of the Fisher information metric (FIM, see **Equation 21**) around the global minimum of $\bar{\sigma}_X$, *Right:* weight of the different channel parameters in the eigenvectors of FIM, obtained from projecting each eigenvector along the channel parameter axes. The index 1 corresponds to the eigenvector with the largest eigenvalue and the index 16 corresponds to the eigenvector with the smallest eigenvalue.

## Choice of objective function

The objective function as defined in **Equation 7** gave equal weight to inference errors at all positions $x$ along the tissue, driving the inference error to reduce at all positions simultaneously. In certain developmental contexts, the objective could be to partition the tissue into cell identity segments (reviewed in **Briscoe and Small, 2015**). In such a case, the partition boundaries would need to be sharp **Gregor et al., 2007a** that is only the errors at the segment boundaries would need to be minimised. We show that even with this choice of objective function, the qualitative results for the optimal channel architectures remain unaltered. We define the inference error for a tissue with $N_p$ segmented cell identities as.

$$\sigma_X^2(x) = \langle (1 - \delta_{g(x),g(x^*)}) (x^* - x)^2 \rangle_y$$
$$\text{where} \quad g(x) = 1 + \sum_{i=1}^{N_p} \Theta(x - \xi_i) \tag{23}$$

where $\delta$ and $\Theta$ denote the Kronecker-delta and Heaviside-theta functions respectively, $\xi_i$ is the position of the boundary between the i-th and (i+1)-th segments, and $g$ is a function that maps position (actual or inferred) to a segment, that is $g : [0, 1] \rightarrow \{1, 2, ..., N_p\}$ with $N_p$ as the total number of segments.

We optimise one-tier one-branch, one-tier two-branch and two-tier two-branch channel architectures for the inference error as defined in **Equation 23** with $N_p = 4$ and equally spaced boundaries located at positions $\xi_1 = 0.25, \xi_2 = 0.5, \xi_3 = 0.75$ along the $x$-axis. As before, this optimisation suggests that an additional branch aids in reducing the inference errors due to extrinsic noise (compare **Figure 12b and d**), with similar opposing receptor profiles as in Section 'Branched architecture with multiple receptors provides accuracy and robustness to extrinsic noise'. Tiers play only a moderate role in reducing the inference errors further in a two-branch channel (compare **Figure 12d and f**). However, just as with the previous objective function, an additional tier provides substantial robustness to intrinsic noise as shown in **Figure 13c**.

## Experimental verification in the *Drosophila* Wg signalling system

The phenomenology of the morphogen reading and processing of Wg in the wing imaginal disc of *Drosophila melanogaster* **Hemalatha et al., 2016** suggests a one-to-one mapping to the two-tier two-branch channel defined above, thus providing an

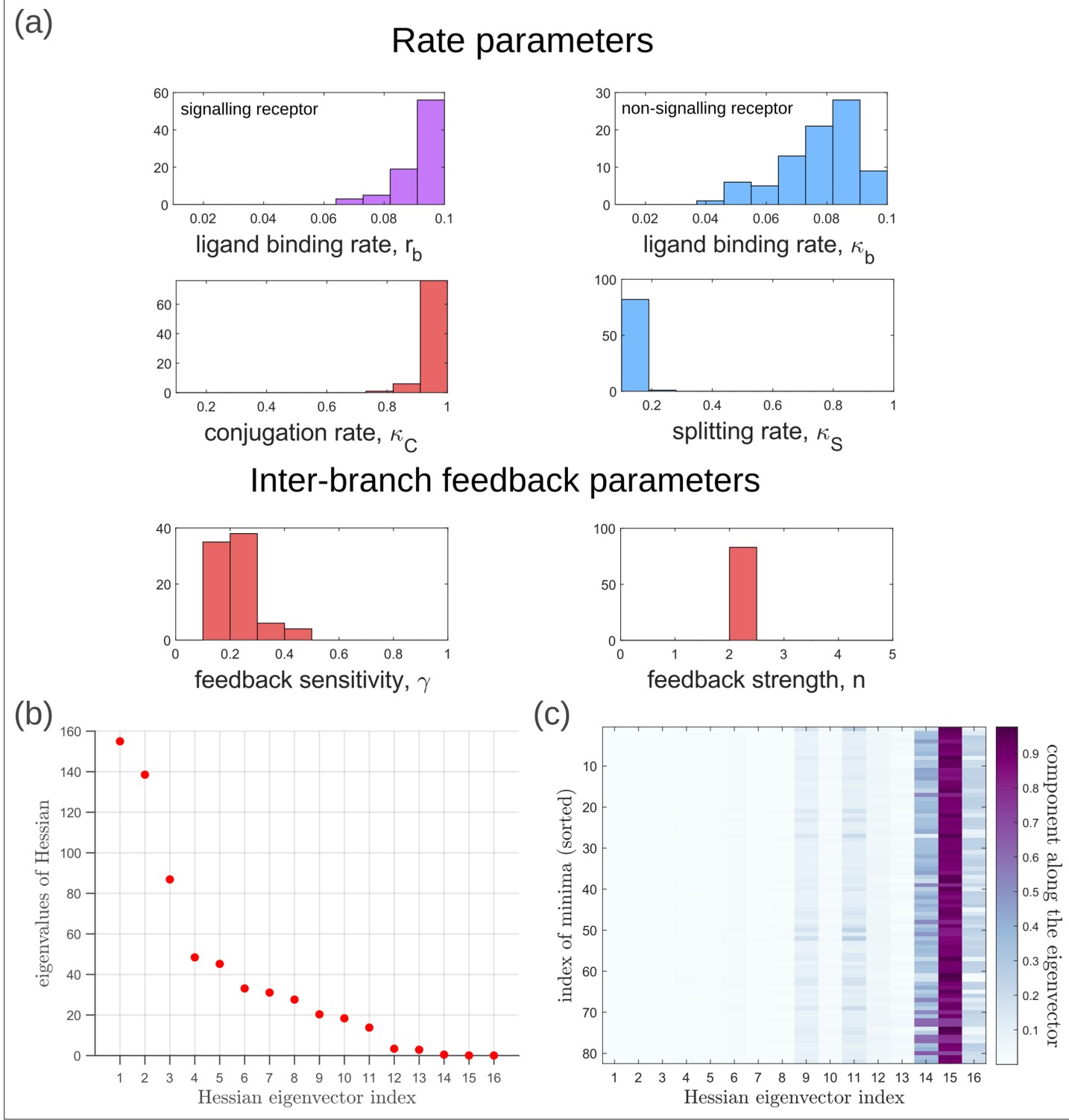

**Figure 11.** Geometry of the low inference error landscape defined by channels within a band $\bar{\sigma}_X \leq 2\%$ about the global minimum. (**a**) Frequency distributions of optimised channel parameters in the low inference error landscape. Here we show the ligand binding rates of the signalling and non-signalling receptors, conjugation and splitting rates, and feedback sensitivity and feedback strength parameters. The distributions of the other optimised channel parameters are shown in Appendix 14. (**b**) Eigenvalues of the Hessian $h_{\mu\nu}$ (see **Equation 22**) of $\bar{\sigma}_X$ around the global minimum. (**c**) Components of the normalised 'position vectors' of the minima $\vec{v}^{\text{opt}} \in \mathcal{V}$ along the eigenvectors of the Hessian $h_{\mu\nu}$, obtained from projecting each position vector along the eigenvector of the Hessian. Here, position vectors in the parameter space $\mathcal{V}$ are defined by the usual Euclidean metric.

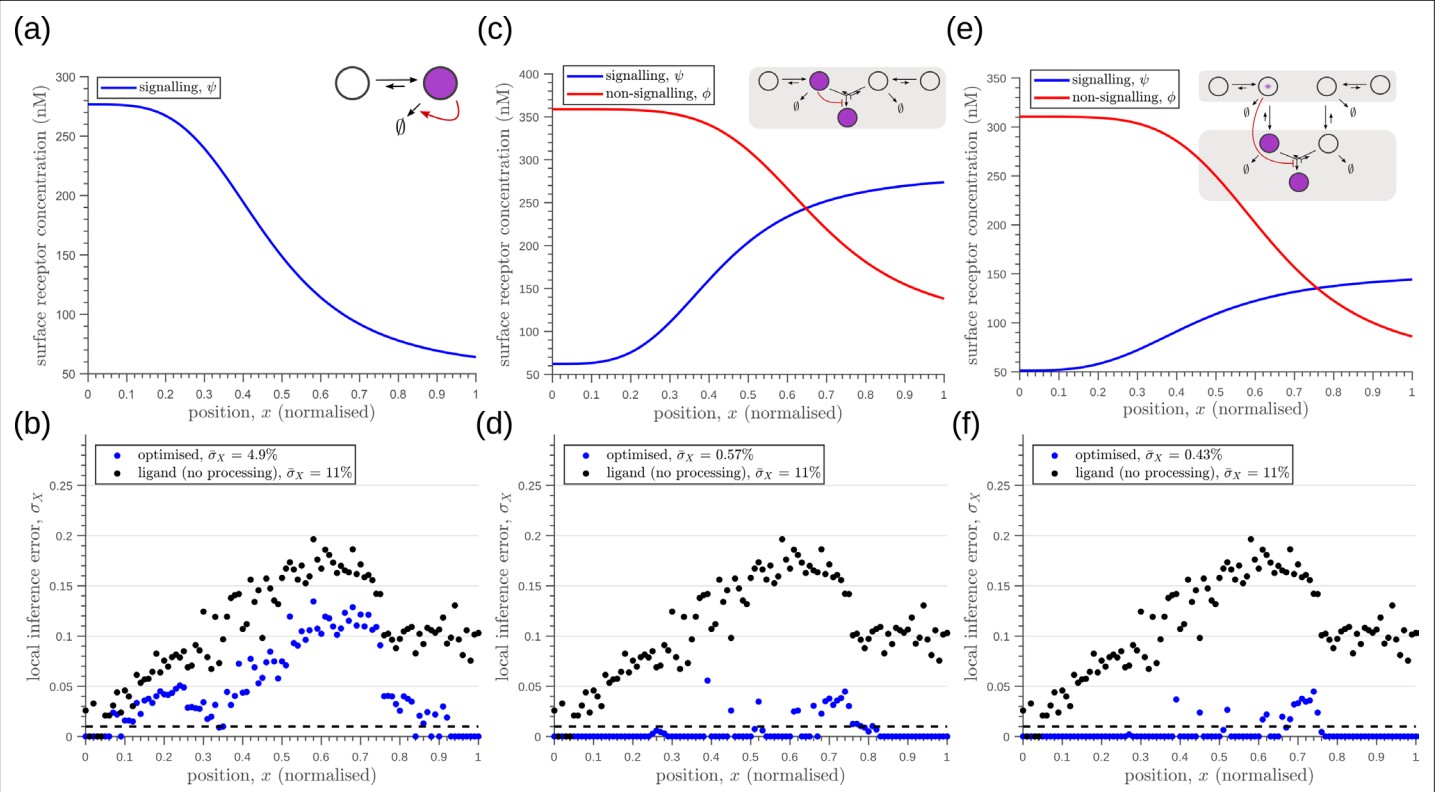

**Figure 12.** Robustness to extrinsic noise with a different choice of objective function (*Equation 23*) for one-tier one-branch channel (**a–b**), one-tier two-branch channel (**c–d**) and two-tier two-branch channel (**e–f**). (**a**) Profile of the signalling receptor for (a, inset) the optimised one-tier one-branch channel. (**b**) Corresponding inference errors due to extrinsic noise in the optimised one-tier one-branch channel. (**c**) Profiles of the signalling (blue) and non-signalling (red) receptor for (c, inset) the optimised one-tier two-branch channel. (**d**) Corresponding inference errors due to extrinsic noise in the optimised one-tier two-branch channel. Errors are predominantly located around the segment boundaries at $x = 0.25, 0.5, 0.75$ and still increase in the direction of reducing morphogen concentrations. (**e**) Profiles of the signalling (blue) and non-signalling (red) receptor for (e, inset) the optimised two-tier two-branch channel. (**f**) Corresponding inference errors due to extrinsic noise in the optimised two-tier two-branch channel. Note that the errors here are predominantly around the segment boundaries ($x = 0.25, 0.5, 0.75$) and diminished compared to the one-tier two-branch channel in (**d**).

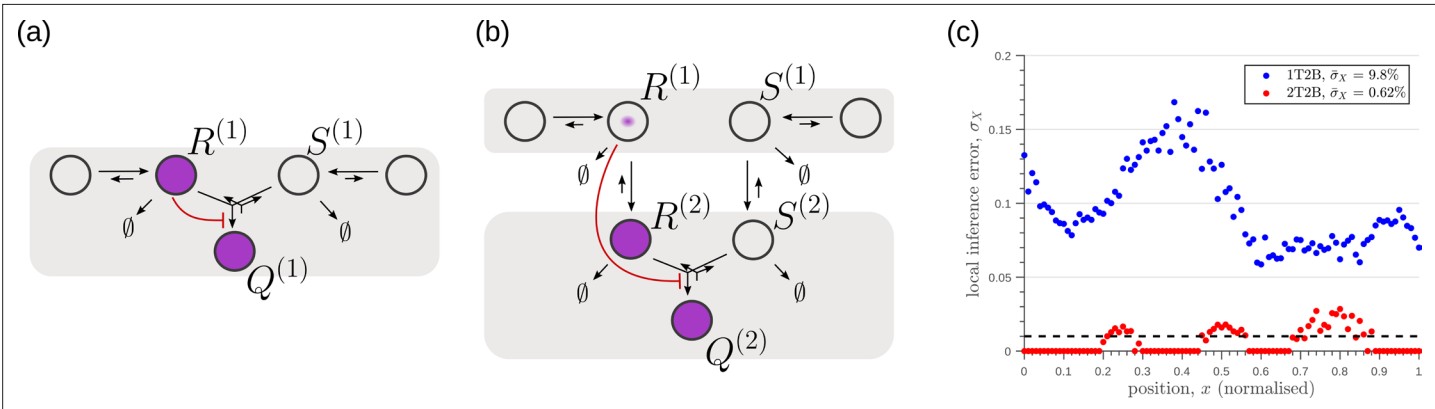

**Figure 13.** Robustness to intrinsic noise, with a different choice of objective function (*Equation 23*), in (**a**) one-tier two-branch (1T2B) channel and (**b**) two-tier two-branch (2T2B) channel architectures, previously optimised for extrinsic noise alone. (**c**) A comparison of local inference errors of the two optimised channels in (**a,b**) in presence of intrinsic noise. Even for this choice of objective function, the two-tier channel shows consistently better performance.

ideal experimental system for a realisation of the ideas presented here (*Figure 14a, b*).

Wingless (Wg) is secreted by a line of cells (1–3 cells) at the dorso-ventral boundary and forms a concentration gradient across the receiving cells *Neumann and Cohen, 1997*. Receiving cells closer to the production domain show higher Wg signalling while those farther away have lower Wg signalling *Neumann and Cohen, 1997*. Several cell autonomous factors influence reading and processing of the morphogen Wg in the receiving cells. Binding of Wg to its signalling receptor, Frizzled-2 (DFz2), initiates signal transduction pathway and nuclear translocation of $\beta$-catenin which further results in activation of Wg target genes (reviewed in *Clevers and Nusse, 2012*). In addition to the signalling receptor, binding receptors such as Heparin Sulphate Proteoglycans (HSPGs) – Dally and Dlp also contribute to Wg signalling *Baeg et al., 2001*; *Franch-Marro et al., 2005*. Further, the two receptors follow distinct endocytic pathways *Hemalatha et al., 2016*: while, DFz2 enters cells via the Clathrin Mediated Endocytic pathway (CME), Wg also enters cells independent of DFz2, possibly by binding to HSPGs, through CLIC/GEEC (CG) endocytic pathway. The two types of vesicles, containing Wg bound to different receptors, merge in common early endosomes *Hemalatha et al., 2016*. However, only DFz2 receptors in their Wg-bound state, both at the cell surface and early endosomes, are capable of generating a downstream signal leading to positional inference through a transcriptional readout *Tsuda et al., 1999*. This phenomenology is faithfully recapitulated in our two-tier two-branch channel architecture (*Figure 14b*) in which DFz2 and HSPG receptors play the role of the two branches. The conjugated state 'Q' represents a combination of the readings from the two branches, possibly realised by the co-receptors HSPGs that bind *Kirkpatrick et al., 2006*; *Capurro et al., 2008* and present *Hemalatha et al., 2016* diffusible ligands to signalling receptors (either on the cell surface or within endosomes).

Since an experimental measurement of positional inference error poses difficulties, we measure the cell-to-cell variation in the signalling output for a given position $x$ as a proxy for inference error (*Appendix 4—figure 3*). Larger the variation, higher is the inference error. This is calculated as coefficient of variation (CV, Appendix 15) in the output across cells in the $y$-direction (*Figure 14c*).

Let us first discuss the results from the theoretical analysis. The optimised two-tier two-branch channel (*Figure 14b*) shows that the magnitude and the fluctuations in the coefficients of variation are small, with a slight increase with position (blue, *Figure 14f*). This is consistent with the low inference error associated with the optimised channel (*Figure 8b*). Upon perturbing this channel via removal of the non-signalling branch, the magnitude and fluctuations in the signalling output variation increases significantly (orange, *Figure 14e*). This qualitative feature of the coefficient of variation in the optimised two -tier two-branch channel is replicated in the Wg measurements of wild type cells.

In the experiments, we first established the method by determining the CV of a uniformly distributed signal, CAAX-GFP (expressed using ubiquitin promoter), and observed that the CV of CAAX-GFP is relatively uniform in $x$, the distance from Wg producing cells (*Figure 14d*). In order to study the steady state distribution of Wg within a cell and within the endosomes, we performed a long endocytic pulse (1 hr) with fluorescently labelled antibody against Wg *Hemalatha et al., 2016*; *Prabhakara et al., 2022*. Following this, we estimated the CV of the Wg endocytic profile as a function of $x$ (*Figure 14f*, and *Figure 14—figure supplement 1*).

We assessed the CV of endocytosed Wg under two conditions: one, where the endocytic pulse of Wg is captured by the two branches and two tiers (control condition), and another, where we disengage one of the tiers by inhibiting the second endocytic pathway using a genetically expressed dominant negative mutant of Garz, a key player in the CG endocytic pathway *Gupta et al., 2009*. This perturbation has little or no effect on the functioning of the CME or the levels of the surface receptors that are responsible for Wg endocytosis (*Hemalatha et al., 2016*; *Prabhakara et al., 2022*). As predicted by the theory (*Figure 14e*), CV in the control shows a slight increase with position (*Figure 14f*) with fluctuations about the mean profile being small. In the perturbed condition, with the CG endocytic pathway disengaged, we find the CV shows a steeper increase with $x$ and has larger fluctuations about the mean profile.

In principle, the coefficient of variation of the output is affected by all the microscopic stochastic processes that intersect with Wg signalling network in the wing imaginal disc and in the ligand input. Therefore, one has to be careful about interpreting the changes in the coefficient of variation of the

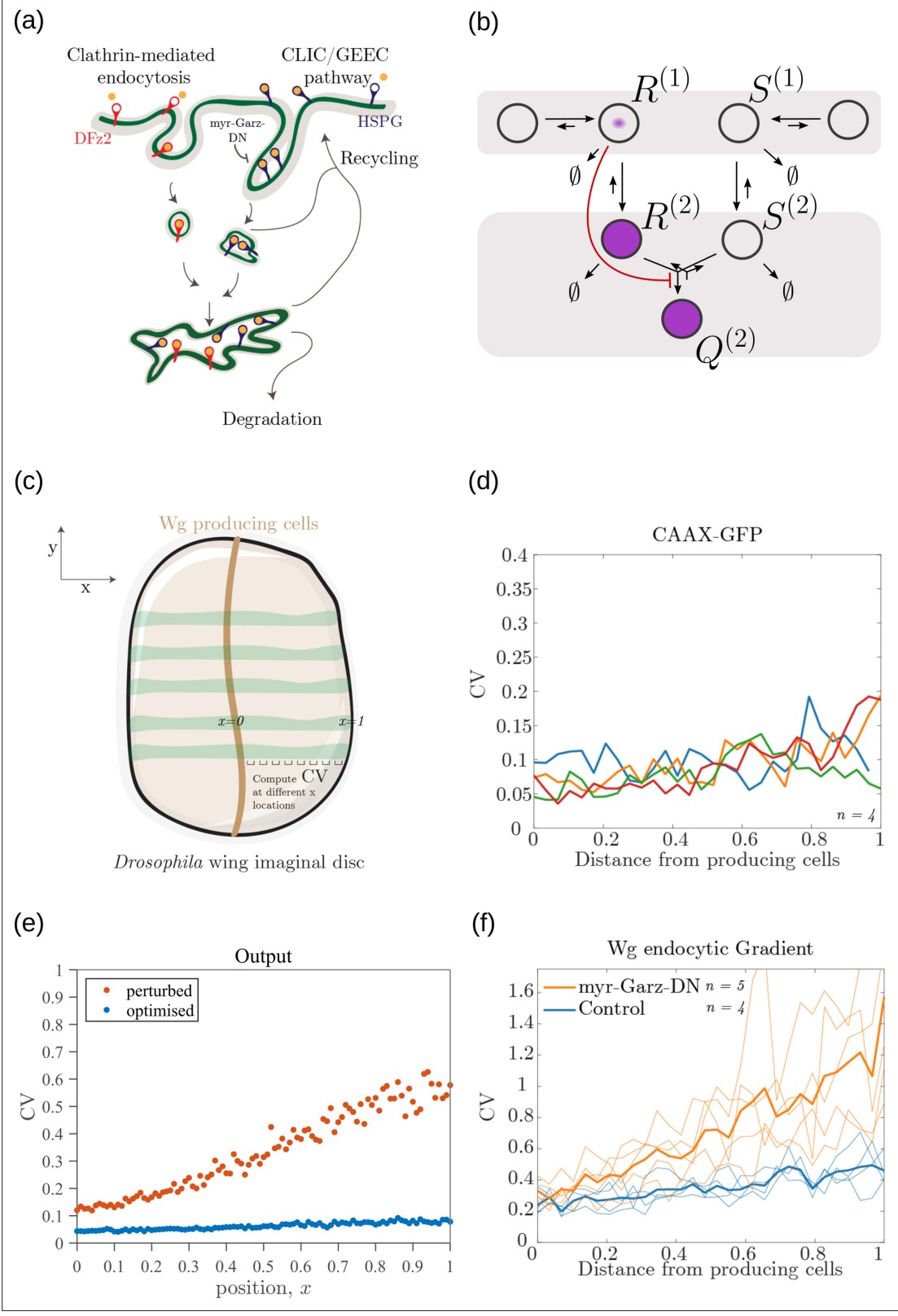

**Figure 14.** Comparison of theoretical results with experimental observations on Wg signalling system in *Drosophila* wing imaginal disc. (**a**) Schematic of the cellular processes involved in Wg signalling, showing the two endocytic routes for the receptors (see text for further description). (**b**) Two-tier two-branch channel architecture corresponding to the Wg signalling system. (**c**) Schematic describing the XY view of wing disc. The vertical brown stripe marks the Wg producing cells. Horizontal green stripes mark the regions in wing disc used for analysis. See *Experimental Methods* (Appendix 15) for

*Figure 14 continued on next page*

*Figure 14 continued*

more information. (**d**) Coefficient of variation (CV) of CAAX-GFP intensity profiles, expressed in wing discs, as a function of (normalized) distance from producing cells (n=4). (**e**) Coefficient of variation in the output of the optimised two-tier two-branch channel (blue), and upon perturbation (orange) via removal of the non-signalling branch, implemented by setting all rates in the non-signalling branch $\kappa$ to zero. The optimised parameter values for the plot can be found in *Table 2* under the column corresponding to $n_T = 2, n_B = 2, r_- = \kappa_C$. (**f**) CV of intensity profiles of endocytosed Wg in control wing discs (C5GAL4Xw1118; blue; n=4) and discs where CLIC/GEEC endocytic pathway is removed using UAS-myr-garz-DN (C5GAL4XUAS-myr-garz-DN; orange; n=5).

The online version of this article includes the following source data and figure supplement(s) for figure 14:

**Source data 1.** CV of intensity measurements of CAAX GFP and endocytosed Wg, in control and myr-Garz-DN, as a function of distance from producing cells in individual samples of wing imaginal discs.

**Figure supplement 1.** Supporting measurements of fluorescence intensity profiles for *Figure 14d* and *Figure 14f*.

**Figure supplement 1—source data 1.** Fluorescence intensity measurements for a-c.

output, based on the such perturbation experiments. Notwithstanding, this qualitative agreement between theory and experiment is encouraging.

## Discussion

In this paper, we have posed the problem of spatial patterning of cell fates in a developing tissue as a *local, cell autonomous* morphogenetic decoding that ensures precise inference of position, that is robust to extrinsic and intrinsic noise. We treat the cells as inference channels capable of reading and processing the morphogen input. We describe the architecture of the inference channels in terms of three elements: branches (number of receptor types), tiers (number of compartments) and feedbacks. We ask for properties of the inference channel architectures that allow for precision and robustness in the task of morphogenetic decoding of cellular position.

### Key results

Taking an information theoretic and systems biology approach, we have addressed the issue of accurate and robust morphogenetic decoding of position. For convenience, we summarise our key results in a point-wise manner:

1. The main result is that given a noisy morphogen input, cells in a developing tissue can achieve low inference error of their positions by deploying a more elaborate *multi-branch multi-tier channel architecture* with feedback control. This ensures a separation between the reading of the morphogen input and buffering against noise.
2. Having a combination of *signalling* and *non-signalling* receptors in the channel can significantly improve the performance of cells in their positional decoding.
3. For a monotonically decaying morphogen input, the signalling and non-signalling receptors exhibit spatially varying profiles with the signalling receptor *increasing* away from the source and the non-signalling receptor *decreasing* away from the source. This implies that the non-signalling receptor 'reads' the morphogen input, while the signalling receptor buffers against noise.
4. The performance of the multi-tier multi-branch channels is enhanced by having a feedback from the signalling branch to the non-signalling branch. Along with control on the levels of signalling receptor, this *inter-branch feedback* provides buffering against extrinsic noise.
5. Having a multi-tier architecture (*cellular compartmentalisation*) tempers the effects of intrinsic noise in the channel by stabilising fluctuations of the output at steady-state.
6. The optimisation shows that the characteristics of the signalling receptor are tightly controlled, whereas those of the non-signalling receptor are flexible. This implies that the signalling receptor is *specific* whereas the non-signalling receptor is *promiscuous*.
7. Analysis of the geometry of the fidelity landscape reveals that the channel parameters corresponding to feedback, binding rates and profile of the signalling receptor are *stiff*, while the rest of the channel parameters are *sloppy* (elaborated in Section 'Geometry of the inference error landscape: implications for control').
8. The efficacy of inter-branch feedback control is enabled by having a *conjugated state* corresponding to a confluence of the signalling and non-signalling branches in a common compartment.

9. Our analysis demonstrates how *local, cell autonomous control* can facilitate the optimisation of a tissue-level task, here morphogenetic decoding of cellular position.

Our theoretical predictions are compared with experimental observations from Wg morphogen system of *Drosophila* wing imaginal disc. We first show that Wg signalling in the experimental system is equivalent to a two-tier two-branch channel. In the experiments, we use signal-to-noise ratio (SNR) of the output as a proxy for robustness of inference. Perturbation of the architecture, i.e. removal of the non-signalling branch, results in reduction of SNR. In a forthcoming manuscript, we will provide a detailed verification of the predicted opposing receptor profiles.

## Geometry of the inference error landscape: implications for control

We have explored the *local* geometry of the fidelity landscape around the optimum, and the *global* geometry of the low inference error states, by perturbing channel parameters and concentration profiles of the receptors.

The local geometry of the fidelity landscape is studied using the Fisher information metric. This shows that steepest variation in the inference error comes from moving along the feedback parameters while perturbations to other channel parameters produces only marginal changes. Further, we explore the global geometry using the spectrum of the Hessian of the inference error. We find that the topography of the low inference error landscape resembles a ravine or a deep valley, which is shallow along the several *sloppy* directions and steep along the few *stiff* directions, the latter being predominantly along the feedback parameters. This dimensional reduction appears to be a recurring feature of such high-dimensional optimisation *Transtrum et al., 2015*; *Yadav et al., 2022*.

Such a geometrical approach also provides insight on the differences between the signalling and the non-signalling receptors, which shows up in the extent to which they influence inference errors in the neighbourhood of the optimum. Slight changes in the signalling receptor away from the optimum lead to a sharp increase in inference error while similar changes in the non-signalling receptor do not affect the inference errors significantly. This gives rise to the notion of stiff and sloppy directions of *control* - with non-signalling receptor placed under sloppy control. In a context with multiple morphogen ligands setting up the different coordinate axes (e.g. Wg, Dpp and Hh in imaginal discs *Lin, 2004*; *Lin and Perrimon, 2000*), the non-specific receptor can potentially facilitate cross-talks between them. A sloppy control on non-specific receptor would allow for accommodation of robustness in the outcomes of the different morphogens. This could potentially be tested in experiments.

## Future directions

We end our discussion with a list of tasks that we would like to take up in the future. First, the information processing framework established here is very general. Obvious extensions of our models, such as adding more branches, tiers and chemical states, will not lead to qualitatively new features. However, one may alter the objective function – for instance, in the case of short range morphogens like Nodal *Liu et al., 2022*, only the positions of certain *regions* (closer to the morphogen source) or *cell fate boundaries* need to be specified with any precision. To this end, we have analysed another objective function which partitions the tissue into cell identity segments. The qualitative features of the optimised channel architectures remain unaltered. Depending on the developmental context, one might explore other objective functions. This would be a task for a future investigation.

Next, our optimisation study ignores cellular costs due to compartmentalisation, additional receptors and implementation of feedback controls, and thus possible trade-offs between cellular economy and precision in inference. Nevertheless, the observation that addition of extra tiers beyond two provides only marginal improvements to inference, already suggests a balance between precision and cellular costs.

Third, our theoretical result that the optimised surface receptor profiles are either monotonically increasing or decreasing from the morphogen source, suggests that the surface receptor concentrations are spatially correlated across cells. Such correlations could have a mechanochemical basis, either via cell surface tension that could in turn affect internalisation rates *Thottacherry et al., 2018* or inter-cellular communication through cell junction proteins *Garcia et al., 2018* or from adaptive feedback mechanisms between the output and receptor concentrations *Barkai and Leibler, 1997*. We emphasize that in the current optimisation scheme, we have

allowed the receptor concentrations to vary over the space of all monotonically increasing, decreasing or flat profiles, and *have not encoded* the positional information in the receptor profiles.

Finally, we have considered the morphogen ligand as an *external input* to the receiving cells, outside the cellular information processing channel. There is no feedback from the output to the receptors and thus no 'sculpting' of the morphogen ligand profile. Morphogen ligand profiles (e.g. Dpp *Romanova-Michaelides et al., 2021*) are set by the dynamics of morphogen production at the source, diffusion via transcytosis and luminal transport, and degradation via internalisation. These cellular processes are common to both the reading and processing modules in our channel architecture. This would suggest a dynamical coupling and feedback between reading and ligand internalisation, which naturally introduces closed-loop controls on the surface receptors and a concomitant sculpting of the morphogen profile.

## Acknowledgements

We thank Thomas Lecuit for insights during the course of investigation. We thank past and present members of the Simons Centre, especially Alkesh Yadav, Amit Kumar, Archishman Raju, Kabir Husain, Mukund Thattai and Sandeep Krishna, for critical inputs. In particular, we thank Archishman Raju for useful discussions on the geometric analysis of the fidelity landscape. We acknowledge support from the Department of Atomic Energy (India), under project no. RTI4006, and the Simons Foundation (Grant No. 287975). MR and SM acknowledge DST (India) for JC Bose Fellowships. SM acknowledges a Margadarshi Fellowship of DBT-Wellcome Trust India alliance (IA/M/15/1/502018).

## Additional information

### Competing interests

Satyajit Mayor: Reviewing editor, eLife. The other authors declare that no competing interests exist.

### Funding

| Funder | Grant reference number | Author |
| --- | --- | --- |
| Department of Atomic Energy, Government of India | RTI4006 | Krishnan S Iyer<br>Chaitra Prabhakara<br>Satyajit Mayor<br>Madan Rao |
| Simons Foundation | 287975 | Madan Rao |
| DBT-Wellcome Trust India Alliance | IA/M/15/1/502018 | Satyajit Mayor |

The funders had no role in study design, data collection and interpretation, or the decision to submit the work for publication.

### Author contributions

Krishnan S Iyer, Conceptualization, Data curation, Software, Formal analysis, Validation, Investigation, Visualization, Methodology, Writing – original draft, Project administration, Writing – review and editing; Chaitra Prabhakara, Resources, Data curation, Formal analysis, Validation, Investigation, Visualization, Methodology, Writing – original draft, Writing – review and editing; Satyajit Mayor, Supervision, Funding acquisition, Validation, Investigation, Visualization, Methodology, Writing – original draft, Project administration, Writing – review and editing; Madan Rao, Conceptualization, Formal analysis, Supervision, Funding acquisition, Validation, Investigation, Visualization, Methodology, Writing – original draft, Project administration, Writing – review and editing

### Author ORCIDs

Krishnan S Iyer ⓘ http://orcid.org/0000-0002-0930-5164
Satyajit Mayor ⓘ http://orcid.org/0000-0001-9842-6963
Madan Rao ⓘ http://orcid.org/0000-0001-6210-6386

Decision letter and Author response

Decision letter https://doi.org/10.7554/eLife.79257.sa1

Author response https://doi.org/10.7554/eLife.79257.sa2

# Additional files

### Supplementary files

• MDAR checklist

### Data availability

Figure 14 and Figure 14—figure supplement 1 contain source data in xlsx format. Modelling code and numerical data is available on Gitlab (https://gitlab.com/eoskrish/morphogenetic-decoding; copy archived at swh:1:rev:59c184aaa46c5e769a95ea39b48e911d0fc4fd5b).

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

## Appendix 1

### Setting up the dynamical equations

The cellular processes involved in a general two-branch channel architecture, described in Section "Quantitative models for cellular reading and processing" and **Figure 4**, with any number of tiers $n_T$ are: production of receptors, binding-unbinding of ligand with the receptors, intracellular transport between cellular compartments, conjugation and splitting in the final tier, and degradation of the chemical species involved. The mass action kinetics for these cellular processes are written as the following set of first-order ordinary differential equations (ODEs):

$$\partial_t R_0^{(1)} = r_p - (r_b L + r_d^{(1)})R_0^{(1)} + r_u R_1^{(1)}$$
$$\partial_t R_1^{(1)} = r_b L R_0^{(1)} - (r_u + r_d^{(1)} + r_I^{(1)})R_1^{(1)} + r_R^{(1)} R_1^{(2)}$$
$$\partial_t S_0^{(1)} = \kappa_p - (\kappa_b L + \kappa_d^{(1)})S_0^{(1)} + \kappa_u S_1^{(1)}$$
$$\partial_t S_1^{(1)} = \kappa_b L S_0^{(1)} - (\kappa_u + \kappa_d^{(1)} + \kappa_I^{(1)})S_1^{(1)} + \kappa_R^{(1)} S_1^{(2)}$$
$$\partial_t R_1^{(2)} = r_I^{(1)} R_1^{(1)} - (r_R^{(1)} + r_d^{(2)} + r_I^{(2)})R_1^{(2)} + r_R^{(2)} R^{(3)}$$
$$\partial_t S_1^{(2)} = \kappa_I^{(1)} S_1^{(1)} - (\kappa_R^{(1)} + \kappa_d^{(2)} + \kappa_I^{(2)})S_1^{(2)} - \kappa_R^{(2)} S^{(3)}$$

$$\cdot$$
$$\cdot$$
$$\cdot$$

$$\partial_t R_1^{(n_T)} = r_I^{(n_T-1)} R_1^{(n_T-1)} - (r_R^{(n_T-1)} + r_d^{(n_T)})R_1^{(n_T)} - \kappa_C R_1^{(n_T)} S_1^{(n_T)} + \kappa_S Q^{(n_T)}$$
$$\partial_t S_1^{(n_T)} = \kappa_I^{(n_T-1)} S_1^{(n_T-1)} - (\kappa_R^{(n_T-1)} + \kappa_d^{(n_T)})S_1^{(n_T)} - \kappa_C R_1^{(n_T)} S_1^{(n_T)} + \kappa_S Q^{(n_T)}$$
$$\partial_t Q^{(n_T)} = \kappa_C R_1^{(n_T)} S_1^{(n_T)} - \kappa_S Q^{(n_T)}$$

$$(24)$$

$R_j^{(k)}$ denotes species in the first branch, associated with the signalling receptor, in tier $k$ and in bound ($j = 1$) or unbound ($j = 0$) states. Likewise, $S_j^{(k)}$ denotes the same for the non-signalling receptor. $Q^{(n_T)}$ denotes the conjugated species in the final tier. Chemical rates related to the first branch are denoted by $r$ while those in the second branch are denoted by $\kappa$. The processes of production, binding, unbinding, internalisation, recycling, conjugation, splitting and degradation are represented by the subscripts $p, b, u, I, R, C, S, d$, respectively (**Table 1**). The superscript on degradation rate indicates the index of the tier in which the degradation takes place. The superscript over internalisation rate stands for the tier-index of the species internalised. Superscript over the recycling rate follows that of the corresponding internalisation rate.

As described in Section 'Quantitative models for cellular reading and processing', we consider local cell-autonomous control on the total cell surface receptor concentrations, in a manner akin to the *open-loop control Stengel, 1994*, such that the sum of free and bound signalling receptors is $\psi(x)$ and that of the non-signalling receptor is $\phi(x)$. This implies that the open-loop control actuated on the production (or secretion) of the cell surface receptors balances the net flux of receptors away from the surface. Consider the dynamics of $R_0^{(1)}$ and $R_1^{(1)}$ in Eq. A, with now an open-loop control realised through $r_p(t)$ on $R_0^{(1)}$.

$$\partial_t R_0^{(1)} = r_p(t) - (r_b L + r_d^{(1)})R_0^{(1)} + r_u R_1^{(1)}$$
$$\partial_t R_1^{(1)} = r_b L R_0^{(1)} - (r_u + r_d^{(1)} + r_I^{(1)})R_1^{(1)} + r_R^{(1)} R_1^{(2)}$$

At any position $x$, if the total receptor concentration $R_0^{(1)}(x,t) + R_1^{(1)}(x,t) = \psi(x)$ are to stay constant in time (through a chemostat), $r_p(t)$ must satisfy

$$r_p(t) = r_d^{(1)}(R_0^{(1)} + R_1^{(1)}) + r_I^{(1)} R_1^{(1)} - r_R^{(1)} R_1^{(2)}$$

We emphasize that here we do not consider a closed-loop version of the receptor control. However, it can be realised through an integral control *Goodwin et al., 2001*, also known as perfect adaptation *Barkai and Leibler, 1997*. With open-loop-like control on total surface receptor concentrations (Section 'Quantitative models for cellular reading and processing'), $R_0^{(1)}$ and $S_0^{(1)}$ are

replaced by $\psi - R_1^{(1)}$ and $\phi - S_1^{(1)}$, respectively. We then drop the subscripts from $R_1^{(k)}$ and $S_1^{(k)}$ to simplify notation.

$$\partial_t R^{(1)} = r_b L(\psi(x) - R^{(1)}) - (r_u + r_d^{(1)} + r_I^{(1)})R^{(1)} + r_R^{(1)} R^{(2)}$$
$$\partial_t S^{(1)} = \kappa_b L(\phi(x) - S^{(1)}) - (\kappa_u + \kappa_d^{(1)} + \kappa_I^{(1)})S^{(1)} + \kappa_R^{(1)} S^{(2)}$$
$$.$$
$$.$$
$$.$$
$$\partial_t R^{(n_T)} = r_I^{(n_T - 1)}R^{(n_T - 1)} - (r_R^{(n_T - 1)} + r_d^{(n_T)})R^{(n_T)} - \kappa_C R^{(n_T)}S^{(n_T)} + \kappa_S Q^{(n_T)}$$
$$\partial_t S^{(n_T)} = \kappa_I^{(n_T - 1)}S^{(n_T - 1)} - (\kappa_R^{(n_T - 1)} + \kappa_d^{(n_T)})S^{(n_T)} - \kappa_C R^{(n_T)}S^{(n_T)} + \kappa_S Q^{(n_T)}$$
$$\partial_t Q^{(n_T)} = \kappa_C R^{(n_T)}S^{(n_T)} - \kappa_S Q^{(n_T)}$$

(25)

To keep the notation simple, we avoid explicitly indicating superscripts over chemical rate symbols when presenting the dynamical equations in the main text (*Equations 10–17*). In the numerical analysis, however, the degradation, internalisation and recycling rates of different tiers are treated separately. The steady-state solutions of the above equations, without feedback controls on chemical rates, are given by,

One-tier channel:

$$R^{(1)} = \frac{r_b L}{r_b L + r_u + r_d^{(1)}} \psi$$
$$S^{(1)} = \frac{\kappa_b L}{\kappa_b L + \kappa_u + \kappa_d^{(1)}} \phi$$
$$Q^{(1)} = \frac{\kappa_C}{\kappa_S} R^{(1)} S^{(1)}$$

(26)

Two-tier channel:

$$R^{(1)} = \frac{r_b L}{r_b L + r_u + r_d^{(1)} + \frac{r_d^{(2)} r_I^{(1)}}{r_R^{(1)} + r_d^{(2)}}} \psi$$
$$S^{(1)} = \frac{\kappa_b L}{\kappa_b L + \kappa_u + \kappa_d^{(1)} + \frac{\kappa_d^{(2)} \kappa_I^{(1)}}{\kappa_R^{(1)} + \kappa_d^{(2)}}} \phi$$
$$R^{(2)} = \frac{r_I^{(1)}}{r_R^{(1)} + r_d^{(2)}} R^{(1)}$$
$$S^{(2)} = \frac{\kappa_I^{(1)}}{\kappa_R^{(1)} + \kappa_d^{(2)}} S^{(1)}$$
$$Q^{(2)} = \frac{\kappa_C}{\kappa_S} R^{(2)} S^{(2)}$$

(27)

For channels with $n_T > 2$, the solutions are written in the following recursive form Channel with $n_T$ tiers:

$$R^{(1)} = \frac{r_b L}{r_b L + r_u + \tilde{r}_d^{(n_T)}} \psi$$
$$R^{(k)} = \frac{r_I^{(k-1)}}{r_R^{(k-1)} + \tilde{r}_d^{(n_T - k + 1)}} R^{(k-1)} \quad \text{for } k = 2, 3, ..., n_T$$
$$S^{(1)} = \frac{\kappa_b L}{\kappa_b L + \kappa_u + \tilde{\kappa}_d^{(n_T)}} \phi$$
$$S^{(k)} = \frac{\kappa_I^{(k-1)}}{\kappa_R^{(k-1)} + \tilde{\kappa}_d^{(n_T - k + 1)}} S^{(k-1)} \quad \text{for } k = 2, 3, ..., n_T$$
$$Q^{(n_T)} = \frac{\kappa_C}{\kappa_S} R^{(n_T)} S^{(n_T)}$$

(28)

where $\tilde{r}_d^{(k)}$ are in turn evaluated recursively as

$$\tilde{r}_d^{(1)} = r_d^{(n)}$$

$$\tilde{r}_d^{(n_T-k+1)} = r_d^{(k)} \left( 1 + \frac{r_I^{(k)}}{r_R^{(k)} + \tilde{r}_d^{(k+1)}} \frac{\tilde{r}_d^{(k+1)}}{r_d^{(k)}} \right) \quad \text{for } k = n_T - 1, n_T - 2, ..., 1$$

$$\tilde{\kappa}_d^{(1)} = \kappa_d^{(n)}$$

$$\tilde{\kappa}_d^{(n_T-k+1)} = \kappa_d^{(k)} \left( 1 + \frac{\kappa_I^{(k)}}{\kappa_R^{(k)} + \tilde{\kappa}_d^{(k+1)}} \frac{\tilde{\kappa}_d^{(k+1)}}{\kappa_d^{(k)}} \right) \quad \text{for } k = n_T - 1, n_T - 2, ..., 1$$

The general form of *Equations 26–28* holds upon introduction of inter-branch feedbacks (refer *Figure 5*) and therefore the set of ODEs (*Equation 25*) can be solved analytically by appropriately replacing the rates under feedback with the Hill-form functions discussed in Section 'Quantitative models for cellular reading and processing'. Cases with other types of feedbacks need to be solved numerically (Appendix 3).

In writing these dynamical equations, we have made two assumptions, simply as a matter of convenience. One may note that only the ligand bound states of receptors are transported between tiers. This is done with the consideration that residence times of unbound receptors within a compartment, other than the cell surface (first tier), is very small that is receptors in enclosed compartments re-bind the ligand quickly after unbinding due to small volume of the compartment. Further, the open loop control on receptors need not be strictly on surface receptors (tier 1). In principle, the control could be on the production rate of receptors. This does not change the solution in any substantial manner. For instance, in the case of two tiers, the solution would be

$$R_1^{(1)} = \frac{r_b L}{r_b L \left( 1 + \frac{r_I^{(1)}}{r_R^{(1)} + r_d^{(2)}} \right) + r_u + r_d^{(1)} + \frac{r_d^{(2)} r_I^{(1)}}{r_R^{(1)} + r_d^{(2)}}} \frac{r_p}{r_d^{(1)}} \tag{29}$$

$$S_1^{(1)} = \frac{\kappa_b L}{\kappa_b L \left( 1 + \frac{\kappa_I^{(1)}}{\kappa_R^{(1)} + \kappa_d^{(2)}} \right) + \kappa_u + \kappa_d^{(1)} + \frac{\kappa_d^{(2)} \kappa_I^{(1)}}{\kappa_R^{(1)} + \kappa_d^{(2)}}} \frac{\kappa_p}{\kappa_d^{(1)}} \tag{30}$$

and the rest of the species would follow hierarchically as in the case with surface receptor control (see *Equation 28*). As long as the deviations in the denominator, $\frac{r_d r_I}{r_R + r_d}$ and $\frac{\kappa_d \kappa_I}{\kappa_R + \kappa_d}$ remain small compared to $r_u$, and $\psi \sim \frac{r_p}{r_d^{(1)}}, \phi \sim \frac{\kappa_p}{\kappa_d^{(1)}}$, the steady solutions for the two cases (control on surface receptors versus total receptors) will not differ significantly.

## Appendix 2

### Heuristics and choice of feedback controls

The figure below illustrates the effect of feedback parameters in *Equations 18; 19* discussed under Section 'Quantitative models for cellular reading and processing'.

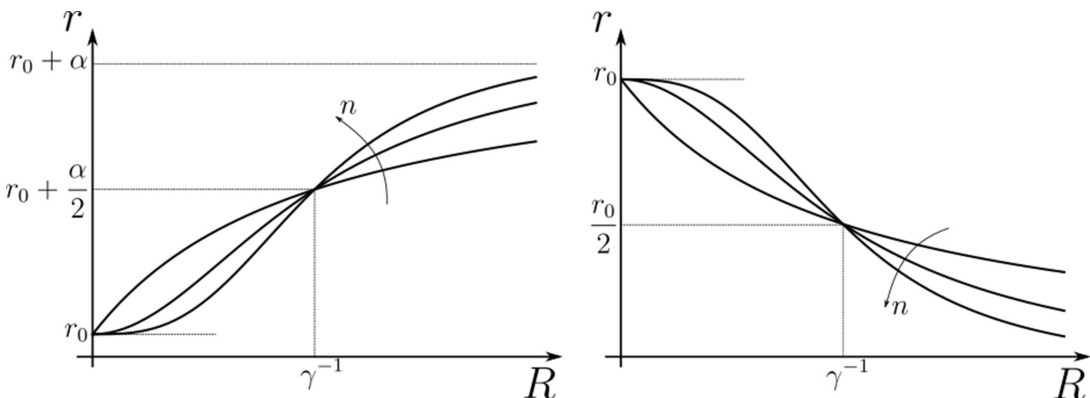

**Appendix 2—figure 1.** Hill functions representing positive (left) and negative (right) feedback control on chemical rates $r$ actuated by chemical species $R$. The effect of feedback amplification $\alpha$, sensitivity $\gamma$ and strength $n$ are indicated. $r_0$ denotes the reference value of the chemical rate $r$ in absence of feedback.

Considering the choices of chemical rate under control $r \in \{r_I, r_R, r_d, \kappa_I, \kappa_R, \kappa_d\}$ and the actuating nodes $R \in \{R^{(1)}, R^{(2)}, ...\}$, there are a large number of possible feedbacks for any given channel architecture. However, categorising the different feedbacks according to their effect can reduce the redundancies and aid numerical analysis. The steady-state output of a channel without the feedback controls on chemical rates is instructive in this categorisation. For instance, consider the output $\theta$ for a two-tier two-branch channel with $n_T = 2$.

$$
\begin{aligned}
\theta &= w_{n_T} Q^{(n_T)} + \sum_{k=1}^{n_T} w_k R^{(k)} \\
&= w R^{(1)} \left( 1 + \frac{1-w}{w} \frac{r_I}{r_R + r_d} \left( 1 + \frac{\kappa_C}{\kappa_S} \frac{\kappa_I}{\kappa_R + \kappa_d} S^{(1)} \right) \right)
\end{aligned}
\tag{31}
$$

This form of the output suggests that feedbacks from $R^{(1)}$ (term outside the bracket) on any of the rates in $\frac{r_I}{r_R + r_d}$ and $\frac{\kappa_C}{\kappa_S} \frac{\kappa_I}{\kappa_R + \kappa_d}$ could cross-correlate $R^{(1)}$ with the other signalling species $R^{(2)}$ and $Q^{(2)}$. Thus, a negative feedback on $r_I$ ($\kappa_I$) would be qualitatively equivalent to a positive feedback on $r_R$ ($\kappa_R$). Beside helping to reduce the number of CRNs that need to be considered, this observation also allows one to forego the numerical analysis of a CRN with feedback on $r_R$ from $R^{(1)}$ that may have non-unique solutions. Similarly, feedbacks from different but related actuators could be interchanged. For example, the feedback on $\kappa_I$ from $R^{(2)}$, i.e. $\kappa_I \to \frac{\kappa_I}{1+(\gamma' R^{(2)})^n}$, could be written in terms of the feedback on $\kappa_I$ from $R^{(1)}$, i.e. $\kappa_I \to \frac{\kappa_I}{1+(\gamma R^{(1)})^n}$, by a simple transformation on feedback sensitivity $\gamma' = \frac{r_I}{r_R + r_d} \gamma$. Note that *Equation 31* also indicates the preferred direction of feedbacks, that is with the signalling receptors as actuators.

## Appendix 3

### A note on numerical methods

Optimisation over the channel parameter vector $\vec{v} \in \mathcal{V}$, belonging to a parameter space $\mathcal{V}$, is highly non-convex (see Section 'Performance of the Channel Architectures'). Moreover, the derivative of $\bar{\sigma}_X$ with respect to $\vec{v}$ is ill-defined due to finite sampling of input distributions. With these considerations, we chose a derivative-free (gradient independent) optimisation algorithm viz. Pattern Search algorithm *Hooke and Jeeves, 1961*. The implementation of this algorithm in MATLAB can be found as *patternsearch* function under the Global Optimization Toolbox.

With this pattern search algorithm, we perform the optimisation routine as described in Section 'Performance of the Channel Architectures' in two rounds. First, with a certain number of initial points $n_{init} = 32$ in the parameter space $\mathcal{V}$, we determine the advantageous feedback topologies (CRNs) that is, that give lower inference errors. In the next round, we go through the same optimisation routine for the advantageous feedback topologies determined through the the first round, but now with $n_{init} = 320$ to converge to the true global minimum. Using parallel computing clusters, the cost of this computation in terms of real running time can be brought down to 12–48 hr.

Ideally, this optimisation ought to be done with extrinsic and intrinsic noises considered together. However, we restrict optimisation to the case of extrinsic noise as the computational cost associated with steady-state solutions of chemical master equations (CMEs), in the case of intrinsic noise, is rather high. Therefore, we evaluate the inference error due to intrinsic noise only of those CRNs that were optimised in the context of extrinsic noise previously. For this, we solve the CMEs using adaptive explicit-implicit tau leaping algorithm *Sandmann, 2009* to determine the steady-state outputs and thus inference error. Implementation of this algorithm requires the definition of some numerical parameters that (i) help switch between Gillespie and implicit/explicit tau-leaping algorithms ($n_a, n_d$), (ii) decide the number of reactions when Gillespie algorithm is selected ($n_b$), and (iii) various threshold ($n_c, \epsilon$). The values chosen for these are $n_a = 10$, $n_b = 10$ and $n_b = 100$ if implicit tau-leaping algorithm was used in the previous step, $n_c = 10$, $n_d = 100$, $\delta = 0.05$ and $\epsilon = 0.1$.

## Appendix 4

### Robustness, sensitivity, and trade-offs

Precision in positional inference requires both that the output variance at a given position be small (i.e. the output is *robust* to the noise in input) and that the mean output at two neighbouring positions be sufficiently different (the output is *sensitive* to systematic changes in mean input).

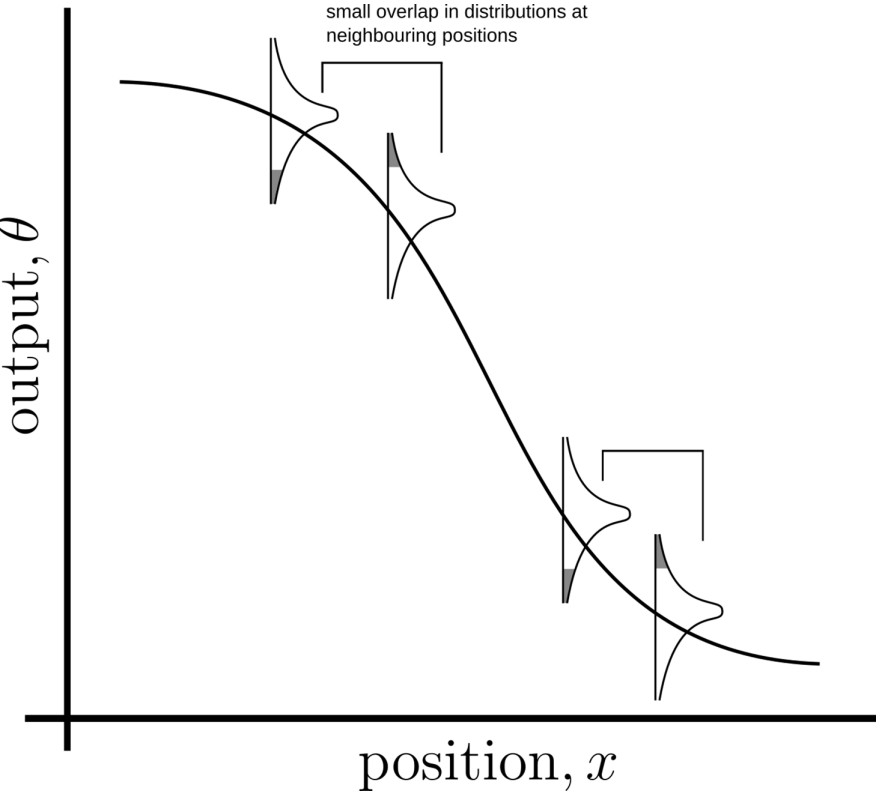

**Appendix 4—figure 1.** Schematic of an *ideal* output profile. The monotonically decreasing curve is the mean output profile. Due to various sources of noise, the output at any point will have a distribution around the local mean. If neighbouring cohorts of cells are to accurately distinguish their inferred positions from their outputs (in the Bayesian sense), the output distributions must have a small overlap (shaded region under the distribution curves).

With this heuristic understanding of an *ideal* output, we define two local measures: (i) $\chi$ measures the noise in the output of a cell cohort as compared to the noise in the received input. We define it as the ratio of coefficient of variation in the output $\theta$ to the coefficient of variation in the input $L$ for the same cohort of cells.

$$\chi \equiv \frac{\mathrm{CV}_\theta}{\mathrm{CV}_L}. \tag{32}$$

Thus, *robustness* of the output increases with decreasing values of $\chi$; (ii) $\xi$ measures the *sensitivity* of the output to a systematic change in the input. We define it as the ratio of relative change in the mean output $\langle\theta\rangle$ to the relative change in the mean input $\mu_L$ for the same cell cohort.

$$\xi \equiv \frac{\Delta\langle\theta\rangle}{\langle\theta\rangle} \bigg/ \frac{\Delta\mu_L}{\mu_L} \tag{33}$$

The angular brackets in the above equation denote averaging over cells belonging to the same cohort. Thus, higher *sensitivity* implies higher values of $|\xi|$. Precision in positional inference, in terms of these two local measures, implies simultaneous minimisation of $\chi$ and $|\xi|^{-1}$.

We calculate $\chi$ and $\xi$ at all positions $x$ in the various optimised channels. We find that addition of a branch allows for better robustness and alleviates the tension between sensitivity to systematic change in mean input and robustness to input noise (*Appendix 4—figure 2*).

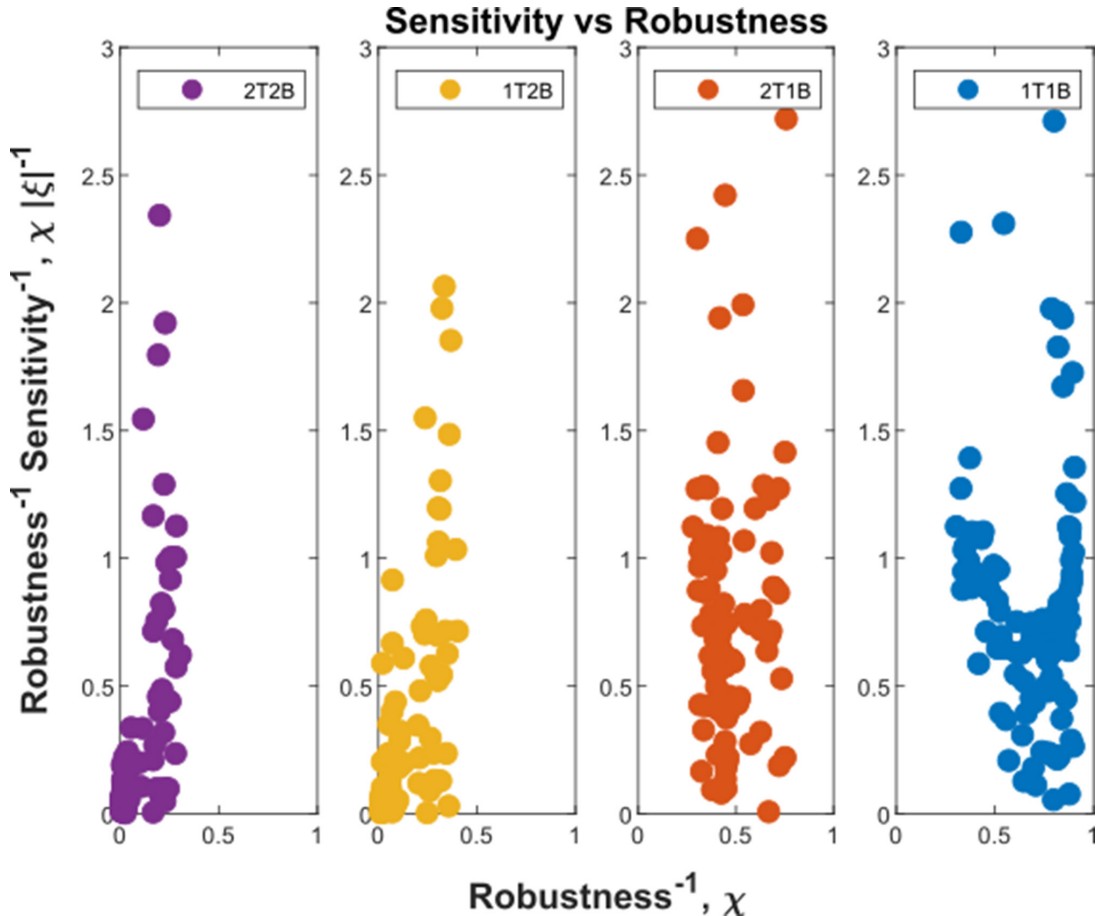

**Appendix 4—figure 2.** Robustness-Sensitivity plots for $n_x - 1 = 100$ cohorts in the optimised channel architectures. The robustness-sensitivity objective is to reach the origin along both the axes. Optimised single-branch architectures (labelled 1T1B and 2T1B) show the two measures as conflicting objectives, such that improvement in robustness is achieved at the expense of sensitivity beyond a certain point. This conflict is absent in the optimised two-branch architectures (labelled 1T2B and 2T2B). An additional tier in two-branch architectures can further improve the two local measures. Note: the choice of coordinate axes reflects the requirement of simultaneous minimisation of $\chi$ (*Equation 32*) and $\xi^{-1}$ (*Equation 33*) and the allowed tolerance in $\xi^{-1}$ when $\chi$ is below a certain level.

Furthermore, a one-tier two-branch channel with an inter-branch feedback control (*Figure 7a* of the main text) shows a preference towards certain concentrations of the two receptors, which provide the cellular output robustness to input noise, i.e. minimise $\chi$. To maintain lower values of $\chi$ (higher robustness), the signalling receptors $\psi$ must decrease with increasing mean of the input $\mu_L$ (*Appendix 4—figure 3*). Increasing the non-signalling receptors $\phi$ as a function of mean input $\mu_L$ helps separate the mean outputs $\langle\theta\rangle$ at neighbouring positions and thus aids in increasing the sensitivity (*Appendix 4—figure 4*). This is consistent with the receptor profiles in the optimised one-tier two-branch channel (*Figure 7b* of the main text).

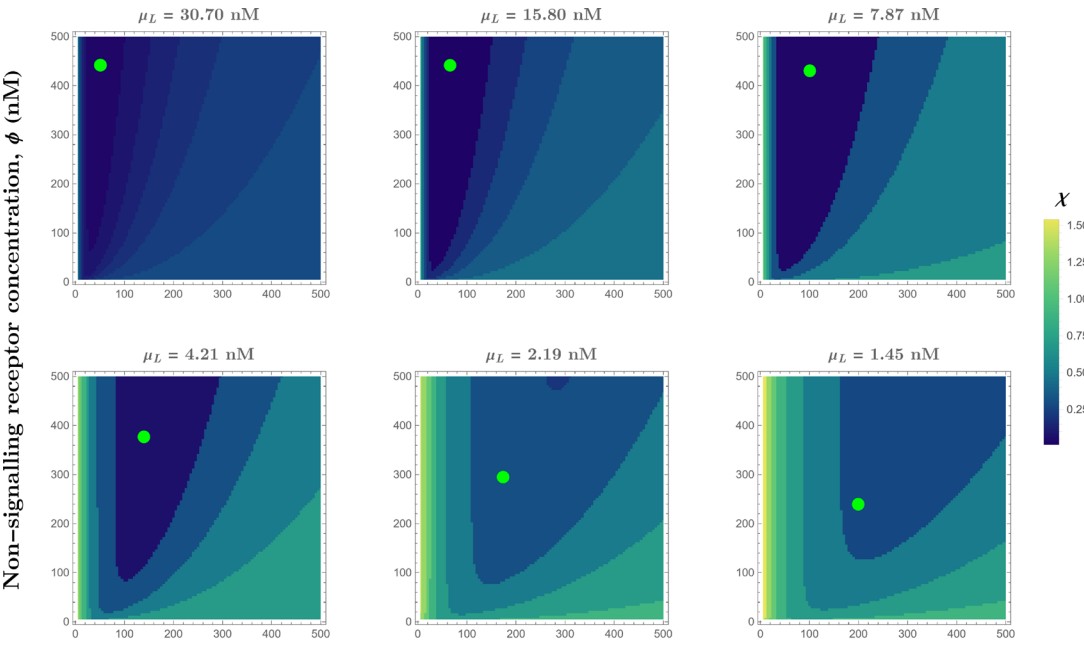

**Appendix 4—figure 3.** Contour plots of $\chi$ (see Eq. *Equation 32*) in the optimised one-tier two-branch channel with inter-branch feedback control (*Figure 7a* of the main text) showing the preferred receptor combinations (deep blue) for different values of mean input $\mu_L$. Green dots denote the receptor concentrations in the optimised channel (*Figure 7b*, inset of the main text) at positions corresponding to the values of mean input $\mu_L$ indicated above the contour plots. The optimised parameter values for the plots can be found in *Table 2* under the column corresponding to $n_T = 1, n_B = 2, r_- = \kappa_C$.

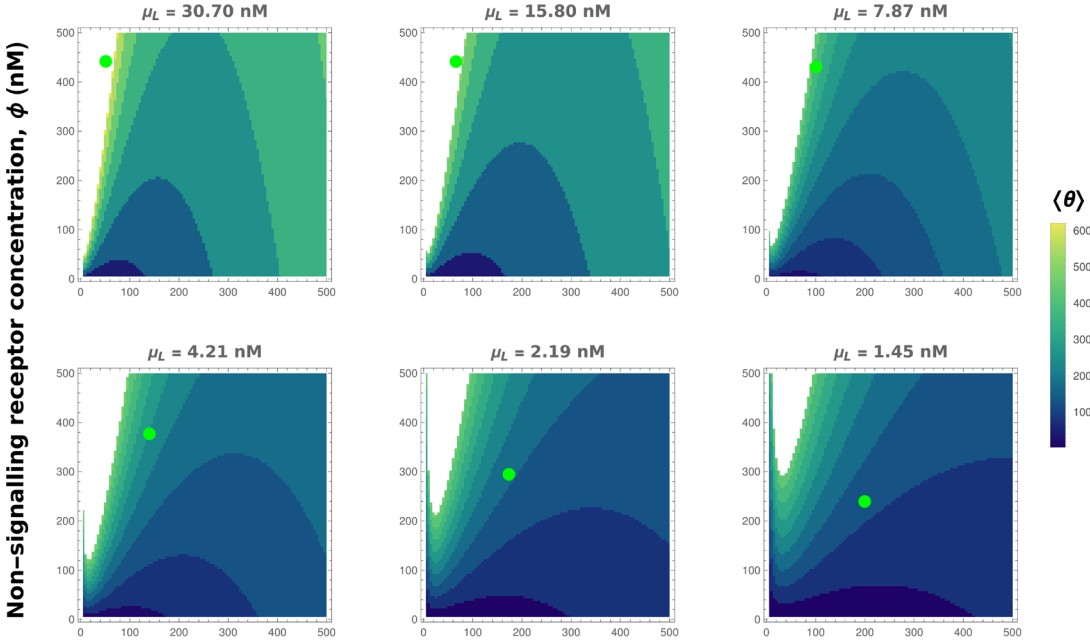

**Signalling receptor concentration, $\psi$ (nM)**

**Appendix 4—figure 4.** Contour plots of mean output $\langle \theta \rangle$ in the optimised one-tier two-branch channel with inter-branch feedback control (*Figure 7a* of the main text). The contours move downward along the axis of the non-signalling receptor $\phi$. Therefore, as the preferred values of signalling receptor $\psi$ decrease with mean ligand input $\mu_L$ (*Appendix 4—figure 3*), non-signalling receptor concentration $\phi$ needs to increase with $\mu_L$ to ensure that the
*Appendix 4—figure 4 continued on next page*

*Appendix 4—figure 4 continued*

mean output $\langle\theta\rangle$ is a monotonically increasing function of.$\mu_L$ Green dots denote the receptor concentrations in the optimised channel (***Figure 7b***,inset of the main text) at positions corresponding to the values of mean input $\mu_L$ indicated above the contour plots. The optimised parameter values for the plots can be found in ***Table 2*** under the column corresponding to.$n_T = 1, n_B = 2, r_- = \kappa_C$

## Appendix 5

### Input-output relations in a *minimal* channel

Here, we try to understand why having an appropriate spatial gradient of receptors helps in separating the outputs in nearby cells, thus facilitating accurate positional inference. This is best done in the simplest case of a *minimal* channel that is with one tier and one branch. This channel 'reads' the ligand input through binding of the ligand on only one, signalling receptor.

*Appendix 5—figure 1a* shows that setting receptor conentrations to decrease with mean ligand input creates an unfavourable scenario. Low receptor availability for higher ligand concentrations ensures a saturation of the output at lower values (blue). On the other hand, higher receptor availability for ligand concentrations is impractical as the output remains low despite increase in potential saturation point (purple). This causes large overlaps of the outputs at neighbouring positions. On the other hand, *Appendix 5—figure 1b* shows that the output are better separated with receptor profiles that increase with the mean ligand input and thus would enable a better positional inference (Appendix 4).

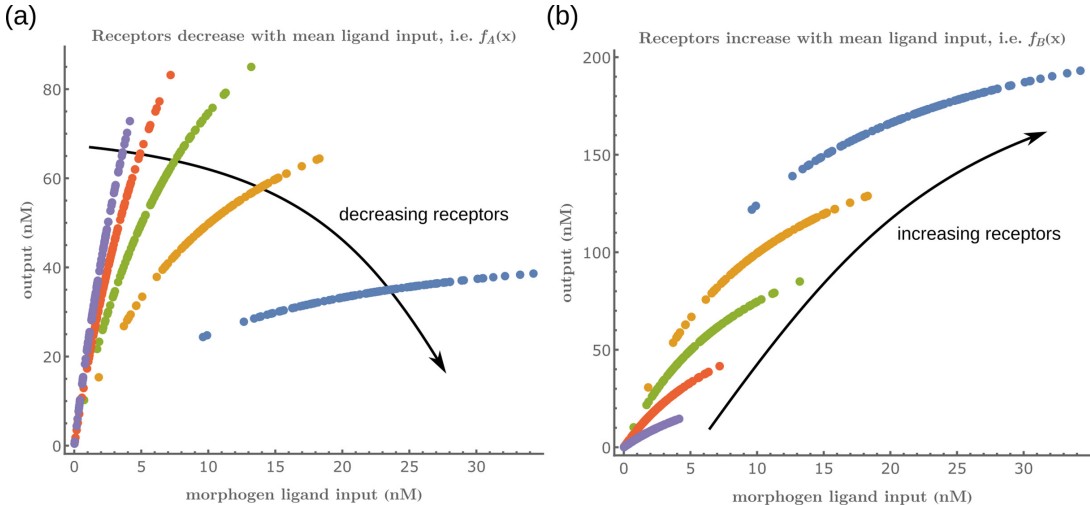

**Appendix 5—figure 1.** Input-Output function of a *minimal* channel shows the importance of choosing the correct receptor profiles.

## Appendix 6

### Robustness due to inter-branch feedback

Here we describe the logic behind the working of an inter-branch feedback control. Consider the case of a one-tier two-branch channel with the feedback on conjugation rate $\kappa_C$ (*Figure 7a* of the main text) actuated by the ligand bound state of the signalling receptor in the first tier $R^{(1)}$. The steady-state solution to *Equations 10–12* corresponding to this channel with $\kappa_C \to \frac{\kappa_C}{1+(\gamma R^{(1)})^n}$ is given by.

$$
\begin{aligned}
R^{(1)} &= \frac{r_b L}{r_b L + r_u + r_d}\psi \\
S^{(1)} &= \frac{\kappa_b L}{\kappa_b L + \kappa_u + \kappa_d}\phi
\end{aligned}
\tag{34}
$$

$$
Q^{(1)} = \frac{\kappa_C}{\kappa_S}\frac{r_b L \psi}{r_b L + r_u + r_d}\frac{\kappa_b L \phi}{\kappa_b L + \kappa_u + \kappa_d}\frac{1}{1 + \left(\gamma\frac{r_b L}{r_b L + r_u + r_d}\psi\right)^n}
\tag{35}
$$

Therefore, the output can be expressed in terms of $L, \psi$ and $\phi$ as follows.

$$
\theta = \frac{r_b L \psi}{r_b L + r_u + r_d}\left(1 + \frac{\kappa_C}{\kappa_S}\frac{\kappa_b L \phi}{\kappa_b L + \kappa_u + \kappa_d}\frac{1}{1 + \left(\gamma\frac{r_b L}{r_b L + r_u + r_d}\psi\right)^n}\right)
\tag{36}
$$

*Appendix 6—figure 1* describes the behaviour of the signalling species $R^{(1)}$ and $Q^{(1)}$ as given by *Equations 34; 35* where the parameters are taken from the optimised one-tier two-branch channel. $R^{(1)}$ increases monotonically with the ligand input $L$ (blue curve, *Appendix 6—figure 1*) and saturates at a value set by the signalling receptor $\psi$. Meanwhile, the conjugate species $Q^{(1)}$ has a non-monotonic behaviour (yellow curve, *Appendix 6—figure 1*): for very low values of input, $Q^{(1)}$ rises sharply due to absence of the feedback effect from the small values of $R^{(1)}$, but then decreases with further increase in input as the value of the feedback actuator $R^{(1)}$ rises. This anti-correlated behaviour of $R^{(1)}$ and $Q^{(1)}$ due to the feedback results in the output $\theta \equiv R^{(1)} + Q^{(1)}$ being a more stable function of the input for an intermediate range of the input (region around the cusp in the green curve, *Appendix 6—figure 1*). Modulating the signalling $\psi$ and non-signalling $\phi$ receptors allows for placement of the stability region (cusp) in accordance with the range of ligand input received at any position $x$, thus tempering the noise in the output (see input-output relations in *Figure 7d* of the main text).

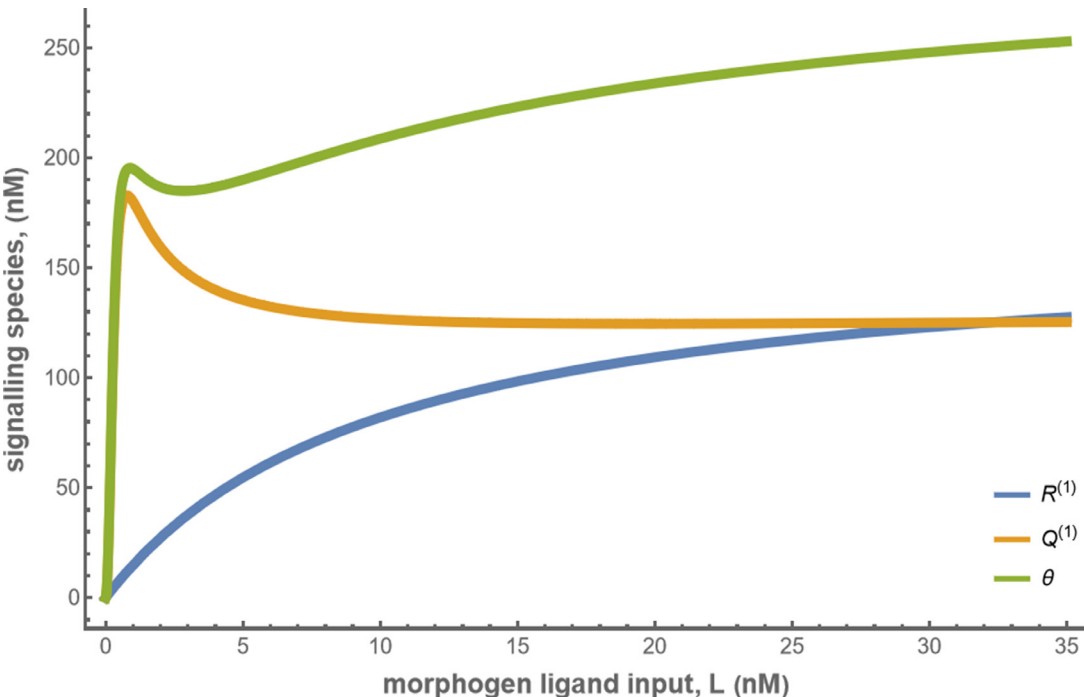

**Appendix 6—figure 1.** Concentrations of the signalling species $R^{(1)}, Q^{(1)}$ and total cellular output $\theta$ in the optimised one-tier two-branch channel with $\psi = 164\,\text{nM}, \phi = 318\,\text{nM}$. The optimised chemical rates and feedback parameters for the above plot can be found in *Table 2* under the column corresponding to $n_T = 1, n_B = 2, r_- = \kappa_C$.

## Appendix 7

### Heuristic for the influence of feedback action on receptor profiles

The optimised one-tier two-branch channel showed a monotonically increasing profile of the signalling receptor $\psi$, and a monotonically decreasing profile of the non-signalling receptor $\phi$ (*Figure 7b*). Here, we provide an understanding for this counter-intuitive result using the forms of the channel output $\theta$ and variation in the output $\partial_L \theta \big|_{\psi, \phi}$.

Consider the output $\theta$ for a one-tier two-branch channel with inter-branch feedback on the conjugation rate $\kappa_C$. *Equation 37* gives the explicit form for the output of this channel in terms of input ligand concentration $L$, signalling $\psi$ and non-signalling $\phi$ receptor concentrations, feedback strength $n$, feedback sensitivity $\gamma$ and binding rates $r_b, \kappa_b$. The last three are small parameters in the optimised channel i.e. $r_b \simeq 0.1, \kappa_b \simeq 0.05, \gamma \simeq 0.2$. Note that the conjugation and splitting rates are absorbed into $\phi$, i.e. $\frac{\kappa_C}{\kappa_S} \phi \to \phi$.

$$\theta = \frac{L\psi}{L + r_b^{-1}} \left[ 1 + \frac{L\phi}{L + \kappa_b^{-1}} \frac{1}{1 + (\gamma \frac{L\psi}{L + r_b^{-1}})^n} \right] \tag{37}$$

We now look at the output in the limit of high ligand input $L \simeq 2r_b^{-1} \simeq \kappa_b^{-1}$, and low ligand input $L \simeq 0.1 r_b^{-1}$. The output forms in these limits are given by *Equation 38* and *Equation 39*.

$$\theta^{\text{high L}} = \frac{2}{3}\psi + \frac{1}{3} \frac{\phi\psi}{1 + (\frac{2}{3}\gamma\psi)^n} \tag{38}$$

$$\theta^{\text{low L}} = r_b\psi + \frac{r_b\psi \, \kappa_b\phi}{1 + (r_b\gamma\psi)^n} \tag{39}$$

To analyse the variation of channel output $\theta$ with ligand input $L$, we compute $\partial_L \theta$ at fixed receptor concentrations $\psi, \phi$.

$$\partial_L \theta \Big|_{(\psi,\phi)} = \frac{r_b^{-1}\psi}{(L + r_b^{-1})^2} \left[ 1 + \frac{L\phi}{L + \kappa_b^{-1}} \frac{1}{1 + (\gamma\frac{L\psi}{L+r_b^{-1}})^n} \right]$$
$$+ \frac{\phi}{L + \kappa_b^{-1}} \frac{L\psi}{L + r_b^{-1}} \frac{1}{1 + (\gamma\frac{L\psi}{L+r_b^{-1}})^n} \left[ \frac{\kappa_b^{-1}}{L + \kappa_b^{-1}} + n \frac{r_b^{-1}}{L + r_b^{-1}} \frac{1}{1 + (\gamma\frac{L\psi}{L+r_b^{-1}})^{-n}} \right] \tag{40}$$

In the limits of high and low ligand input, the form simplifies to

$$\partial_L \theta \Big|_{(\psi,\phi)}^{\text{high L}} = \frac{r_b\psi}{9} + \frac{1}{18} \frac{\psi\phi}{1 + (\frac{2}{3}\gamma\psi)^n} \left( r_b + 3\kappa_b - r_b \frac{n}{1 + (\frac{2}{3}\gamma\psi)^{-n}} \right) \tag{41}$$

$$\partial_L \theta \Big|_{(\psi,\phi)}^{\text{low L}} = r_b\psi + \frac{r_b\psi \, \kappa_b\phi}{1 + (r_b\gamma\psi)^n} \left( 2 - \frac{n}{1 + (r_b\gamma\psi)^{-n}} \right) \tag{42}$$

We provide numerical estimates for the output and the variation in the output, in the high and low ligand input limits, with all possible receptor profile combinations (*Figure 3*). As shown in the table below (highlighted in bold), it is optimal to have lower levels of the signalling receptor $\psi \sim 5r_b^{-1}$ and higher levels of the non-signalling receptor $\phi \sim 50r_b^{-1}$ in the high $L$ region (close to source). On the other hand, in the low $L$ region (far from source), it is optimal to have higher levels of the signalling receptor $\psi \sim 15r_b^{-1}$ and lower levels of the non-signalling receptor $\phi \sim 15r_b^{-1}$. These receptor combinations in the two limits on the input (highlighted rows in the table below) help maximally separate the output at the two ends of the tissue while keeping the variation in the output low at both ends. Taking the receptor profiles as monotonic functions of position, this would imply that for a one-tier two-branch channel with an inter-branch feedback, the optimum signalling receptor $\psi$ has a monotonically increasing spatial profile while the optimum non-signalling receptor $\phi$ has a monotonically decreasing spatial profile. This qualitatively explains the result obtained for

the optimised one-tier two-branch channel in Section 'Branched architecture with multiple receptors provides accuracy and robustness to extrinsic noise', *Figure 7*.

| | high $L$ | | low $L$ | |
|---|---|---|---|---|
| | **output** $\theta$ | **variation** $\partial_L\theta$ | **output** $\theta$ | **variation** $\partial_L\theta$ |
| low $\phi$, low $\Phi$ | 293.95 | 0.59 | 23.75 | 165.93 |
| high $\phi$, low $\Phi$ | 156.53 | 1.67 | **22.52** | **1.12** |
| low $\phi$, high $\Phi$ | **902.07** | **0.69** | 67.5 | 543.46 |
| high $\phi$, high $\Phi$ | 288.43 | 1.69 | 40.06 | −25.19 |

## Appendix 8

### Results of optimisation of two-tier two-branch channels

As discussed in Section 'Branched architecture with multiple receptors provides accuracy and robustness to extrinsic noise', additional tiers in a two-branch channel only marginally reduced the inference errors due to extrinsic noise when compared to the optimised one-tier two-branch channel (*Figure 8*). Here, we show that both the receptor profiles and the input-output relations of these two optimised two-tier two-branch channels are qualitatively similar (*Appendix 8—figure 1*).

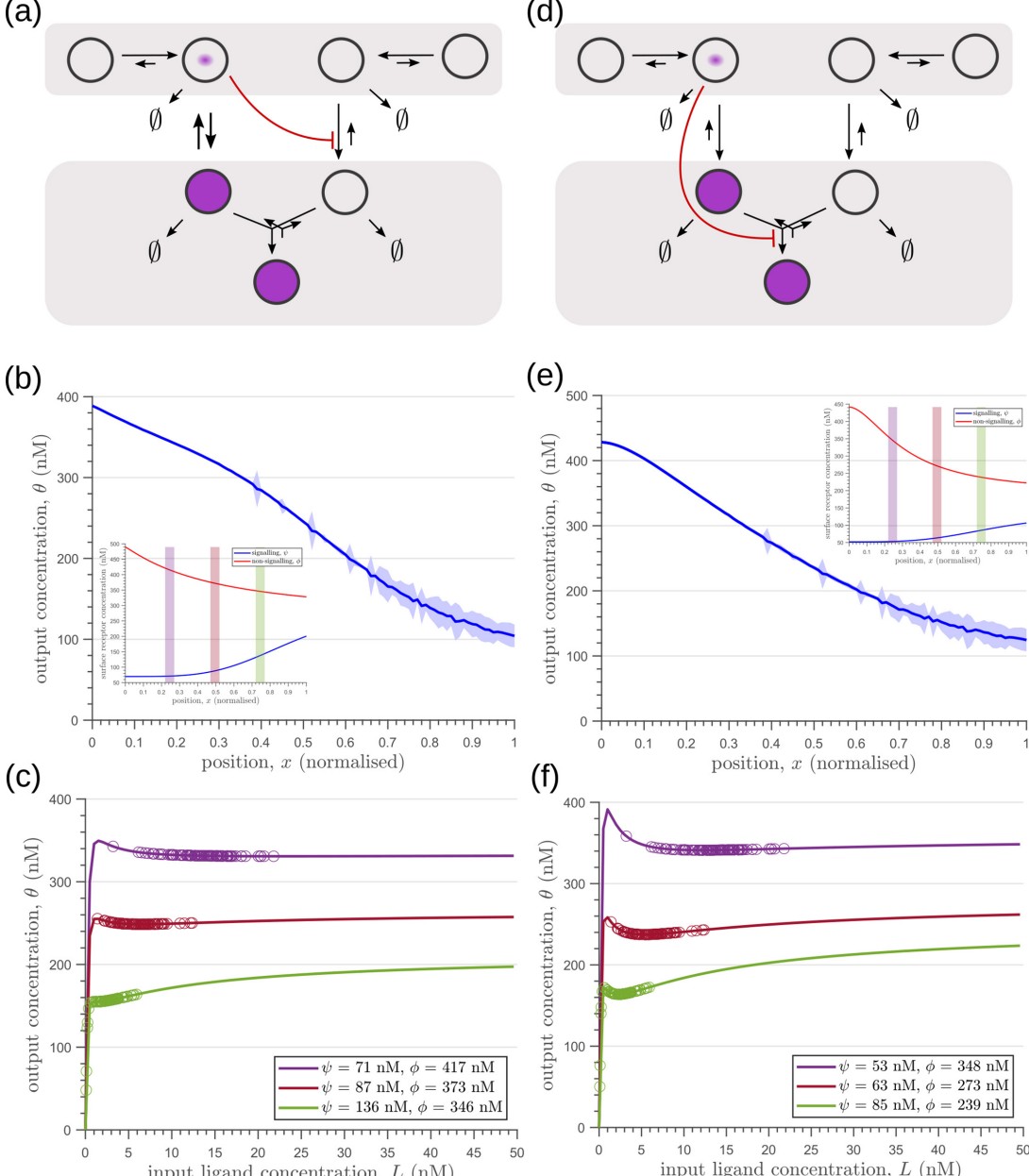

**Appendix 8—figure 1.** Results of optimisation of (**a,d**) two-tier two-branch channels. (**b,e**) The output profiles (with standard error in shaded region) and (insets) the corresponding optimised signalling (blue) and non-signalling (red) receptor profiles. (**c,f**) The input-output relations at selected positions $x = 0.25, 0.5, 0.75$ are shown as solid lines, shaded with the same colour as the position-markers (coloured rectangles in b,e insets). The signalling $\psi(x)$ and non-signalling $\phi$ receptor concentrations are mentioned in the legend. For a fixed distribution of ligand input (*Equation 1*), the range of input values recorded by the receptors at the selected positions gives rise to a range of outputs (circles). Tuning of input-output relations through receptor concentrations reduces output

*Appendix 8—figure 1 continued on next page*

*Appendix 8—figure 1 continued*
variance and minimises overlaps in the outputs of neighbouring cell cohorts. The optimised parameter values for the plots in (**b–c,e–f**) can be found in *Table 2* under the column corresponding to $n_T = 2, n_B = 2, r_- = \kappa_I$ and $n_T = 2, n_B = 2, r_- = \kappa_C$, respectively.

## Appendix 9

### Additional tiers dampen the effects of feedback to provide stability

Stochastic simulations of the chemical reaction network (CRN) corresponding to the optimised one-tier two-branch channel (*Figure 9a* of the main text) show large fluctuations in the time trajectories of signalling species $Q^{(1)}$ about its steady-state mean (*Figure 9d–f* of the main text). $Q^{(1)}$ increases due to absence of feedback from small values of $R^{(1)}$. However, beyond some amount of increase in $Q^{(1)}$, the trajectories veer back towards their mean values. The state of small $R^{(1)}$ and increasing $Q^{(1)}$ can be maintained by replenishment of $R^{(1)}$ from binding reaction followed by immediate conjugation with large pools of $S^{(1)}$ due to high availability of the non-signalling receptor $\phi$. Such amplified fluctuations are absent in the two-tier two-branch channel (*Figure 9g–i* of the main text). Here, we provide heuristics of the differing behaviours in the optimised one-tier and two-tier channels by analysing a simpler set of CRNs (*Appendix 9—figure 1*) with the essential elements of these channels.

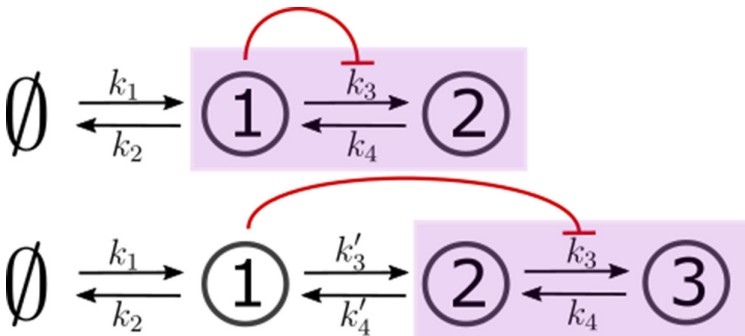

**Appendix 9—figure 1.** Two- and three-species CRNs with production $k_1$ and degradation $k_2$ rates of species 1 (**s₁**) and inter-conversion $k_3, k_4$ rates between the "signalling" (output generating) species (in purple box). These rates mimic the binding $r_b$, unbinding $r_u$ and conjugation-splitting $\kappa_C, \kappa_S$ rates respectively in the optimised one-tier two-branch and two-tier two-branch channels (*Figure 9a and b* of the main text). Consistent with this mapping, the feedback is from species 1 on $k_3$. The three-species CRN has additional rates $k_3', k_4'$ mimicking internalisation $r_I$ and recycling $r_R$ rates, respectively, in the optimised two-tier two-branch channel (*Figure 9b* of the main text). In both cases, output is the sum of the last two nodes in the purple box.

The dynamics for the first, two-species CRN is given by.

$$\dot{s_1} = k_1(c - s_1) - (k_2 + k_3(s_1))s_1 + k_4 s_2$$
$$\dot{s_2} = k_3(s_1)s_1 - k_4 s_2$$

(43)

and that of the second, 3-species CRN by

$$\dot{s_1} = k_1(c - s_1) - (k_2 + k_3')s_1 + k_4' s_2$$
$$\dot{s_2} = k_3' s_1 - (k_4' + k_3(s_1))s_2 + k_4 s_3$$
$$\dot{s_3} = k_3(s_1)s_2 - k_4 s_3$$

(44)

with constant $c$ playing the role of $\psi$, thus providing an upper bound to species 1. In both these sets of equations, we consider a feedback $k_3$ such that $k_3 \rightarrow \frac{k_0}{1 + (\gamma s_1)^n}$ where $k_0$ represents the reference value of $k_3$ in absence of feedback. *Appendix 9—figure 2* shows the phase portrait for the dynamics of the 2-species CRN when $k_0 \gg k_1$ with a moderately strong feedback $n = 2$. This is representative of $\kappa_C S^{(1)} \gg r_b L$ in the optimised one-tier two-branch channel (*Figure 9a* of the main text). The nullcline $\dot{s_1} = 0$ (dashed, green curve) acts as a separatrix for the behaviour of this system: if due to fluctuations in species $s_1$ the system $(s_1, s_2)$ crosses the nullcline from its steady-state (pink point), it sets out on a large trajectory (black line) such that $s_1$ remains close to zero while $s_2$ grows fast until the system turns back towards the steady state. This is similar to the trajectories of $R^{(1)}$ and $Q^{(1)}$ discussed earlier. The non-linearity of the separatrix is due to the feedback from $s_1$ on the rate $k_3$ that couples to $s_1$ in *Equation 43*. Higher production rates $k_1$ (akin to higher values of ligand $L$ in the optimised channel) bring the steady-state closer to the separatrix, making crossing of the separatrix due to fluctuations more likely. Having an additional node in between the actuator species and the

controlled rate in the three-species CRN removes the non-linearity in $\dot{s_1} = 0$ and provides buffering against this effect (*Equation 44*).

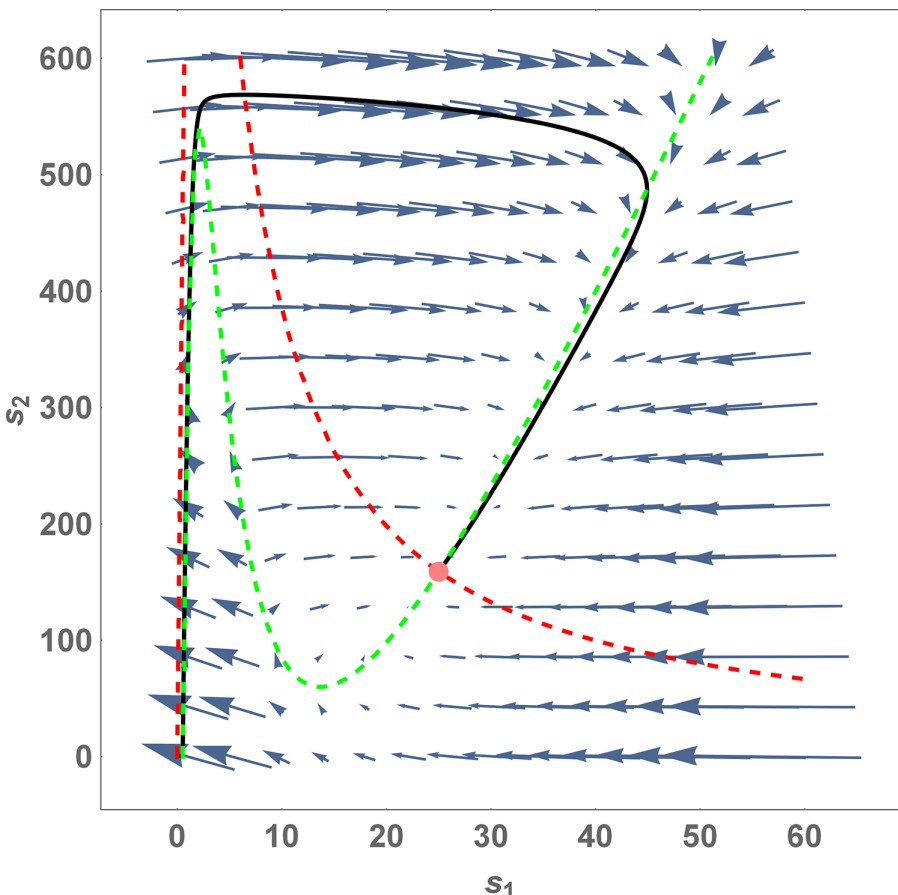

**Appendix 9—figure 2.** Phase portrait for *Equation 43* with $k_0 \gg k_1$. The red and green dashed lines are the nullclines $\dot{s_2} = 0$ and $\dot{s_1} = 0$, respectively. The pink dot denotes the steady-state solution (stable fixed point) of this system. The solid black line is a trajectory with initial point at the origin. Note that at steady-state, $s_1 \ll s_2$. If $s_1$ drops beyond the green nullcline due to a fluctuation, the system goes back to the steady state through a long trajectory with $s_1$ essentially remaining close to zero for a long period of time while $s_2$ increases dramatically. Parameter values for the plot: $k_1 = k_2 = k_4 = 1, k_3 = 100, c = 50, \gamma = 0.5, n = 2$.

## Appendix 10

### Two-tier two-branch channel with no feedback

Here we show that the two-tier two-branch channel without any inter-branch feedback control has fundamentally different optimisation characteristics and a poorer positional inference. Crucially, the optimised profiles of both signalling $\psi$ and non-signalling $\phi$ receptors are monotonically decreasing away from the source (*Appendix 10—figure 1b*, inset).

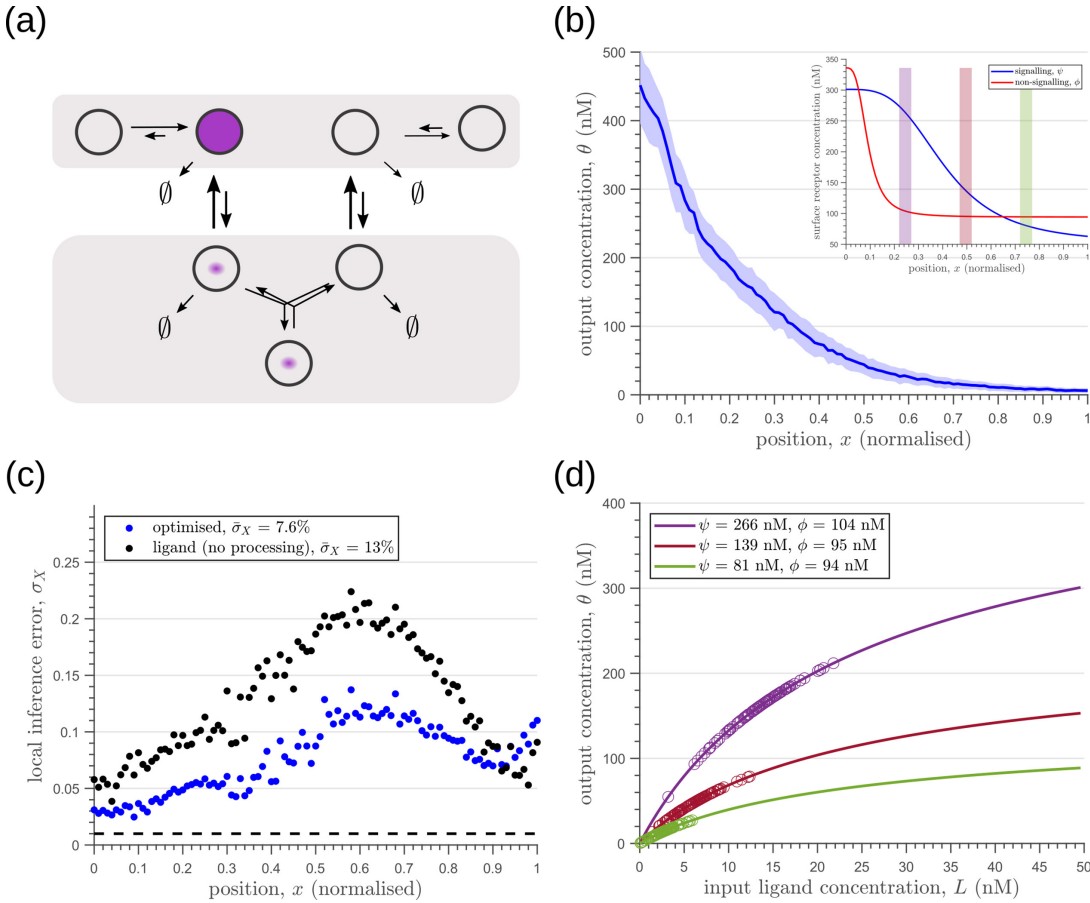

**Appendix 10—figure 1.** Results of optimisation of (**a**) two-tier two-branch channel with no feedback on rates. (**b**) The output profile (with standard error in shaded region) corresponding to the (inset) optimised signalling (blue) and non-signalling (red) receptor profiles. (**c**) The local inference error $\sigma_X(x)$ is only marginally reduced throughout the tissue, when compared to the expected inference errors from ligand with no processing. The dashed line corresponds to a local inference error of one cell's width $\sim 1/n_x$. (**d**) The input-output relations in this channel are monotonically increasing sigmoid functions saturating at only large values of input. The solid lines correspond to the input-output relations at selected positions, $x = 0.25, 0.5, 0.75$ shaded with the same colour as the position-markers in (b inset, coloured rectangles). The signalling $\psi(x)$ and non-signalling $\phi(x)$ receptor concentrations are mentioned in the legend. For a fixed distribution of ligand input (*Equation 1*), the range of input values recorded by the receptors at the selected positions gives rise to a range of outputs (circles). It is clear that neighbouring positions have significant overlaps in their outputs. The optimised parameter values for the plots in (**b–d**) can be found in *Table 2* under the column corresponding to. $n_T = 2, n_B = 2, r_- = \{\}$.

## Appendix 11

### Uniform receptor profiles with uncorrelated noise

Note that in arriving at the optimised channel characteristics for a given morphogen profile, we go through all possible monotonic receptor profiles, including flat profiles. The optimised receptor profiles show a spatial gradient. Here, we ask why can't a flat receptor profile (possibly modified by noise) infer positions accurately from a noisy morphogen gradient? Following the arguments in Appendix 4, we reason that if the morphogen gradient was not corrupted by noise, then flat receptor profiles would have sufficed to infer positions accurately. It is because one wants to discriminate between morphogen concentrations in neighbouring cells in a noisy background that there is a need for a spatial variation in the receptor profiles.

To demonstrate this, we consider uniform spatial profiles, with or without uncorrelated noise, for both the signalling and non-signalling receptors (*Appendix 11—figure 1b, c*), and optimise the rates and feedback parameters anew (*Table 3*) to show that this leads to a higher inference error compared to the optimal (*Appendix 11—figure 1a*). In fact, the inference error in these cases, even with an inter-branch feedback, is only marginally smaller than a channel with no processing of the ligand (black dots in *Appendix 11—figure 1a*). The inference from flat receptor profiles reflects the noise in morphogen gradient itself. This provides the motivation for choosing monotonically increasing or decreasing profiles for both the signalling and non-signalling receptors (*Equations 8; 9*). Note that this implicitly assumes spatial correlations in the surface concentrations of receptors across the cells.

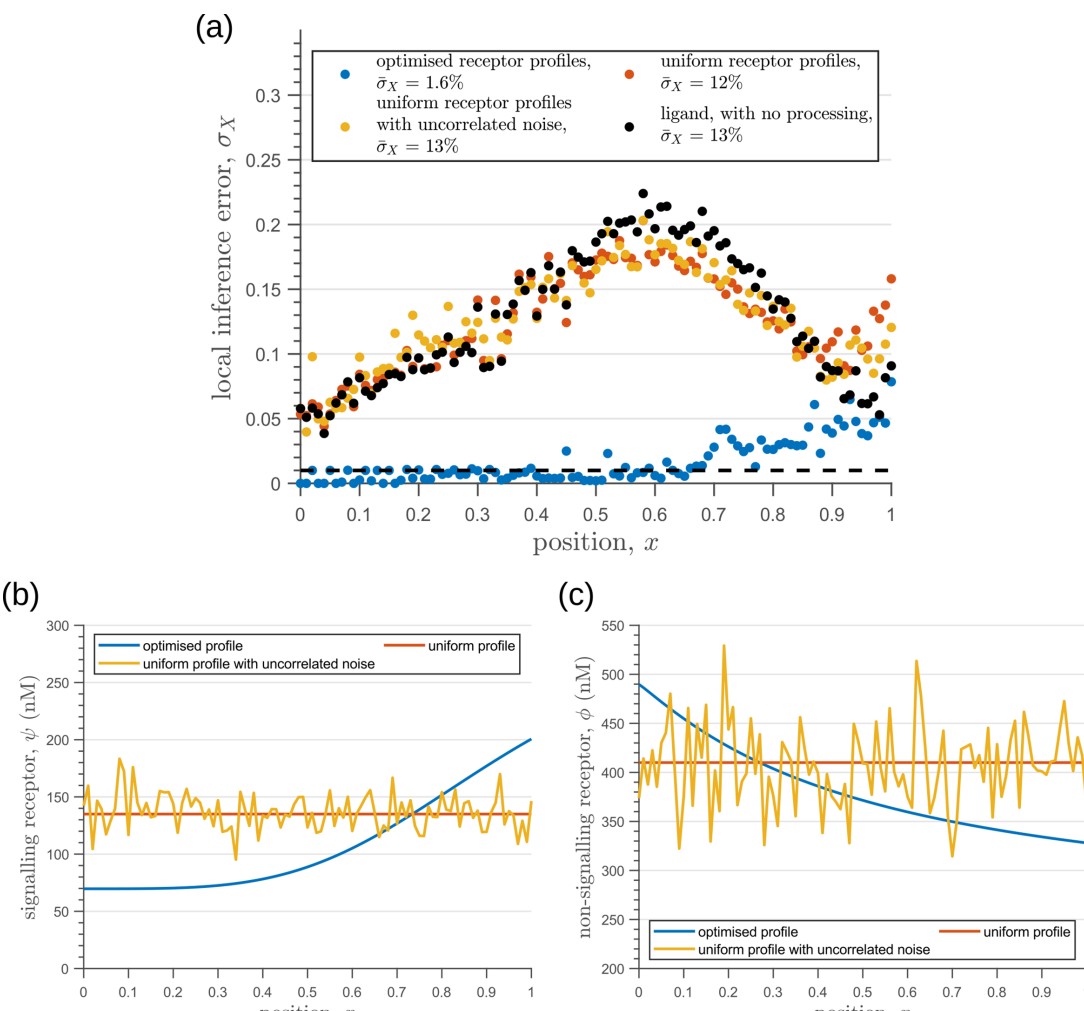

**Appendix 11—figure 1.** Behaviour of inference error on varying receptor profiles. (**a**) Inference error profiles due to extrinsic noise in the optimised two-tier two-branch channels with optimal receptor profiles (blue), uniform receptor profiles (red) and uniform receptor profiles with uncorrelated noise (orange). Having uniform receptor profiles simply reflects the noise in the ligand input (black). (**b**) Signalling receptor profiles corresponding to the cases in (**a**). (**c**) Non-signalling receptor profiles corresponding to the cases in (**a**). In (**b,c**), the mean uniform receptor concentration is set to the mid point of the optimised receptor profile while the strength of the uncorrelated noise is 0.1 times the mean. The chemical rates, receptor profile parameters and feedback parameters for the optimised two-tier two-branch channel can be found in **Table 2** under the column corresponding to $n_T = 2, n_B = 2, r_- = \kappa_I$. The optimised chemical rates and feedback parameters for the two-tier two-branch channels with uniform receptors can be found in **Table 3**.

## Appendix 12

### Dependence of inference errors on input characteristics

The general qualitative features of the optimised channels remain invariant to changes in input characteristics. We find the same feedback topology and qualitative results when the one-tier two-branch channel architecture is optimised for input distributions with different decay lengths of mean ligand input $\lambda$ (*Appendix 12—figure 1*). Additionally, lowering noise in the ligand input reduces the inference error of the optimised channel and extends the region of robustness in the tissue (*Appendix 12—figure 2*).

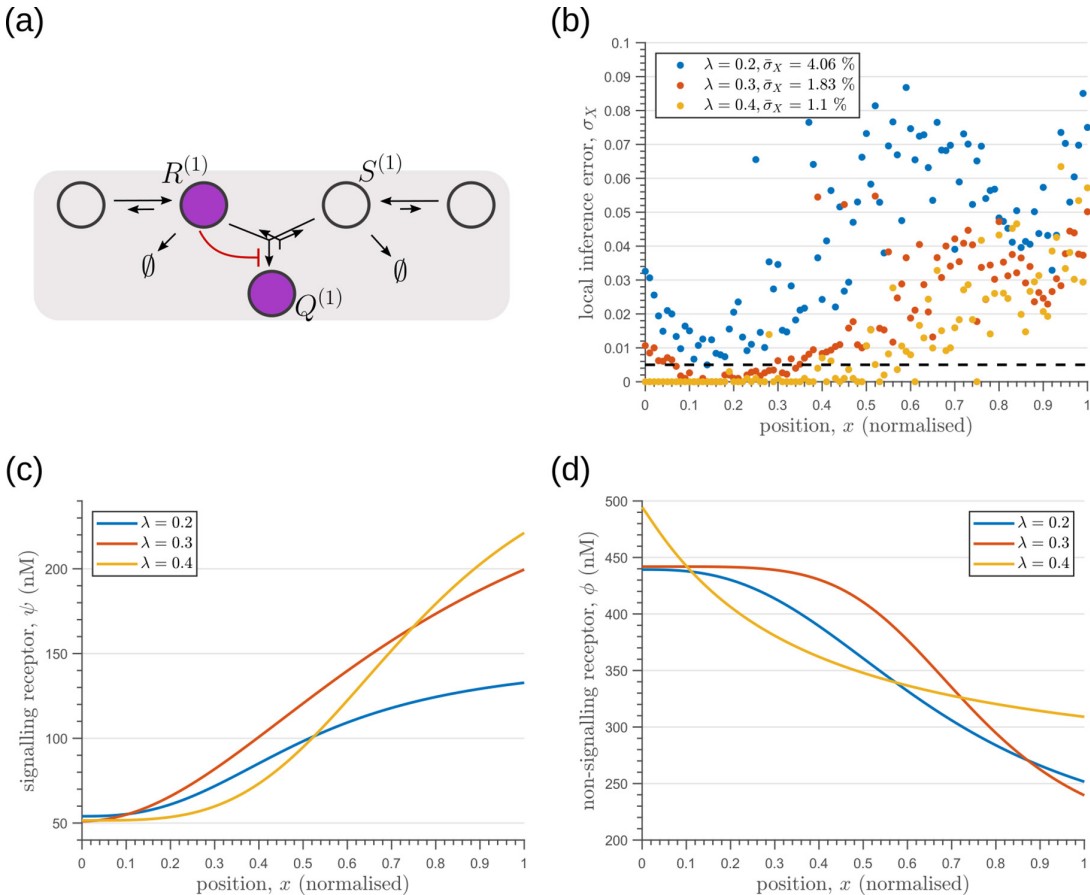

**Appendix 12—figure 1.** Optimisation of one-tier two-branch channels for extrinsic noise with varying mean input decay lengths $\lambda$ (*Equation 2*). (**a**) The channel architecture with inter-branch feedback shows the lowest inference error for all values of $\lambda$ considered. (**b**) The minimum local and average inference errors decrease with $\lambda$. (**c**) Optimised profiles of the signalling receptors are increasing functions of $x$ for the different values of $\lambda$. (**d**) Optimised profiles of the non-signalling receptors are decreasing functions of $x$ for the different values of $\lambda$.

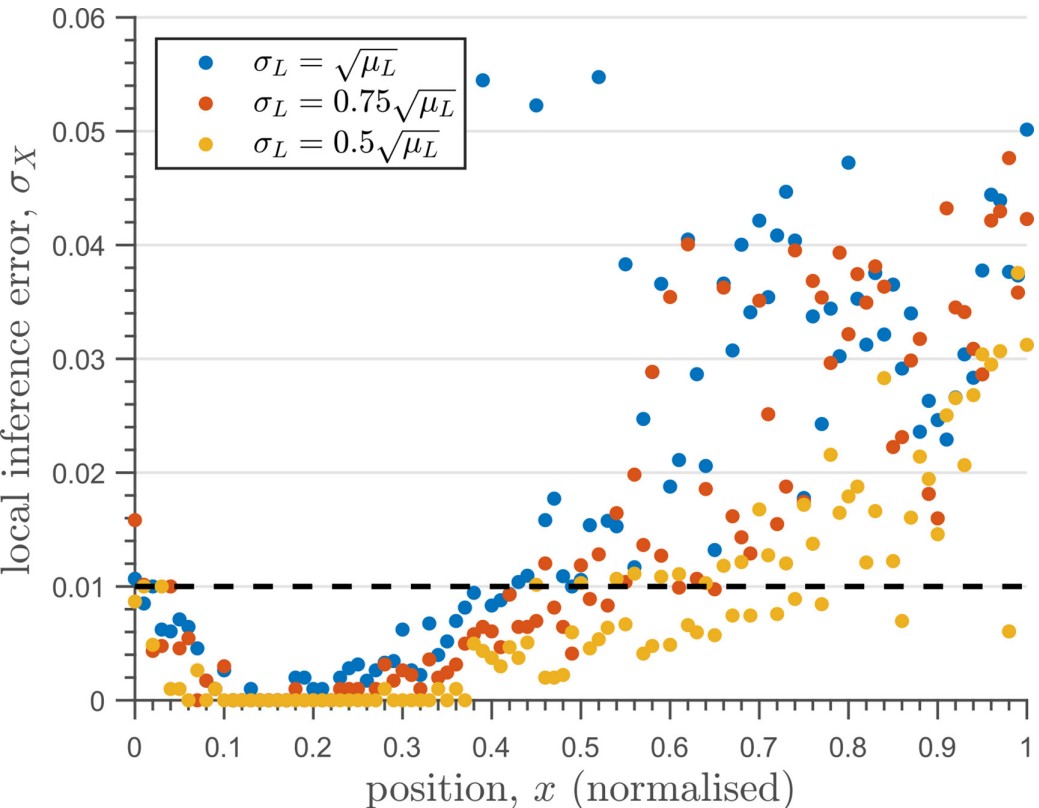

**Appendix 12—figure 2.** The one-tier two-branch channel optimised for ligand distribution with standard deviation equal to the square root of mean, $\sigma_L = \sqrt{\mu_L}$, and decay length $\lambda = 0.3$ shows smaller inference error for lower levels of input noise. The dashed line corresponds to a local inference error of one cell's width $\sim 1/n_x$. Note that the point at which local inference error departs away from 1% (one cell width error) extends further away from the source.

## Appendix 13

### Response in inference error due to perturbations in receptor concentrations

The definition of cellular output $\theta$ in a branched architecture involves making a distinction between the signalling and non-signalling receptors. In conjunction, the direction of the inter-branch feedback is from the signalling to the non-signalling branch (*Figure 8b* of the main text). This gives rise to the possibility of an asymmetric response of the optimised channel to perturbations in the two receptors around the optimal point. *Appendix 13—figure 1a* shows the average inference error $\bar{\sigma}_X$ due to the receptor profiles in *Appendix 13—figure 1b, c* resulting from perturbations in the receptor control parameters $A_2$ and $B_2$ (*Table 1*, *Equations 8; 9*) around their optimal values. Each perturbed receptor profile (black curves in *Appendix 13—figure 1b, c*) leads to a net deviation $\Delta\psi, \Delta\phi$ from the optimal receptor profile (blue curves in *Appendix 13—figure 1b, c*), which is computed as follows.

$$\Delta\psi = \int dx \, (\psi(x; A_2^* + \delta A_2) - \psi(x; A_2^*)) \tag{45}$$

$$\Delta\phi = \int dx \, (\phi(x; B_2^* + \delta B_2) - \phi(x; B_2^*)) \tag{46}$$

where $A_2^*$ and $B_2^*$ are the optimum values, and $\delta A_2, \delta B_2$ are the perturbations in the receptor control parameters. This is simply the signed area between the optimised and perturbed receptor profiles. Note that the perturbed receptor profiles are such that they maintain the nature of monotnonicity. The local curvature of $\bar{\sigma}_X(\Delta\psi, \Delta\phi)$ around the optimal point ($\Delta\psi = \Delta\phi = 0$, red point in *Appendix 13—figure 1a*) in the $\psi - \phi$ plane has eigenvalues $\lambda_1 \simeq 0.016, \lambda_2 \simeq 6.1$ corresponding to the eigenvectors that are nearly parallel to $\Delta\phi$ and $\Delta\psi$ axes, respectively (white arrows in *Appendix 13—figure 1a*). This indicates that the inference error is much more sensitive to changes in the signalling receptor $\psi$ than to changes in the non-signalling receptor $\phi$, implying a *stiff* direction of control along the former and a *sloppy* direction of control along the latter.

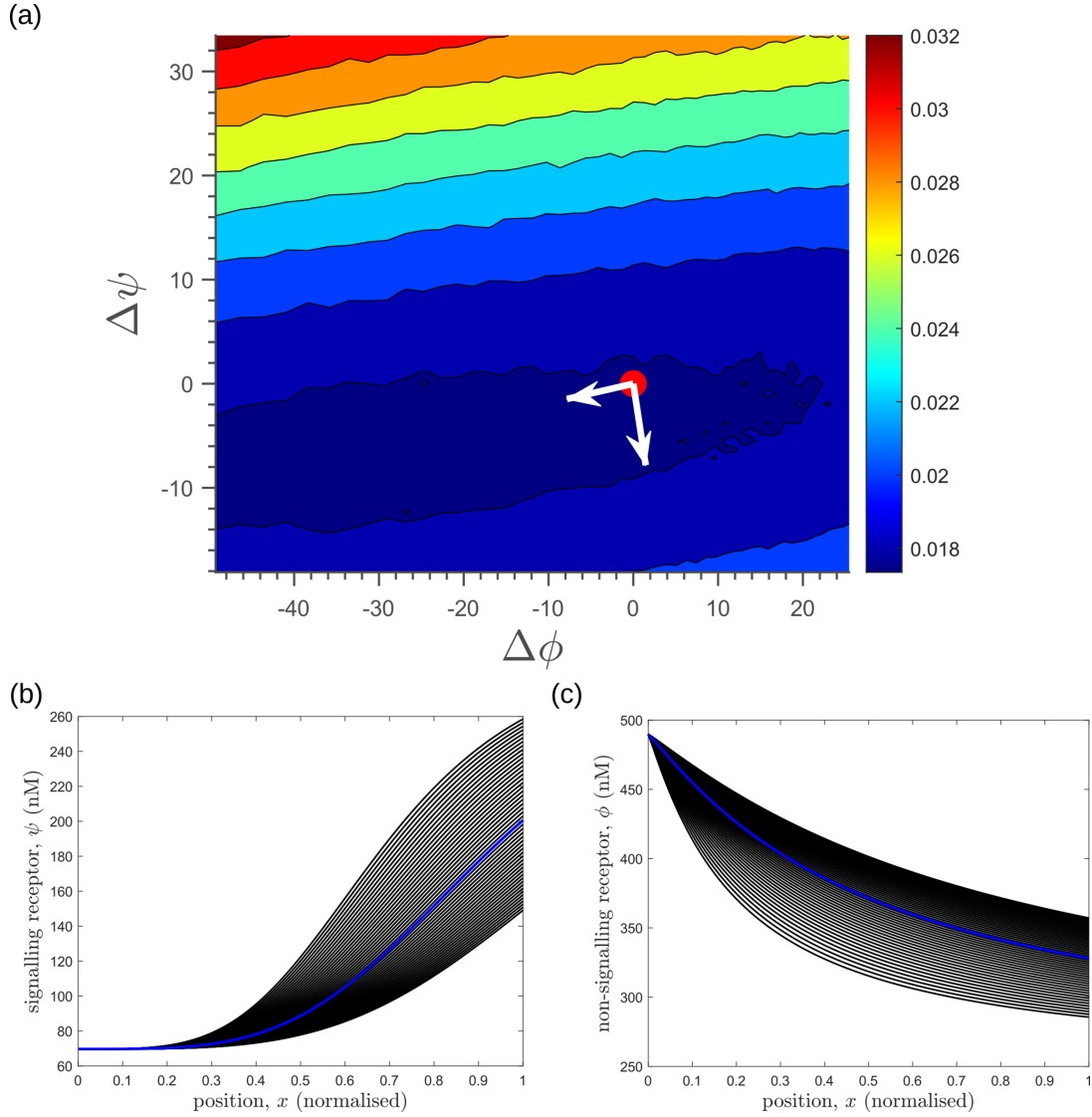

**Appendix 13—figure 1.** Response of average inference error in the optimised two-tier two-branch channel (*Figure 8b* of the main text) to changes in receptor profiles shows the *stiff-sloppy* directions of control on receptors. (**a**) Contours of average inference error $\bar{\sigma}_X$ as functions of the net deviation from the optimal receptor profiles, as defined in *Equations 45; 46*. The white arrows indicate the directions of eigenvectors of the local Hessian (curvature) around the optimum (red point). The shorter arrow corresponds to the smaller eigenvalue (*sloppy* direction) while the longer arrow corresponds to the larger eigenvalue (*stiff* direction). (**b,c**) The allowed perturbations in receptor profiles, $\psi(x)$ and $\phi(x)$ (black) around the optimal receptor profiles (blue), maintaining the nature of monotonicity. The optimised parameter values for the plots can be found in *Table 2* under the column corresponding to $n_T = 2, n_B = 2, r_- = \kappa_I$.

## Appendix 14

### Distribution of optimum channel parameters

In Section "Geometry of fidelity landscape", we commented on the nature of low inference error landscape as defined by optimum channel parameters that yield an average inference error $\bar{\sigma}_X \leq 2\%$. Of the 16 channel parameters, we showed six parameters corresponding to ligand binding rates to the signalling and non-signalling receptors, conjugation and splitting rates, and feedback sensitivity and feedback strength. These parameters were *stiff*, that is small changes in these parameters led to strong variations in the inference error. For completeness, here we present the frequency distributions of all the optimum channel parameters that yield an inference error of $\bar{\sigma}_X \leq 2\%$. While some of these parameters are narrowly distributed about the upper or lower bounds of their permissible ranges, others are more broadly distributed across the range.

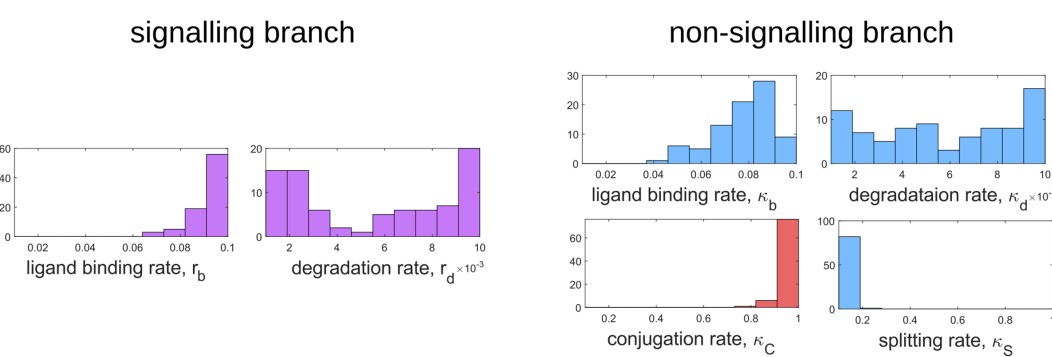

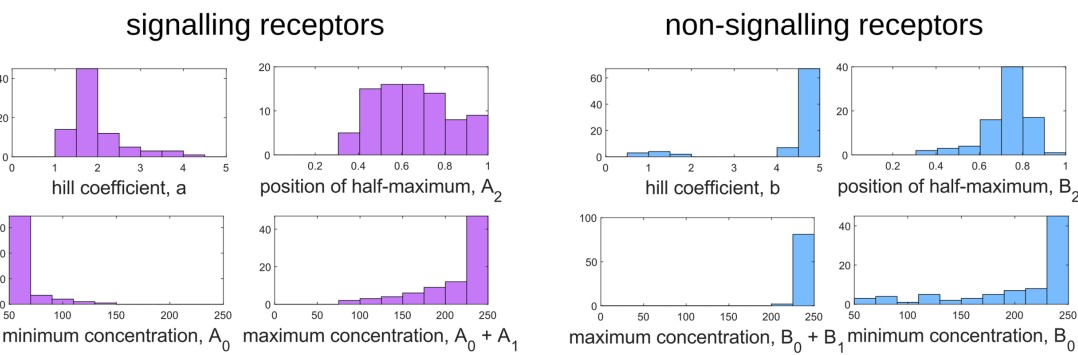

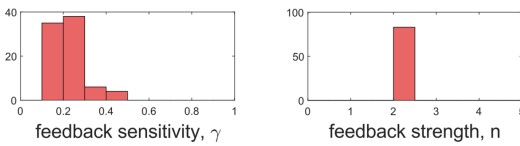

**Appendix 14—figure 1.** Frequency distributions of optimum channel parameters yielding $\bar{\sigma}_X \leq 2\%$. Symbols below each panel represent channel parameters listed in *Table 1*.

## Appendix 15

### Experimental methods

#### Fly stocks, endocytic assays, and imaging

Fly stocks used in this study are wildtype w1118, c5-GAL4, CAAX-GFP (Kyoto DGRC - 109823), *Port et al., 2014* and UAS-myr-Garz-E740K. Flies were reared on corn flour medium containing sugar, yeast and agar along with antibacterial and antifungal agents. Flies were grown in 25°C incubators with 12 hr light/dark cycles for experiments and otherwise maintained at 18°C or 22°C. Third instar larval wing discs were dissected in Grace's live imaging media *Dye et al., 2017*. Dissected discs were incubated with labelled Wg antibodies (Wg-AF-568) on ice for 45 min. Discs were transferred to room temperature for indicated time of pulse, washed with ice cold 1XPBS buffer and fixed using 4% PFA (5 min on ice +15 min at room temperature). Discs were then mounted in imaging chambers and imaged on a FV3000 laser-scanning confocal microscope using a 60 X/1.42 NA oil objective (with acquisition XY pixel dimensions of $0.138 \mu m$ and Z stacks of size $0.5 \mu m$).

#### Analysis

The slice-by-slice images of the dome shaped wing disc from the confocal microscope were transformed into images from the outer most surface of the wing disc. This allowed us to compare the intensities of Wg from similar apico-basal height of cells across the dome-shaped epithelial. Wg production plane (Plane Q) and a perpendicular plane (Plane R) were defined. Intensity of different probes in curved tissues is affected by sample geometry and imaging depth. A data-based correction matrix was constructed using a uniform marker – CAAX GFP expressed uniformly under a ubiquitin promoter. Intensity for each disc, for each probe, was corrected using this data-based correction matrix. Detailed experimental and analysis methodology for extracting gradients from a curved tissue is described in *Prabhakara et al., 2022*. For computing the coefficient of variation, 18 bins parallel to Plane R were defined and intensities at different distances from the production plane was computed (schematic in *Figure 14c* of the main text). Mean and standard deviation (SD) for each disc across multiple parallel bins at different distances from the producing cells $x$ were used to estimate the coefficient of variation.

$$\mathrm{CV}(x) = \frac{\mathrm{SD}(x)}{\mathrm{Mean}(x)} \tag{47}$$

Computation of CV was done using intensity collected from the apical region of cells (20% of the entire length of cells). Normalized distance from the production plane is represented in all plots. Here, we have considered dorsal and ventral gradients to be equivalent.

