## [Editor Report]

The manuscript introduces a compelling theoretical framework to investigate architectures of signal processing. The predictions of the computational model have been convincingly validated with data from fly wing precursor tissues. The work is important and will be highly valuable to biological physicists and developmental biologists interested in morphogenesis and pattern formation.

---

## [Decision Letter]

**Decision letter after peer review:**

Thank you for submitting your article "Cellular compartmentalisation and receptor promiscuity as a strategy for accurate and robust inference of position during morphogenesis" for consideration by *eLife*. Your article has been reviewed by 2 peer reviewers, and the evaluation has been overseen by a Reviewing Editor and Naama Barkai as the Senior Editor. The following individual involved in the review of your submission has agreed to reveal their identity: Marcin Zagorski (Reviewer #2).

Essential revisions:

Both reviewers and reviewing editor agreed that no additional data is needed to support the major conclusions. Therefore a revision focused on improving clarity and presentation will be sufficient. The following two major concerns should be addressed; additional detailed comments and suggestions in the reviews below should be considered for clarity and overall readability.

1) One important point to clarify is whether the presented solutions (parameters identified) are representative of a larger class of sub-optimal or optimal solutions. This can be checked by perturbing the solutions. Presumably, the authors have investigated this point already as in the SM there are parts about, robustness, sensitivity and trade-offs, so improving presentation in the main part should solve that. This will also shed more light on the generality of the obtained results.

2) The important question to address is whether the choice of Eq. 5, which describes how the cell evaluates its position, influences other results in the manuscript.

*Reviewer #1 (Recommendations for the authors):*

– Extrinsic noise coming from the stochastic ligand production at the source is included in the model by letting cells along the direction parallel to the source boundary experience a spatially-fluctuating level of ligand, with statistics given in (1). From what we understand, this noise is quenched in space. Is it clear to what extent this is equivalent to the more realistic annealed disorder originating from Brownian motion and stochastic source production?

– In some cases, non-specific receptors (e.g. Dlp, Hanandal, Development, 2005) have been shown to increase in expression levels away from the morphogen source. Can the authors comment on this observation in light of their model?

– In appendix L, the local inference error for a (optimised) graded receptor expression is compared to that of a uniform receptor expression pattern. We find that the way this result is presented slightly misleading since the expression level in the uniform case is not optimised; the two setups should be compared after optimisation.

*Reviewer #2 (Recommendations for the authors):*

– In my opinion, Figure 2 is misleading and is disrupting the flow of the manuscript. Almost the same information is conveyed in Figure 4 and Figure 5. Further Figure 2 suggests some very regular arrangement of the nodes (regular topology of signalling architecture), which is not the case. Presenting a model with tiers and branches, or some different network-like schematic to indicate reaction and flux imbalances could improve presentation.

– In lines 85-87, different timescales of signalling processing are mentioned corresponding to branches (fast) and tiers (slow), but this aspect of regulation seems to be not discussed in the later parts of the paper. It might be worth drawing this analogy again when discussing how noise is integrated by different architectures. Possibly, there is the separation of the timescales or signalling integration takes place on the same timescales.

– Line 98, Figure 5 is mentioned just after Figure 2 (line 91). Please amend the ordering or rephrase so figures are referenced in the text as they appear in the paper.

– Line 114, I would avoid statements "of course any choice consistent with experimental observations would do". For instance, would the model work for stationary wave-like patterns of morphogens that could emerge via Turing mechanism (Green and Sharpe, Development 2015)? Basically, I would rephrase providing a description of acceptable ligand profiles (e.g. monotonically decreasing).

– The optimization procedure employed is computationally quite-consuming as the intrinsic noise is calculated by solving a chemical master equation. Can the model be solved without directly solving CME? What are the differences? If the differences with and without CME are small this might help to have faster optimization and hence more explicitly explore the space of available signalling architectures that result in optimal or close to the optimal solution.

– The intrinsic and extrinsic noises are varied separately as far as I understand. Are there any arguments/heuristics that would indicate the resulting global solutions would be the same if two types of noise would be varied simultaneously?

– In appendix C, the screening is described for the model parameters, but it is still slightly obscure what process was carried out.

– In table 2, in the middle column, we see that the weight of tier 2 is orders of magnitude greater than that for tier 1. In this case, what role does the first tier really play? And does such a strong weighting make sense biologically?

– I don't really understand the need for appendix E, but I might be missing something.

– In Appendix G, Figure 8, we see that adding a third tier does not reduce the minimum average inference error very much at all. Were any simulations done for four-tiered systems? What sort of computational cost does adding one extra tier there?

---

## [Author Response]

Essential revisions:Both reviewers and reviewing editor agreed that no additional data is needed to support the major conclusions. Therefore a revision focused on improving clarity and presentation will be sufficient. The following two major concerns should be addressed; additional detailed comments and suggestions in the reviews below should be considered for clarity and overall readability.1) One important point to clarify is whether the presented solutions (parameters identified) are representative of a larger class of sub-optimal or optimal solutions. This can be checked by perturbing the solutions. Presumably, the authors have investigated this point already as in the SM there are parts about, robustness, sensitivity and trade-offs, so improving presentation in the main part should solve that. This will also shed more light on the generality of the obtained results.

We thank the reviewers/editor for suggesting this. Indeed the original manuscript has a detailed discussion on sensitivity and robustness to variations in receptor concentrations (Section “Asymmetry in branched architecture: promiscuity of non-signalling receptors”, Appendix 13 and paragraph 6 of Discussion in the revised manuscript). We have now included an additional Subsection “Geometry of fidelity landscape” of Results, where we have described the geometry of the *fidelity landscape* (inference error) around the optimum and the low inference error states. This has involved additional computation and we have presented our detailed results on this as Figure 10 and Figure 11.

Briefly, we measure (i) percent change in inference error upon perturbation of the channel parameters, and (ii) the Fisher information metric (FIM) around the optimum. We use the optimised one-tier two-branch channel for this analysis which yields a minimum inference error of σ¯x∼1.9% i.e. approximately two cell widths. We see that the inference error does not change significantly with most parameters, i.e. it remains within σ¯x≤2.2% margin, except for the feedback strength *n*. The eigenspectrum of the FIM shows only a few *stiff* directions (eigenvectors with high eigenvalues), which have a significant component along the feedback parameters.

We now explicitly connect this detailed perturbative analysis in channel parameters to the discussion on perturbations of receptor profiles in Appendix 13.

To address the geometry of low inference error states, we look at the frequency distributions of channel parameters in a small cutoff region around the minimum identified, i.e. σ¯x≤2%. We find that the frequency distribution of optimal feedback parameters whose inference error lies within the range 1.9%≤σ¯x≤2% is narrowly distributed about the global optimum. Parameters corresponding to forward and backward rates are skewed towards the upper and lower bounds of the allowed parameter range respectively. All other optimal parameters corresponding to degradation rates, minimum and maximum receptor values and steepness of the receptor profiles, show a very broad spread over this range. Further, position vector of the optimum parameters lie predominantly along the sloppy directions of the Hessian evaluated at the global minimum. This suggests that geometry of the low inference error landscape resembles a narrow valley, which is shallow along several *sloppy* directions and steep along the few *stiff* directions.

This has now been included as a part of Subsection “Geometry of fidelity landscape” in the Results section along with new Figures 10 and 11 in the revised manuscript.

2) The important question to address is whether the choice of Eq. 5, which describes how the cell evaluates its position, influences other results in the manuscript.

We thank the reviewers for this comment. We presume here that the question refers to Eq. 6 since this describes the choice of objective function and how the cell evaluates its position. On the other hand, Eq. 5 refers to a standard inference of position from an output distribution computed by the cell. We discuss the assumptions behind Eq. 5 in our response to Reviewer #1’s first question.

Indeed the original manuscript contains a discussion on the choice of objective function i.e. inference error (last paragraph of Section “Mathematical Framework” and subsection “Future directions” of the Discussion in the revised manuscript). To highlight this important point, we have now included Subsection “ Choice of objective function” of Results on the choice of objective function. With the alternate objective function which specifies positions of certain *regions* or *cell fate boundaries*, we observe that the general, qualitative results remain the same, i.e. an additional branch helps reduce the inference errors due to extrinsic noise and an additional tier helps with the same when faced with intrinsic noise. Along with these results, we see that the receptor profiles too remain qualitatively the same as before.

Reviewer #1 (Recommendations for the authors):– Extrinsic noise coming from the stochastic ligand production at the source is included in the model by letting cells along the direction parallel to the source boundary experience a spatially-fluctuating level of ligand, with statistics given in (1). From what we understand, this noise is quenched in space. Is it clear to what extent this is equivalent to the more realistic annealed disorder originating from Brownian motion and stochastic source production?

In our problem setup, each cellular output is calculated from the steady-state distribution of the morphogen concentration in its bound and unbound states within the cell. This implicitly assumes that the relaxation timescale of morphogen concentrations to its steady-state is faster than the timescales associated with individual cellular processing. This is a fair assumption as the timescales associated with diffusive transport (0.1 – 0.01s) and extracellular degradation (seconds to minutes) of the morphogen are faster than cellular processes which are in the order of minutes to tens of minutes.

These estimates follow from FCS measurements on the diffusion of free Wg (manuscript in preparation by C Prabhakara *et al*) and for free Dpp [10] in *Drosophila* wing discs.

– In some cases, non-specific receptors (e.g. Dlp, Han et al, Development, 2005) have been shown to increase in expression levels away from the morphogen source. Can the authors comment on this observation in light of their model?

We thank the reviewer for pointing out this important paper by Han *et al.* These authors demonstrate that there are *two* non-signalling receptors, Dally and Dlp, involved in the Wg signalling, one of which Dlp interacts strongly with Wg and facilitates its sculpting. As mentioned in response to Reviewer #1 (comment #4), in the current manuscript, we have treated the morphogen profiles as *independent* of the receptor dynamics and hence cannot address the issue of morphogen sculpting. We will take up this important issue in a separate study.

– In appendix L, the local inference error for a (optimised) graded receptor expression is compared to that of a uniform receptor expression pattern. We find that the way this result is presented slightly misleading since the expression level in the uniform case is not optimised; the two setups should be compared after optimisation.

We have indeed optimised the chemical rates and feedback parameters in the two-tier two-branch channels with spatially uniform receptor profiles. We have modified the corresponding appendix (Appendix 11 in the revised manuscript) and included Table 3 in the revised manuscript to emphasize this.

In arriving at the optimised channel characteristics for a given morphogen profile, we go through all possible monotonic receptor profiles *including flat profiles*. Thus we know that any deviation from the optimum will lead to inferior positional inferences. However, the question posed in Appendix 11 of the revised manuscript is the following: “why can’t a flat receptor profile (modified by uncorrelated noise) infer positions accurately from a noisy morphogen gradient?”. If the morphogen gradient was not corrupted by noise, then surely flat receptor profiles would have sufficed. It is because one wants to discriminate between morphogen concentrations in neighbouring cells in *a noisy background* that the you need a dynamic and spatially varying profile for the receptors. As seen from Appendix 11—figure 1 (of the revised manuscript), the inference from flat receptor profiles reflect the noise in morphogen gradient itself.

We elaborate on this point in Appendix 11 in the revised manuscript.

Reviewer #2 (Recommendations for the authors):– In my opinion, Figure 2 is misleading and is disrupting the flow of the manuscript. Almost the same information is conveyed in Figure 4 and Figure 5. Further Figure 2 suggests some very regular arrangement of the nodes (regular topology of signalling architecture), which is not the case. Presenting a model with tiers and branches, or some different network-like schematic to indicate reaction and flux imbalances could improve presentation.

We agree with the reviewer on this point. Thanks to the suggestions, we have now modified the Figure 2 in the revised manuscript. We hope updated figure is more appealing and conveys the point better.

– In lines 85-87, different timescales of signalling processing are mentioned corresponding to branches (fast) and tiers (slow), but this aspect of regulation seems to be not discussed in the later parts of the paper. It might be worth drawing this analogy again when discussing how noise is integrated by different architectures. Possibly, there is the separation of the timescales or signalling integration takes place on the same timescales.

The timescale separation in signal processing is between the direction of chemical reactions (receptor state transitions) of the same receptor type i.e. within the same branch (fast, horizontal) and the direction of transport corresponding to tiers (slow, vertical). This is reflected in our choice of parameter values (listed in Table 1, consistent with experimental values) over which all our optimisation is carried out. The unbinding rate, which corresponds to transition in receptor state, is an order of magnitude faster than the internalisation and recycling rates, which correspond to intracellular transport processes.

We have added a line to this effect in the first paragraph of the Results section in the revised manuscript.

– Line 98, Figure 5 is mentioned just after Figure 2 (line 91). Please amend the ordering or rephrase so figures are referenced in the text as they appear in the paper.

Thank you. We have attended to this point in the revised manuscript.

– Line 114, I would avoid statements "of course any choice consistent with experimental observations would do". For instance, would the model work for stationary wave-like patterns of morphogens that could emerge via Turing mechanism (Green and Sharpe, Development 2015)? Basically, I would rephrase providing a description of acceptable ligand profiles (e.g. monotonically decreasing).

Indeed, the morphogen profiles applicable here belong to the family of monotonically behaving functions. We have attended to this point in the revised manuscript.

– The optimization procedure employed is computationally quite-consuming as the intrinsic noise is calculated by solving a chemical master equation. Can the model be solved without directly solving CME? What are the differences? If the differences with and without CME are small this might help to have faster optimization and hence more explicitly explore the space of available signalling architectures that result in optimal or close to the optimal solution.

Solving the CME is the most accurate approach which is applicable in any parameter regime or any network topology. In some conditions such as low (additive) noise, other approaches e.g. solving an equivalent Langevin equation, may solve for steady-state faster with minor deviations from the CME solution. However, there is no guarantee that the intrinsic noise is small in any given network topology and across the parameter range.

– The intrinsic and extrinsic noises are varied separately as far as I understand. Are there any arguments/heuristics that would indicate the resulting global solutions would be the same if two types of noise would be varied simultaneously?

Yes, the extrinsic and intrinsic noises have been considered separately in our work. As the reviewer would appreciate, doing an optimisation considering intrinsic and extrinsic noises simultaneously is computationally exorbitant. Instead, for a fixed number of tiers and branches, we first optimise the channel architecture subject to extrinsic noise alone over all possible rates, receptor profiles and feedback topologies. For each of these optimised channels with fixed number of tiers and branches (e.g. as given by Table 2), we calculate the inference errors in face of intrinsic noise. By doing so, we compare *robustness* of each such optimised channel’s performance to intrinsic noise. This is appropriate considering the timescale of variation in the extracellular morphogen concentrations is taken to be faster than the cellular processing timescales, as discussed in our response to Reviewer #1’s comment #6.

This is the scope of our exercise. Were we to address the *different* question of what is the optimum channel characteristics subject to both extrinsic and intrinsic noises then we would have had to perform a simultaneous optimisation.

– In appendix C, the screening is described for the model parameters, but it is still slightly obscure what process was carried out.

Thank you for the suggestion. We have now made minor edits to Appendix 3 in the revised manuscript for clarity.

– In table 2, in the middle column, we see that the weight of tier 2 is orders of magnitude greater than that for tier 1. In this case, what role does the first tier really play? And does such a strong weighting make sense biologically?

As discussed in the original manuscript (Section “Quantitative models for cellular reading and processing” of the revised manuscript), the weights associated with a tier correspond to the residence time of the signal transducing molecules (or secondary messengers) in that tier. We optimise over these weights along with the other channel parameters. A large weight to the second tier in an optimised channel would imply that signal transduction takes place predominantly from the second tier. The first tier, nevertheless, is still important in generating the characteristics of the second tier e.g. through actuating the feedback (or feedforward) control.

The Wg signalling in *Drosophila* wing disc provides a concrete example. Hemalatha *et al.*, 2016 [4], Seto *et al.*, 2006 [11] and Rives *et al.*, 2006 [12] have shown that endocytosis is important for signalling. Removal of different components of the endocytic network abrogates both short range as well as long range Wg signalling. Intracellular Wg endosomes often appear to be co-localized with the downstream signalling mediator Dishevelled (Dsh). However, a quantitative study comparing the extent of localization of Dsh at the plasma membrane versus endosomal membrane is currently missing. Given the relevance of the endocytic network in Wg signalling and co-localization of downstream adaptor proteins to endosomes containing Wg-DFz2, we believe that these endosomes (corresponding to the second tier) act as signalling hubs to transduce positional information.

– I don't really understand the need for appendix E, but I might be missing something.

In Appendix 5, we are trying to understand why having an appropriate spatial gradient of receptors helps in separating the outputs in nearby cells, thus facilitating accurate positional inference. This is best done in the simplest case of a *minimal* channel i.e. with one tier and one branch.

We have now made minor edits to Appendix 5 in the revised manuscript to reflect the above.

– In Appendix G, Figure 8, we see that adding a third tier does not reduce the minimum average inference error very much at all. Were any simulations done for four-tiered systems? What sort of computational cost does adding one extra tier there?

The number of network topologies increases as a quadratic function of the number of tiers. Further, the number of parameters to be optimised would also increase linearly with the number of tiers, resulting in a net cubic increase in computation time. Practically, even with parallelisation, the number of computing hours (in real time) would increase significantly with additional tiers. From our analysis of two- and three-tier channels, we would predict that further addition of tiers provides a marginal advantage in reduction of inference errors while incurring additional cellular costs.

References